# SARS-CoV-2 infection induces DNA damage, through CHK1 degradation and impaired 53BP1 recruitment, and cellular senescence

Ubaldo Gioia [1,11], Sara Tavella [1,11], Pamela Martínez-Orellana[2], Giada Cicio[1,3], Andrea Colliva[2], Marta Ceccon[1], Matteo Cabrini[1], Ana C. Henriques[1], Valeria Fumagalli [4], Alessia Paldino[2,5], Ettore Presot [6], Sreejith Rajasekharan[2,10], Nicola Iacomino[7], Federica Pisati[8], Valentina Matti[1], Sara Sepe[1], Matilde I. Conte[1], Sara Barozzi[1], Zeno Lavagnino [1], Tea Carletti [2], Maria Concetta Volpe[2], Paola Cavalcante[7], Matteo Iannacone [4], Chiara Rampazzo [6], Rossana Bussani[5], Claudio Tripodo [1,3], Serena Zacchigna [2,5], Alessandro Marcello [2] & Fabrizio d'Adda di Fagagna [1,9] ✉

Severe acute respiratory syndrome coronavirus 2 (SARS-CoV-2) is the RNA virus responsible for the coronavirus disease 2019 (COVID-19) pandemic. Although SARS-CoV-2 was reported to alter several cellular pathways, its impact on DNA integrity and the mechanisms involved remain unknown. Here we show that SARS-CoV-2 causes DNA damage and elicits an altered DNA damage response. Mechanistically, SARS-CoV-2 proteins ORF6 and NSP13 cause degradation of the DNA damage response kinase CHK1 through proteasome and autophagy, respectively. CHK1 loss leads to deoxynucleoside triphosphate (dNTP) shortage, causing impaired S-phase progression, DNA damage, pro-inflammatory pathways activation and cellular senescence. Supplementation of deoxynucleosides reduces that. Furthermore, SARS-CoV-2 N-protein impairs 53BP1 focal recruitment by interfering with damage-induced long non-coding RNAs, thus reducing DNA repair. Key observations are recapitulated in SARS-CoV-2-infected mice and patients with COVID-19. We propose that SARS-CoV-2, by boosting ribonucleoside triphosphate levels to promote its replication at the expense of dNTPs and by hijacking damage-induced long non-coding RNAs' biology, threatens genome integrity and causes altered DNA damage response activation, induction of inflammation and cellular senescence.

Severe acute respiratory syndrome coronavirus 2 (SARS-CoV-2) is an RNA virus, responsible for the ongoing coronavirus disease 2019 (COVID-19) pandemic[1]. Its 30 kb genome encodes 26 polypeptides encompassing 16 non-structural proteins (NSPs), 4 structural proteins such as the nucleocapsid (N) protein, and 6 accessory ones[2].

Viral infections can impact on several cellular pathways, including the autophagy pathway[3], the ubiquitin–proteasome system (UPS)[4] and the DNA damage response (DDR). While the interplay between some DNA viruses and DDR has been studied[5], much less is known about RNA viruses[6]. Although SARS-CoV-2 infection has been suggested to engage

components of the DDR machinery[7–10], a thorough characterization and a mechanistic probing of the impact of SARS-CoV-2 on genome integrity and DDR engagement is lacking.

The DDR is a network of pathways that sense DNA lesions, signal their presence and coordinate their repair[11]. DNA single-strand and double-strand breaks (SSBs and DSBs) are detected by replication protein A (RPA) and by the MRE11–RAD50–NBS1 (MRN) complex[12], respectively, which guide the recruitment of the apical DDR kinases ataxia telangiectasia and Rad3-related (ATR) or ataxia-telangiectasia mutated (ATM) at SSBs or DSBs[12], respectively. ATR and ATM undergo autophosphorylation and phosphorylate several DDR factors, including the effector kinases CHK1 and CHK2, which contribute to enforce cell-cycle arrest[12]. DDR activation can cause cellular senescence[13,14] and inflammation[15] or cell death[13].

We demonstrated that the induction of a DSB results in the recruitment of the RNA polymerase II complex, which transcribes a novel class of RNA molecules named damage-induced long non-coding RNAs (dilncRNAs)[16–18]. These RNAs, by interacting with DDR factors such as p53-binding protein 1 (53BP1), are necessary for their condensation into foci at DSBs[16,17] by promoting liquid–liquid phase separation (LLPS)[17,19]. Inhibiting the synthesis or function of dilncRNAs disrupts DDR foci and impairs DNA repair[16,17,20]. Interestingly, also SARS-CoV-2 N-protein phase-separates in an RNA-dependent manner[21,22].

In this Article, we demonstrate that SARS-CoV-2 infection causes DNA damage and activation of an altered DDR. DNA damage is the consequence of the degradation of CHK1 by ORF6 and NSP13 viral factors through the proteasome and the autophagy pathways, respectively. Depletion of CHK1 causes loss of RRM2, a component of the ribonucleotide reductase (RNR) complex[23], which leads to deoxynucleoside triphosphate (dNTP) shortage that causes impaired S-phase progression, DNA damage accumulation, DDR activation, induction of inflammatory pathways and establishment of cellular senescence. The supplementation of deoxynucleosides (dNs) is sufficient to contrast this cascade of events. In addition to that, SARS-CoV-2 N-protein impairs 53BP1 recruitment at DSB by competing with dilncRNAs binding, ultimately hampering DNA repair. These events occur also in vivo in mice infected by SARS-CoV-2 and in patients with COVID-19.

Overall, our results reveal the impact of SARS-CoV-2 infection on genome integrity and its contribution to the inflammatory response observed in COVID-19 patients and the recently reported virus-induced cellular senescence[24].

## Results

### SARS-CoV-2 causes DNA damage and an altered DDR activation

We studied the engagement of the DDR pathways at different timepoints upon infection by SARS-CoV-2 of Huh7 cells, a human cell line naturally permissive to SARS-CoV-2 (refs. [25,26]), by immunoblotting of whole cell lysates. As negative control we used mock-infected cells; as positive control we exposed cells to hydroxyurea (HU), which induces DNA replication stress and activates the ATR–CHK1 axis[27,28], or ionizing radiation (IR) that causes DSBs and activates the ATM–CHK2 pathway[11] (Fig. 1a). We observed that SARS-CoV-2 infection triggered the autophosphorylation, and thus activation, of the master kinases DNA-PK (pDNA-PK[S2056], involved in DNA repair[12]) and ATM (pATM[S1981]) but not ATR (pATR[T1989]) (Fig. 1a,b). CHK2, the direct downstream target of ATM, was not detectably phosphorylated on its activating site (T68); similarly, CHK1, a target of ATR, was not phosphorylated on S317. Also P53 was not significantly phosphorylated on S15, an ATM/ATR target site (Fig. 1a,b). Differently, KAP1 (also known as TRIM28), a chromatin-bound ATM target[12], was strongly phosphorylated (pKAP1[S824]) together with phosphorylated H2AX (γH2AX) and RPA (pRPA[S4/8]), markers of DSB and SSB, respectively[11] (Fig. 1a,b). Similar results were generated in infected human lung epithelial Calu-3 cells[29,30] (Extended Data Fig. 1a,b).

To confirm and extend at single-cell resolution the impact of SARS-CoV-2 infection on DDR, we performed quantitative immunofluorescence analyses of the conditions aforementioned. We observed increased numbers of pKAP1[S824], pRPA[S4/8] and γH2AX foci per cell in infected Huh7 compared with mock-infected cells (Fig. 1c,d). In addition, SARS-CoV-2 infection of human nasal epithelial primary cells (HNEpCs) confirmed DDR activation, as detected by pRPA[S4/8] and γH2AX foci (Extended Data Fig. 1c,d).

To directly monitor the impact of the virus on physical DNA integrity, we performed comet assays in Huh7 and Calu-3 cells. We observed DNA fragmentation induction in both SARS-CoV-2-infected cell lines compared with control conditions, as measured by tail moment (Fig. 1e,f and Extended Data Fig. 1e,f).

Damaged DNA released in the cytoplasm can be sensed by the cGAS–STING pathway triggering an inflammatory response[31]. We therefore investigated cGAS–STING and other inflammatory pathways in cells upon SARS-CoV-2 infection and observed a higher number of micronuclei, which also stained positive for cGAS (Extended Data Fig. 1g,h) in Calu-3 cells, suggestive of the release of damaged nuclear DNA in the cytosol. In infected Huh7 cells, which do not express cGAS and STING[32], P38 and STAT1, factors involved in the pro-inflammatory response[7], were activated (Extended Data Fig. 1i).

To test the consequences of the activation of these pro-inflammatory pathways, we monitored by quantitative reverse transcription polymerase chain reaction (RT–qPCR) the transcriptional induction of *IL6*, *IL8*, *CXCL9*, *CXCL10* and *TNFα* genes in Huh7 and Calu-3 upon SARS-CoV-2 infection. We detected their significant upregulation in both cell types, although generally stronger in Calu-3 (Extended Data Fig. 1j). Since increased expression of pro-inflammatory genes is consistent with the induction of cellular senescence by SARS-CoV-2, as recently reported[33,34], we tested and confirmed the establishment of cellular senescence following SARS-CoV-2 infection in our settings, as demonstrated by increased senescence-associated β-galactosidase (SA-β-gal) activity (Extended Data Fig. 1l,m), augmented P21- and reduced KI67-positive cells (Extended Data Fig. 1n,o), and no significant induction of apoptosis (Extended Data Fig. 1k).

In sum, our results obtained by different techniques and in three independent cell types indicate that SARS-CoV-2 infection causes DNA damage and an altered DDR; this is associated with the induction of pro-inflammatory pathways and cytokines and cellular senescence.

### SARS-CoV-2 causes dNTP shortage by decreasing CHK1 levels

While studying activation of individual DDR proteins, we noticed that total CHK1 protein levels progressively decreased in infected Huh7 and Calu-3 cells (Figs. 1a and 2a–d and Extended Data Fig. 2c,d), mainly post-transcriptionally (Extended Data Fig. 2e,f). CHK1 loss is reportedly sufficient to cause DNA replication stress and DNA damage accumulation[35]. CHK1 controls the expression of the ribonucleoside-diphosphate reductase subunit M2 (RRM2), the small subunit of the RNR enzyme that converts ribonucleoside triphosphates (rNTPs) into dNTPs, necessary for DNA synthesis[23,36]. By testing RRM2 messenger RNA and protein levels by RT–qPCR, immunoblotting and immunofluorescence, we consistently observed their progressive and significant decrease following SARS-CoV-2 infection (Fig. 2a–d and Extended Data Fig. 2a–f).

Then, we measured individual dNTP concentrations in SARS-CoV-2 infected Huh7 and Calu-3 cells and observed reduced levels of cellular dNTPs compared with mock-infected conditions (Fig. 2e).

dNTP shortage can impair DNA synthesis, ultimately hampering S-phase progression[23,27]. To monitor cell-cycle progression, we measured DNA content in infected or mock-infected cells by propidium iodide (PI) staining followed by flow cytometry analysis. We observed a significant accumulation of infected cells in S-phase compared with control samples (Fig. 2f and Extended Data Figs. 3d and 8c). This was confirmed by strongly reduced levels of CDT1, a G1-phase marker[37–39] (Extended Data Fig. 2g). By pulse labelling with 5-bromo-2'-deoxyuridine (BrdU) for 1 h before flow cytometry, we

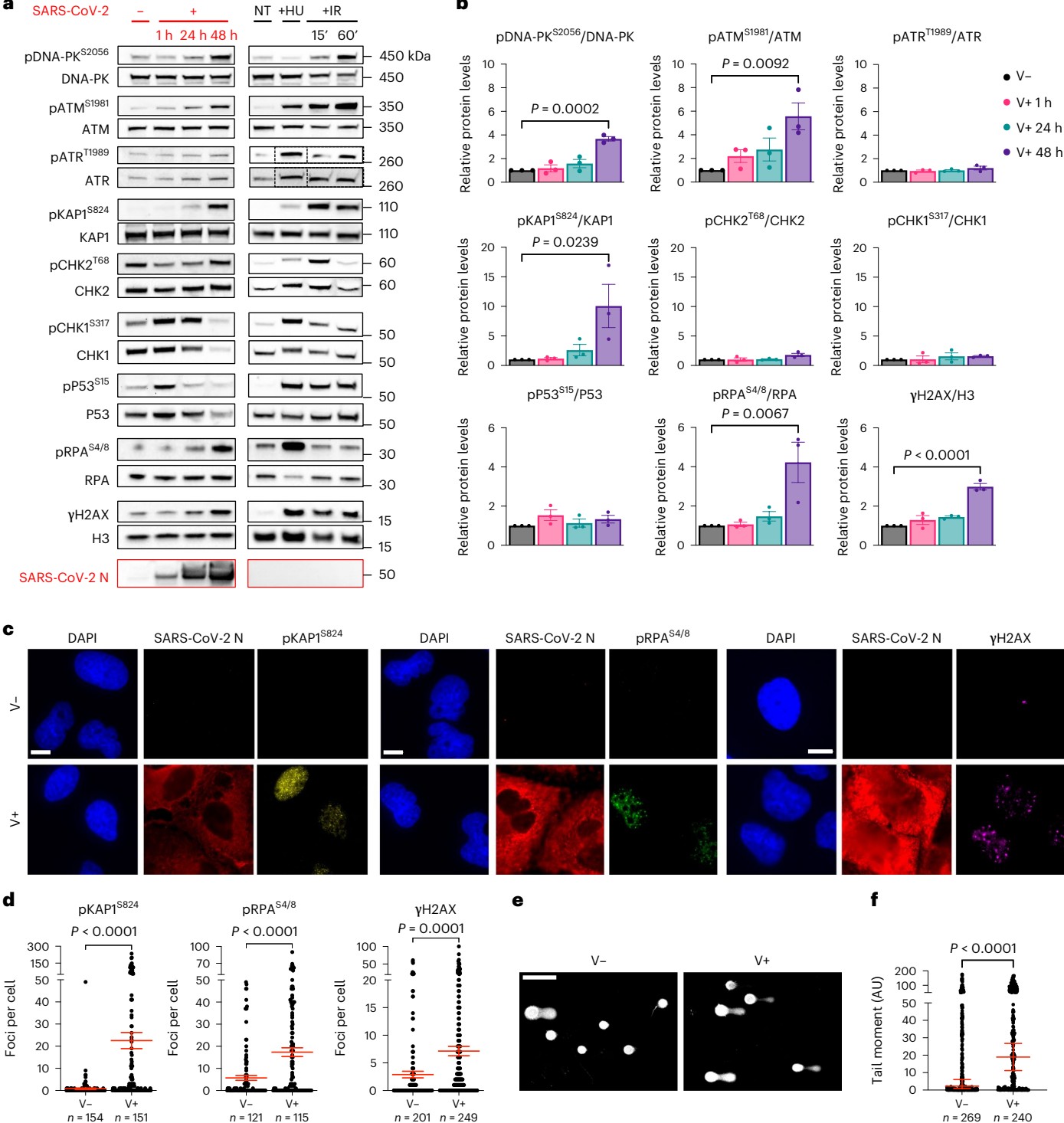

**Fig. 1 | SARS-CoV-2 infection causes DNA damage and altered DDR activation.**
**a**, Immunoblotting of whole cell lysates of Huh7 cells infected, or not, with SARS-CoV-2 analysed at different timepoints post-infection for markers of DDR activation. Lysates from Huh7 cells not treated (NT) or treated with 6 mM HU or exposed to 2 Gy IR and collected at different timepoints were used as positive controls. Viral infection was monitored by probing for SARS-CoV-2 N-protein. Where present, dashed lines indicate where the blot was cropped. **b**, Quantification of activated protein levels shown in **a**. Values are normalized to mock-infected samples. **c**, Representative immunofluorescence (IF) images of SARS-CoV-2-infected (V+) or mock-infected (V−) Huh7 cells fixed at 48 h post-infection and stained for DDR markers. SARS-CoV-2 N-protein was used to label infected cells. Nuclei were stained with DAPI. Scale bar, 10 μm. **d**, Quantification of DDR activation shown in **c**. Each dot is a nucleus. **e**, Images of comet assays of infected or mock-infected Huh7 cells. Scale bar, 100 μm. **f**, Quantification of comet tail moment shown in **e**. Horizontal bars represent the median values ± 95% confidence interval (CI) of three independent infections. Source numerical data and unprocessed blots are available in source data.

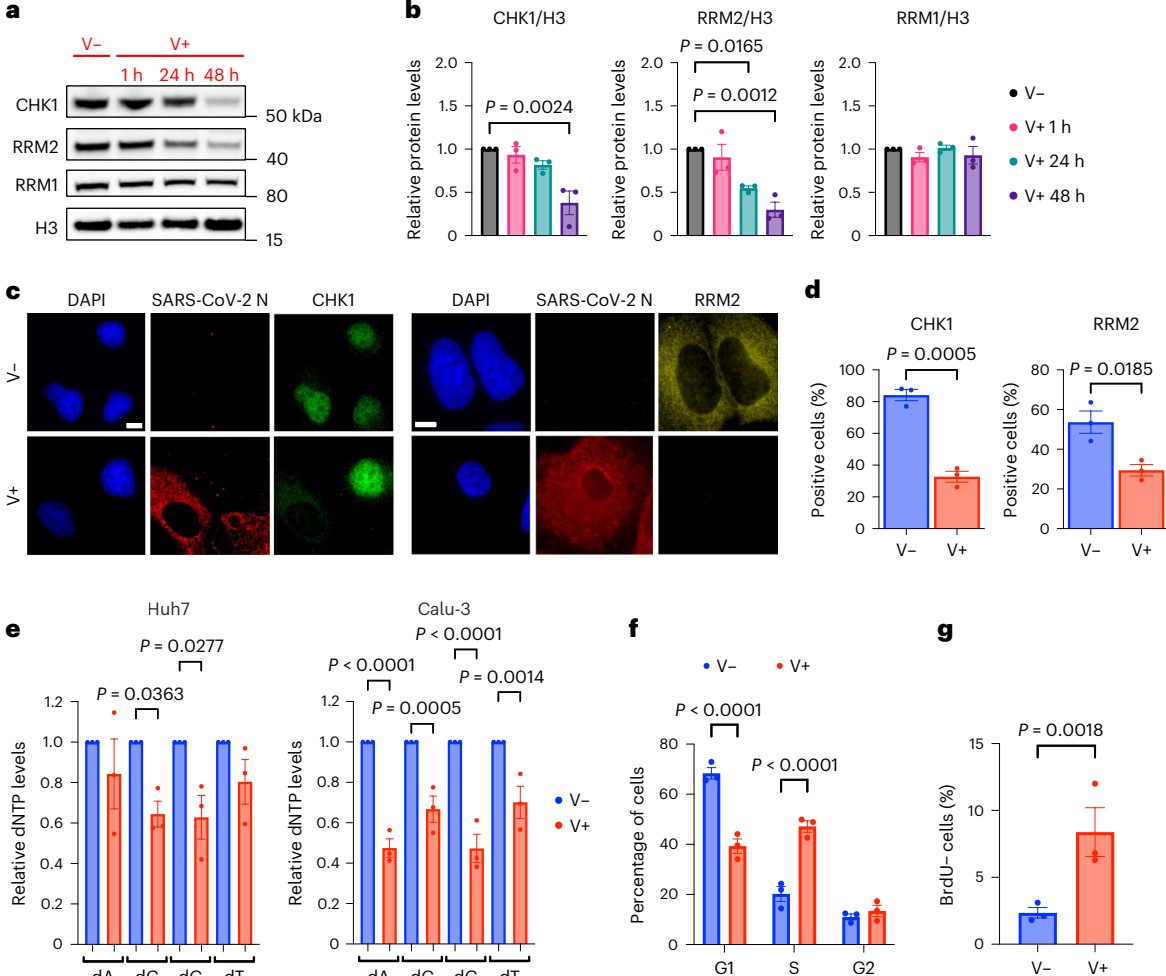

**Fig. 2 | SARS-CoV-2 reduces CHK1 and RRM2 levels leading to dNTP shortage.** **a**, Immunoblotting of whole cell lysates of Huh7 infected, or not, with SARS-CoV-2 and analysed at different timepoints post-infection. **b**, Quantification of protein levels shown in **a**; values are shown as relative to mock-infected samples. **c**, Immunofluorescence (IF) images of infected (V+) or mock-infected (V−) Huh7 cells fixed 48 h post-infection; nuclei were stained with DAPI. Scale bar, 10 μm. **d**, Quantification of CHK1- or RRM2-positive cells shown in **c**; *n* = 3 independent experiments. **e**, dNTP concentration was measured in V− or V+ Huh7 and Calu-3; values are shown as relative to V−. **f**, Histograms show the percentage of cells in each phase of the cell cycle in V− or V+ Huh7 fixed 48 h post-infection. **g**, Fraction of V− or V+ Huh7 cells that did not incorporate BrdU (BrdU−) measured by flow cytometry 48 h post-infection. Source numerical data and unprocessed blots are available in source data.

observed an increased percentage of BrdU-negative cells in S-phase in infected samples (Fig. 2g and Extended Data Figs. 3d and 8c). Altogether, these results indicate reduced dNTP levels and impaired S-phase progression following infection.

To determine the causal role of reduced dNTP levels, we tested the impact of dN supplementation to culture medium of infected cells. We observed that dN supplementation was sufficient to reduce DDR activation as shown by immunofluorescence and immunoblots of γH2AX, pRPA[S4/8] and pKAP1[S824] (Fig. 3a–d), DNA damage accumulation detected by comet assays (Fig. 3e,f) and transcription of several pro-inflammatory cytokines (Fig. 3g).

Overall, these results are consistent with a model in which SARS-CoV-2 gene products cause CHK1 loss, which reduces RRM2 levels and consequently the pool of available dNTPs, causing impaired DNA replication and S-phase progression, DNA damage accumulation and ultimately fuelling an inflammatory response. Supplementation of dNs is sufficient to tame these events.

## CHK1 loss is sufficient to cause DNA damage and inflammation

To determine whether CHK1 loss is sufficient to recapitulate the events here described following SARS-CoV-2 infection, we studied the impact of CHK1 depletion by RNA interference. Consistent with previous reports[23], we observed by flow cytometry that cells knocked down for CHK1 accumulate in S-phase (Extended Data Figs. 3a–d and 8c) and pulse labelling with BrdU for 1 h before flow cytometry analysis revealed a higher fraction of BrdU-negative S-phase cells compared with control samples (Extended Data Figs. 3d and 8c).

We also observed that CHK1 depletion was sufficient to reduce RRM2 levels and cause DNA damage, as shown by increased pRPA[S4/8] and γH2AX signals (Extended Data Fig. 3e,f). In addition, CHK1 knockdown led to the activation of P38 and STAT1 (Extended Data Fig. 3e,f) and formation of γH2AX foci and micronuclei, often positive for cGAS (Extended Data Fig. 3g–j), indicating that CHK1 loss in infected cells probably contributes to the activation of pro-inflammatory pathways. Indeed, cells depleted for CHK1 displayed increased expression of most of the cytokine and chemokine genes tested (Extended Data Fig. 3k,l) and increased secretion of IL6, CXCL9 and CXCL10 in Calu-3 cells as monitored by immunoassays (Extended Data Fig. 3m).

In sum, CHK1 loss is sufficient to recapitulate several of the events observed in SARS-CoV-2 infected cells, namely, RRM2 reduction, S-phase progression impairment, DNA damage and secretion of inflammatory cytokines.

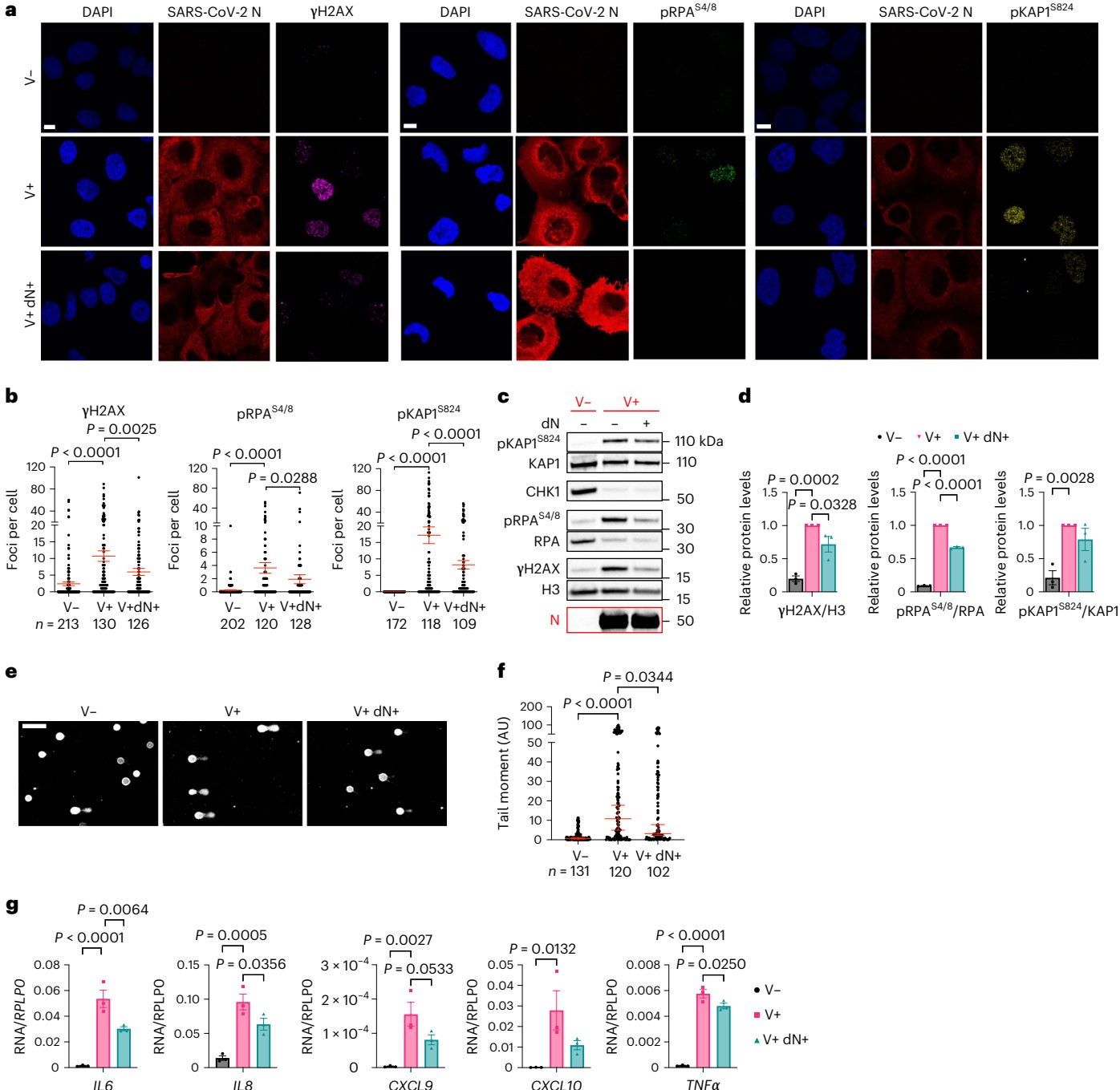

**Fig. 3 | dN supplementation is sufficient to reduce DNA damage and inflammation. a**, Immunofluorescence (IF) images of V− or V+ Huh7 cells, treated or not with dNs, fixed 48 h post-infection; nuclei were stained with DAPI. Scale bar, 10 μm. **b**, Quantification of DDR activation shown in **a**. Each dot is a nucleus. **c**, Immunoblots of Huh7 cells treated as in **a**. **d**, Quantification of protein levels shown in **c**. Values are normalized to untreated V+ cells. **e**, Images of comet assays of Huh7; conditions are as in **a**. Scale bar, 100 μm. **f**, Quantification of comet tail moment shown in **e**; horizontal bars represent the median values ± 95% CI of three independent infections. **g**, RT–qPCR of pro-inflammatory cytokine expression in V− or V+ Calu-3 cells, treated or not with dNs. Values are shown as relative to *RPLP0* mRNA. Source numerical data and unprocessed blots are available in source data.

## SARS-CoV-2 ORF6 and NSP13 trigger CHK1 protein degradation

To identify the viral gene products responsible for CHK1 downregulation, we individually expressed 24 of the 26 annotated SARS-CoV-2 proteins[40] (SARS-CoV-2 reference genome, NC_045512.2) and analysed by immunoblotting their impact on CHK1 levels. Among the gene products tested, ORF6 and NSP13 were the factors with the strongest and most consistent impact on CHK1 protein levels (Fig. 4 and

Extended Data Fig. 4a,b). Their sole expression was also sufficient to reduce RRM2 levels and increase γH2AX and RPA phosphorylation (S4/8) (Fig. 4a–d and Extended Data Fig. 4a,b).

## ORF6 causes CHK1 degradation through the proteasome pathway

SARS-CoV-2 ORF6 has been shown to associate with the nuclear pore and to interfere with proteins' nuclear–cytoplasmic trafficking[41,42].

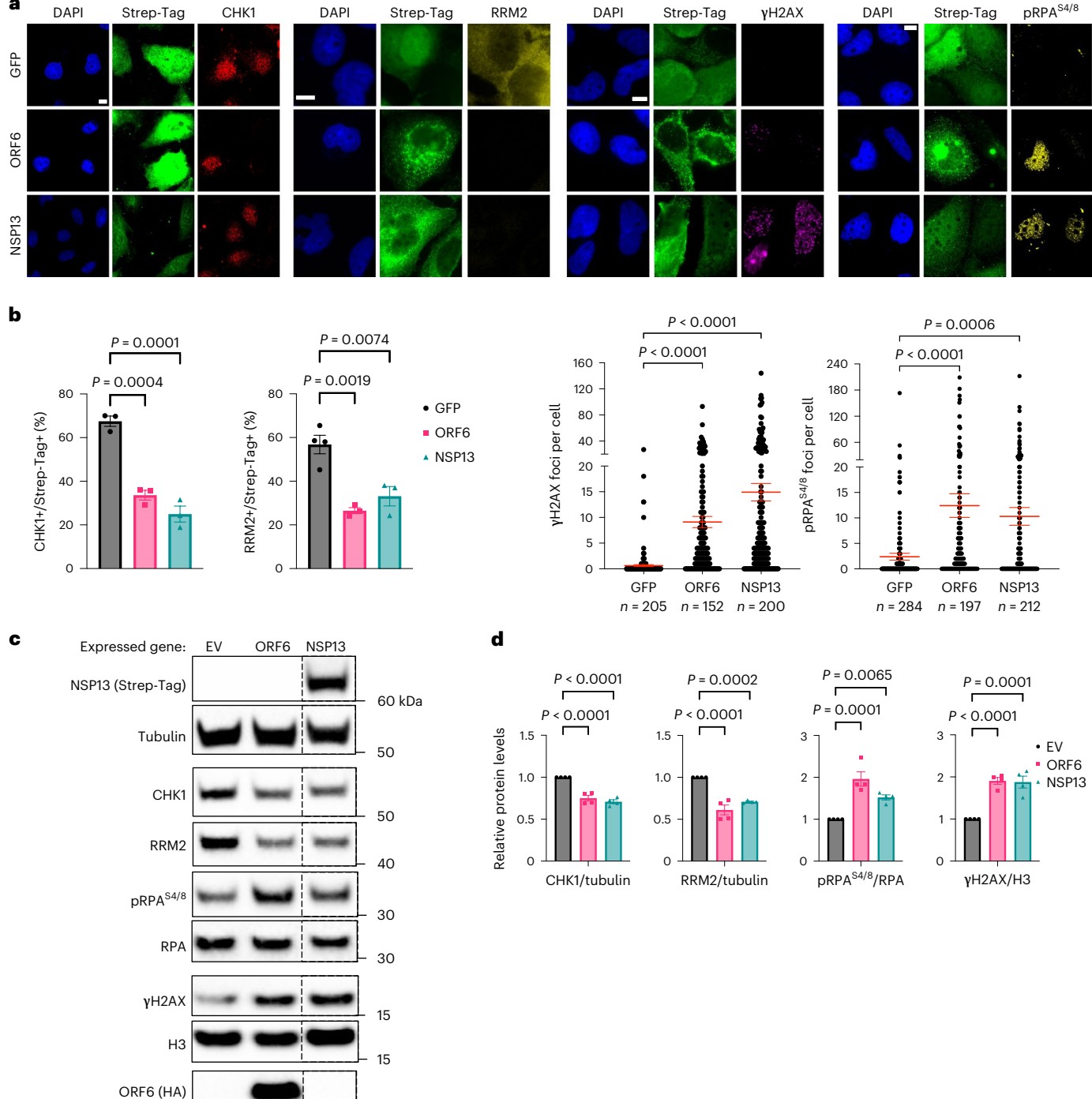

**Fig. 4 | SARS-CoV-2 ORF6 or NSP13 expression is sufficient to cause CHK1 loss. a**, Immunofluorescence (IF) images of Huh7 cells expressing Strep-tagged SARS-CoV-2 ORF6 or NSP13 fixed 48 h post-transfection and stained for DDR markers; GFP was used as control; staining with anti-Strep-tag was used to label transfected cells; nuclei were stained with DAPI. Scale bar, 10 μm. **b**, The histograms show the percentage of CHK1- or RRM2-expressing cells among the transfected ones (Strep-Tag+) as determined in **a**; $n = 3$ independent experiments ($n = 4$ for GFP-expressing cells in RRM2 analysis). The dot plots show the number of γH2AX or pRPA$^{S4/8}$ foci in the samples described in **a**. **c**, Representative immunoblots of whole cell lysates from Huh7 cells transfected with plasmids encoding for HA-tagged SARS-CoV-2 ORF6, or Strep-tagged SARS-CoV-2 NSP13, or empty vector (EV) as a control. Where present, dashed lines indicate where the blot was cropped. **d**, Quantification of protein levels shown in **c**; values are the mean ± s.e.m. of four independent experiments and shown as relative to the control sample (EV). Source numerical data and unprocessed blots are available in source data.

We observed that SARS-CoV-2-infected cells displayed cytoplasmic CHK1 localization compared with mock-infected cells in which CHK1 is mainly nuclear (Fig. 2c). It has been reported that accumulation of CHK1 in the cytoplasm leads to its degradation through the UPS[43].

To investigate the potential involvement of ORF6 in the cytoplasmic accumulation of CHK1, we took advantage of a point mutant form of ORF6 (ORF6$^{M58R}$) unable to interact with the nuclear pore complex[41]. Interestingly, CHK1 protein levels did not decrease in Huh7

cells expressing the mutant ORF6, as detected both by immunofluorescence and immunoblots (Fig. 5a–d). Consistent with that, mutant ORF6 expression also had no impact on RRM2 protein levels and DNA damage accumulation as detected by pRPA$^{S4/8}$ and γH2AX (Fig. 5c,d). This suggests that ORF6, by altering CHK1 nuclear–cytoplasm shuttling, may cause its degradation.

We therefore probed the engagement of UPS in ORF6-dependent CHK1 loss. Treatment of ORF6-expressing Huh7 cells with MG132—a proteasome inhibitor—recovered CHK1 protein levels, detected mostly in the cytoplasm (Fig. 5e–h), consistent with impaired protein trafficking of ORF6-expressing cells.

To demonstrate that SARS-CoV-2 ORF6 expression causes CHK1 poly-ubiquitination, a prerequisite of proteosome-dependent degradation, we immunoprecipitated endogenous CHK1 in Huh7 cells expressing either ORF6 or ORF6$^{M58R}$, treated or not with MG132, and probed for ubiquitinated CHK1. Proteasomal inhibition led to a higher accumulation of ubiquitinated CHK1 in ORF6-expressing cells compared with samples that overexpressed ORF6$^{M58R}$ (Fig. 5i,j).

These results indicate that SARS-CoV-2 ORF6 prevents CHK1 nuclear import, causing its accumulation in the cytoplasm and its consequent degradation through the UPS.

### NSP13 causes CHK1 degradation through autophagy
It has been shown that NSP13 can promote protein degradation in an autophagy-dependent manner[44]. Therefore, to test whether CHK1 loss in NSP13-expressing cells was dependent on the autophagic route, we transiently expressed the viral NSP13 gene in Huh7 cells in the presence of either Bafilomycin A1 (BafA1) or chloroquine (CQ), two specific inhibitors of autophagy[45]; efficacy was confirmed by the observed accumulation of P62 cytoplasmic aggregates[45] (Extended Data Fig. 5a,b). NSP13-mediated reduction of CHK1 protein levels was abolished by both treatments (Fig. 5k,l). A short BafA1 exposure highlighted a clear co-localization of CHK1 with P62 cytoplasmic aggregates, suggesting its accumulation in autophagosomes (Fig. 5m,n). To confirm and extend these results, we individually knocked down Beclin 1 (BECN1) and LC3B—two key regulators of autophagy[46]—and observed a significant restoration of CHK1 protein levels in NSP13-expressing cells (Fig. 5o,p and Extended Data Fig. 5c,d).

These results indicate that SARS-CoV-2 NSP13 causes the accumulation of CHK1 in the cytoplasm, where it co-localizes with P62, in this way promoting its degradation through autophagy.

### N-protein impairs 53BP1 recruitment at DSB and hinders NHEJ
We noticed that γH2AX foci accumulation was not accompanied by co-localizing 53BP1 foci in SARS-CoV-2-infected Huh7, Calu-3 and HNEpC (Fig. 6a–d and Extended Data Fig. 6a,b), despite unaltered 53BP1 protein levels (Extended Data Fig. 6c).

SARS-CoV-2 N-protein is an RNA-binding protein capable to undergo RNA-dependent–LLPS[21,22,47–49]. We previously reported that also 53BP1 phase-separates in an RNA-dependent manner[17]. To test the potential impact of N-protein on 53BP1 foci formation, we expressed the viral N gene[40] in Huh7 cells and exposed them to IR. We observed that irradiated cells expressing N showed increased numbers of γH2AX foci per cell, but fewer 53BP1 foci compared with control cells (Fig. 6e,f). To reduce the possibility of an indirect effect mediated by altered gene expression, we micro-injected purified recombinant N-protein into the nuclei of irradiated cells stably expressing 53BP1-GFP[50] and immediately studied the kinetics of 53BP1 foci by live imaging. We observed 53BP1 foci number decreasing with a faster (~8.5-fold) kinetic in cells injected with the N-protein compared with control cells (Fig. 6g, Extended Data Fig. 6d and Supplementary Video 1).

Next, we sought to elucidate the molecular mechanisms underlying N-protein impact on 53BP1 functions. In co-immunoprecipitation experiments, 53BP1 did not interact with N-protein (Extended Data Fig. 6e). We previously reported that dilncRNAs generated at DSB drive LLPS of 53BP1 (refs. [16,17]). Intriguingly, both viral and cellular RNAs have been reported to associate with N-protein and promote its phase separation[48], as we confirmed (Extended Data Fig. 6f,g).

Since N-protein, although mainly cytoplasmic, also localizes in the nucleus[51–53] (Extended Data Fig. 6j,k), we tested whether N associates with cellular dilncRNAs by performing RNA immunoprecipitation (RIP) against the N-protein in NIH2/4 cells, which we previously characterized for the expression of dilncRNAs upon DSB induction by I-SceI endonuclease[16]. Therefore, following SARS-CoV-2 N gene expression into NIH2/4 cells and DSB induction by I-SceI, we immunoprecipitated N-protein and analysed the associated RNAs by RT–qPCR. We observed that N-protein was associated with dilncRNA upon DSB generation, but not with H2AX mRNA used as a negative control (Fig. 6h). Next, we immunoprecipitated endogenous 53BP1 in I-SceI-induced NIH2/4 cells expressing or not the viral N-protein, and monitored dilncRNA association with 53BP1. We observed that 53BP1 association with dilncRNAs was reduced in cells expressing N-protein (Fig. 6i), despite unaltered 53BP1 protein levels or immunoprecipitation (IP) efficiency following N-protein overexpression (Extended Data Fig. 6h).

Since 53BP1 plays important DNA repair functions through non-homologous end-joining (NHEJ)[54], we tested the impact of N-protein on NHEJ. We took advantage of a cell line bearing an integrated GFP construct flanked by two I-SceI recognition sites (EJ5-GFP)[55]: following I-SceI expression, DSBs are generated and repair can be quantified by qPCR on genomic DNA (gDNA) with primers flanking the re-joined site[56]. EJ5-GFP U2OS were transfected with a plasmid expressing I-SceI together with N-protein or an EV. Seventy-two hours post-transfection, gDNA was collected and analysed. NHEJ efficiency in cells expressing N-protein was significantly decreased compared with control samples (Fig. 6j) to an extent comparable to that previously observed upon 53BP1 depletion[17,56,57] while leaving I-SceI levels unchanged (Extended Data Fig. 6i).

In sum, our evidence indicates that SARS-CoV-2 N competes with 53BP1 for dilncRNAs binding and thus reduces 53BP1 focus formation at DSB, ultimately hampering DNA repair by NHEJ.

**Fig. 5 | ORF6 and NSP13 causes CHK1 reduction through the proteasome and autophagy pathways, respectively. a**, Images of Huh7 expressing GFP (negative control), ORF6 or its mutant form ORF6$^{M58R}$. **b**, Quantification of the percentage of transfected cells expressing CHK1 shown in **a**. **c**, Immunoblotting of Huh7 treated as in **a**; EV-transfected cells were used as negative control. **d**, Quantification of the protein levels shown in **c**. Values are the mean ± s.e.m. of four independent experiments. **e**, Confocal images of GFP- or ORF6-expressing Huh7 ± MG132. **f**, Quantification of nuclear (n) and cytoplasmic (c) CHK1 levels in the cells described in **e**. **g**, Immunoblots of the samples described in **e**; EV-transfected cells were used as negative control. **h**, Quantification of the protein levels shown in **g**. **i**, Ubiquitination assay of CHK1 immunoprecipitated from ORF6- or ORF6$^{M58R}$-expressing Huh7 ± MG132. **j**, Quantification of the samples shown in **i**. Values are shown as relative to immunoprecipitated CHK1 amounts (IP-CHK1); n = 3 independent experiments. **k**, Immunofluorescence (IF) images of GFP- or NSP13-expressing Huh7 ± BafA1 or CQ. **l**, CHK1 quantification in cells described in **k**. Values are the mean ± s.e.m. of four independent experiments, except for GFP and NSP13 not treated (NT) conditions (n = 6). **m**, IF images of NSP13-expressing Huh7 treated with BafA1 (1 h). Arrow points to CHK1 and P62 co-localization. **n**, Quantification of the percentage of cells displaying co-localizing CHK1 and P62 signals shown in **m**. **o**, IF images of CHK1 levels in Huh7 transfected with the indicated siRNAs before viral NSP13 overexpression. **p**, Quantification of the percentage of CHK1-expressing cells in the transfected samples represented in **o**. Values are the mean ± s.e.m. of four independent experiments. Scale bar, 10 μm and DAPI-stained nuclei in **a,e,k,m** and **o**. Source numerical data and unprocessed blots are available in source data.

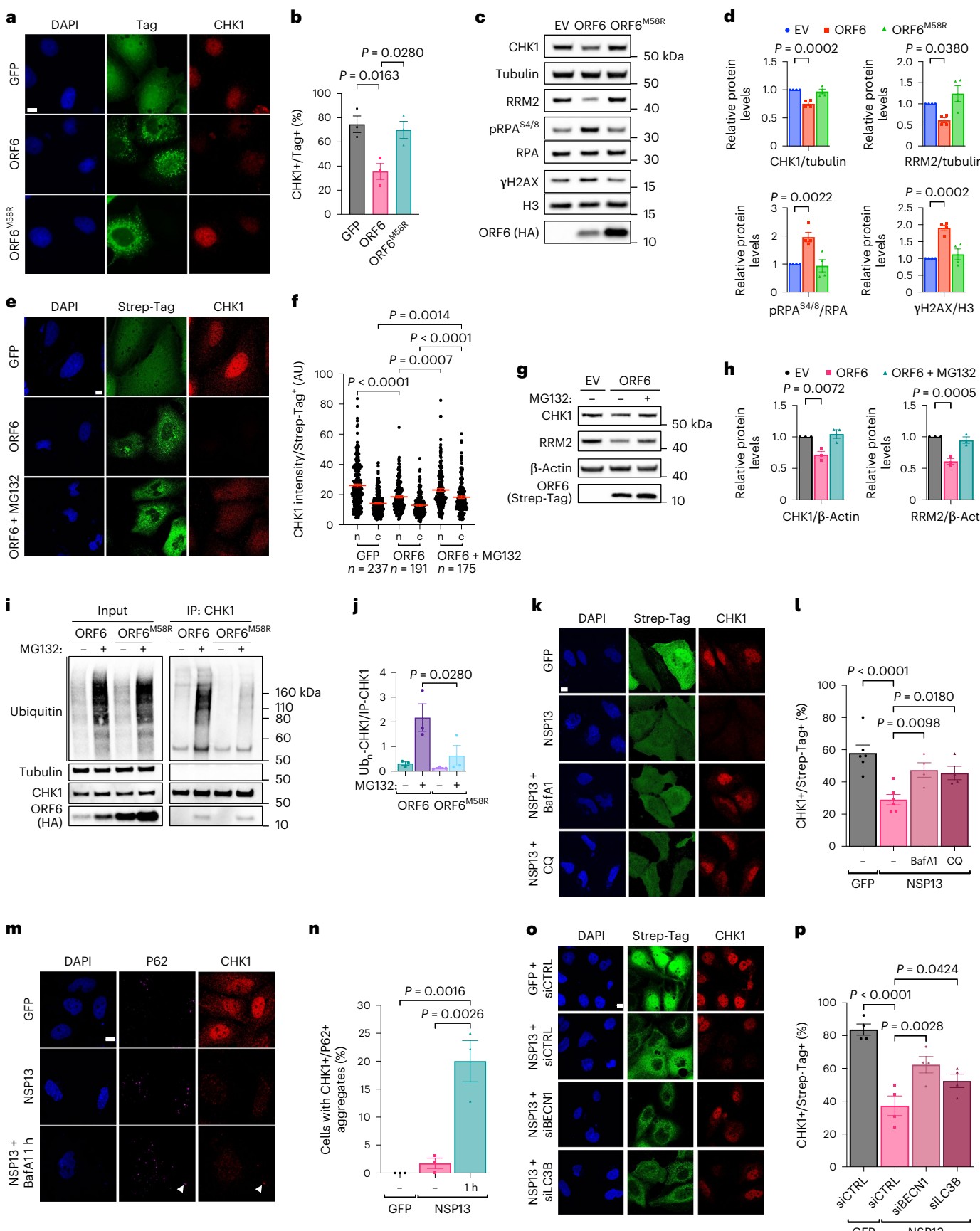

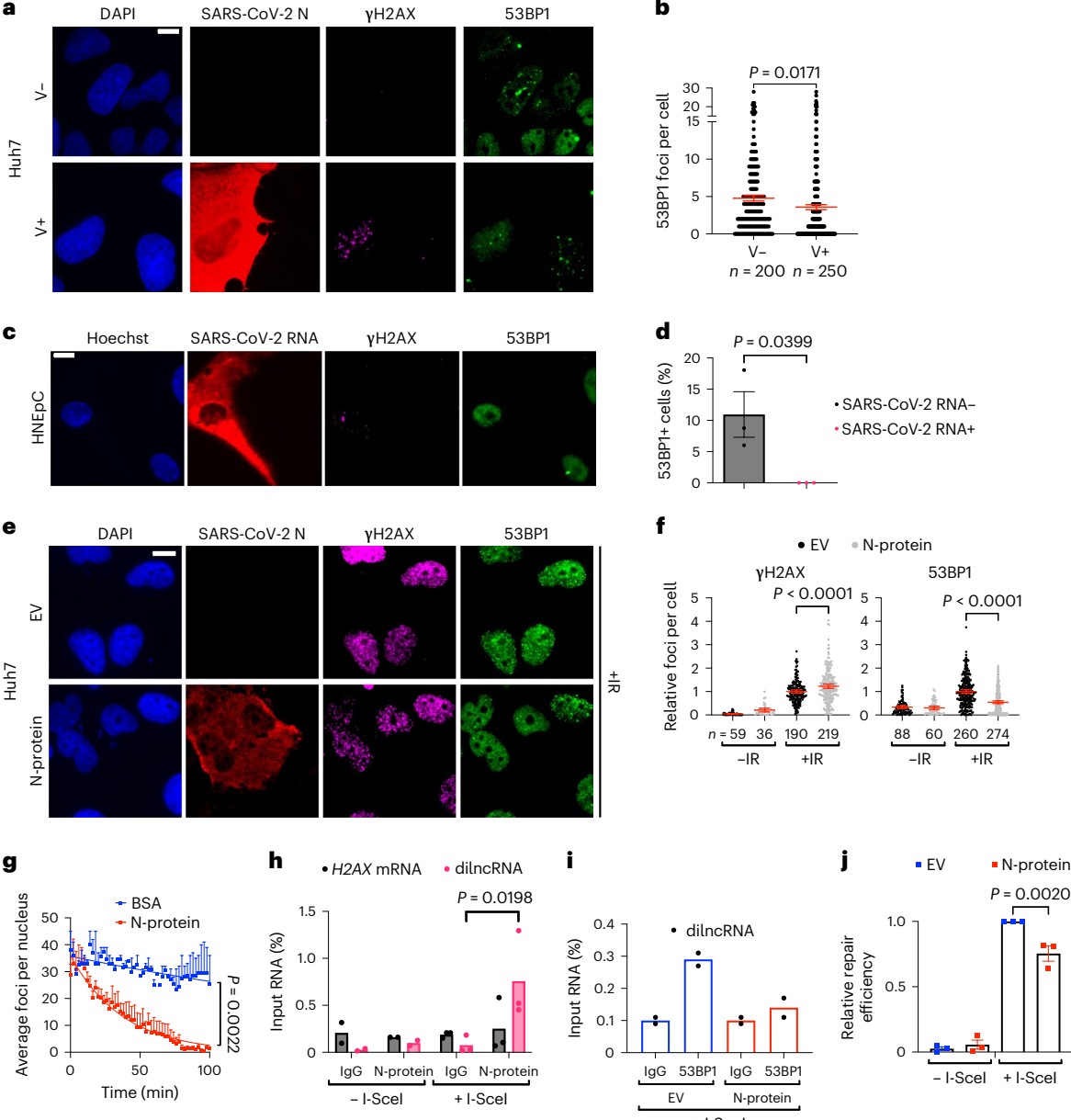

**Fig. 6 | SARS-CoV-2 N suppresses 53BP1 activation and inhibits repair by NHEJ.**
**a**, Immunofluorescence (IF) images of V+ or V− Huh7; nuclei were stained with DAPI. **b**, Quantification of 53BP1 foci shown in **a**. Each dot represents the number of 53BP1 foci per nucleus. **c**, IF images of infected HNEpC in which SARS-CoV-2 RNA was detected by FISH; nuclei were stained with Hoechst. **d**, Quantification of 53BP1 foci shown in **c**; the histograms show the percentage of nuclei with 53BP1 foci (>1) in cells expressing (+) or not (−) SARS-CoV-2 RNA; n = 3 independent infections. **e**, IF images of irradiated Huh7 transfected with N-protein or EV as control; nuclei were stained with DAPI. **f**, Quantification of DDR foci shown in **e**; the dot plots show the number of γH2AX and 53BP1 foci per nucleus in N-protein- or EV-expressing samples. Values are relative to irradiated cells transfected with EV; bars represent the mean ± 95% CI of three independent experiments. **g**, Quantification of 53BP1 foci per nucleus over time in irradiated cells injected with recombinant N-protein or BSA as control. Error bars represent

s.e.m.; the experiment was repeated three times with similar results. **h**, NIH2/4 expressing (n = 3) or not (n = 2) I-SceI were transfected with N-protein. Cell lysates were incubated with anti-N-protein or normal rabbit IgG and co-precipitated RNA analysed by strand-specific RT–qPCR. *H2AX* mRNA was used as an unrelated transcript. Values are shown as percentage of input RNA. **i**, Endogenous 53BP1 was immunoprecipitated from I-SceI-expressing NIH2/4 transfected with N-protein or EV as control. 53BP1-bound transcripts were monitored as in **h** and shown as percentage of input RNA. Values are the average of two independent experiments. **j**, EJ5-GFP U2OS were transfected with N-protein or EV, ± I-SceI. DSB re-joining events were evaluated by qPCR on gDNA isolated at 72 h post-transfection. Values are relative to I-SceI-transfected cells not expressing N-protein. Scale bar, 10 μm (**a**, **c** and **e**). Source numerical data are available in source data.

## SARS-CoV-2 causes DNA damage in mice and patients with COVID-19

We next extended our analyses in in vivo settings of SARS-CoV-2 infection. Lung sections of mice expressing human ACE2 (hACE2) were stained for DDR markers following intranasal administration of

SARS-CoV-2 (Fig. 7a,b and Extended Data Fig. 7a,b); mice exposed to IR were used as positive control (Extended Data Fig. 7c,d). Immunostaining of γH2AX and pRPA^S4/8 together with N-protein demonstrated strong γH2AX and pRPA^S4/8 signal induction in infected samples compared with mock-infected ones. Instead, 53BP1, CHK1 and RRM2 signals were

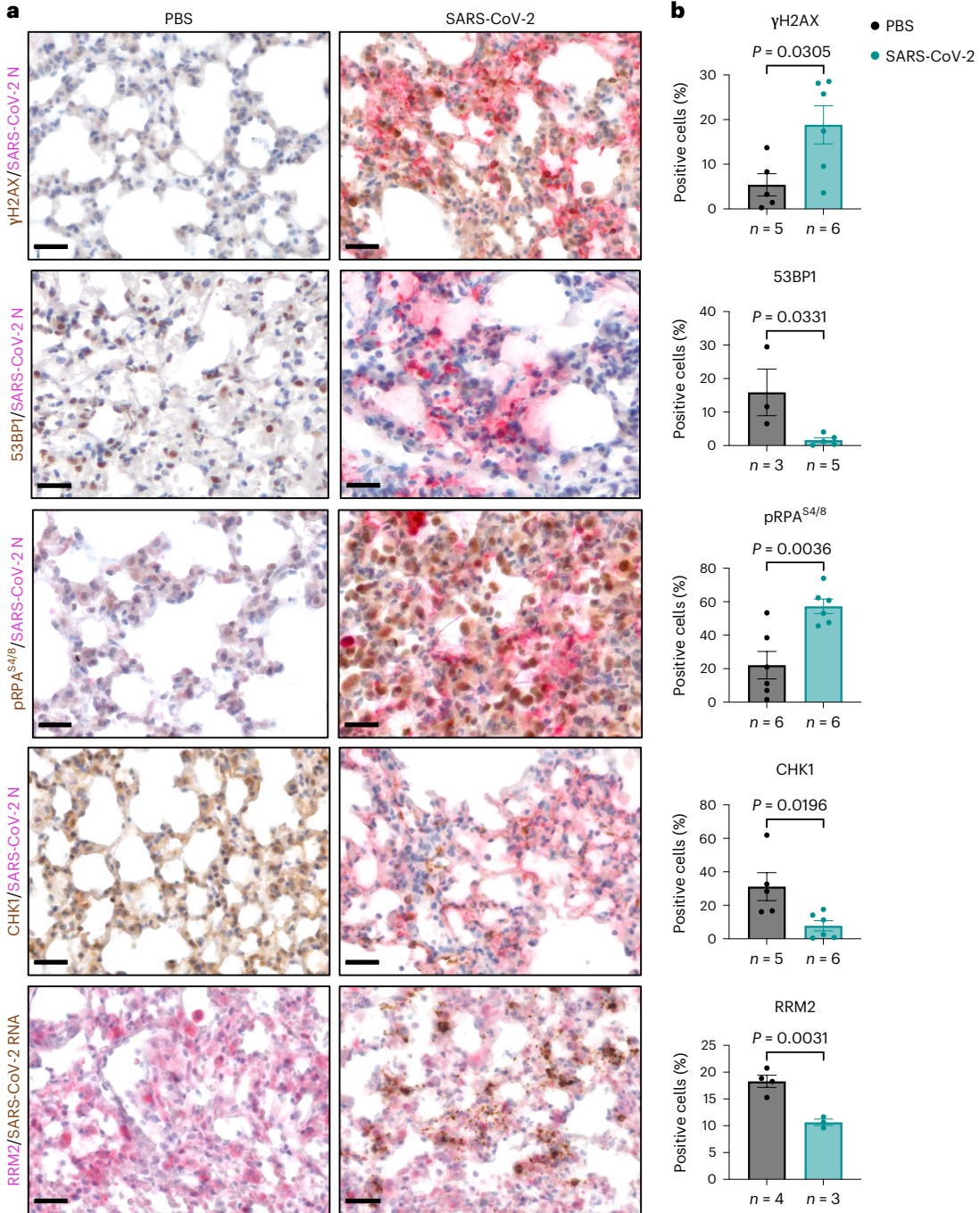

**Fig. 7 | SARS-CoV-2 infection causes DNA damage in *hACE2*-mouse lungs.**
**a**, IHC images of lungs from SARS-CoV-2-infected *hACE2*-mice or mock-infected (PBS) wild-type mice that were stained for the indicated markers at 6 days following intranasal administration; the presence of the virus was assessed by staining for SARS-CoV-2 N or by probing the viral genome through RNA ISH; nuclei were stained with haematoxylin (light blue). Scale bar, 50 μm.
**b**, Quantification of the percentage of cells positive for the indicated markers as shown in **a**. Values are the mean ± s.e.m.; at least three mice were studied for each condition; the precise number is indicated below each histogram. Source numerical data are available in source data.

reduced in infected murine lungs (Fig. 7a,b). Thus, consistent with our observations in cultured cells, SARS-CoV-2 infection in mice causes DNA damage accumulation, impairs 53BP1 activation and reduces CHK1 and RRM2 protein levels.

As SARS-CoV-2 infection has been recently shown to induce cellular senescence and contribute to inflammation in vivo[9,10,33,34], we probed the same tissues for p16 and p21 and observed an increase of both in infected lungs (Extended Data Fig. 7e,f), with p21 detected almost exclusively in pneumocytes and in bronchial epithelium of the infected

lungs, while p16 also associated with inflammatory cells populating the lung parenchyma of infected mice (Extended Data Fig. 7g).

We next analysed lungs and nasal mucosa sections from patients who died with a clinical diagnosis of COVID-19, determined by a positive naso-pharyngeal swab or bronchoalveolar lavage during their last hospitalization[58]. In situ hybridization (ISH) and immunohistochemistry (IHC) were used to confirm the presence of SARS-CoV-2 RNA or N-protein, respectively. As negative control, we studied the lungs of patients not diagnosed for the pathology (non-COVID) but affected

by viral pneumonia of different aetiologies[58]. Since for ethical reasons we were not able to collect nasal mucosa from non-COVID subjects, we compared samples scored for the presence or absence of SARS-CoV-2, as detected by fluorescence in situ hybridization (FISH).

We probed lung and nasal mucosa for γH2AX, SARS-CoV-2 genome and cytokeratin-18 (CK18), a marker of epithelial cells, and observed increase γH2AX foci in epithelial cells of infected specimens, compared with controls (Fig. 8a,c). Similarly, pRPA[S4/8] levels were strongly induced in the lungs of patients with COVID-19 (COVID) compared with non-COVID subjects (Fig. 8e,f).

We also analysed 53BP1 in both lung and nasal mucosa. In lung samples of patients with COVID, γH2AX foci were significantly less associated with 53BP1 than ex vivo bleomycin-treated sections of human non-COVID lungs (non-COVID bleo+) used as positive control, indicating impaired 53BP1 recruitment in infected lungs (Fig. 8b,d). Similarly, in nasal mucosa, COVID FISH+ cells displayed a reduced recruitment of 53BP1 compared with COVID FISH− cells (Fig. 8b,d). Both CHK1 and RRM2 expression was invariably and significantly lower in infected tissues compared with not infected ones (Fig. 8e–h).

Staining of lung tissues for P16 and P21 together with SARS-CoV-2 markers revealed that both were increased in patients with COVID compared with non-COVID individuals (Extended Data Fig. 8a,b).

Overall, these results indicate that SARS-CoV-2 infection in vivo causes an altered DDR activation that is associated with increased levels of DNA damage and cellular senescence in mouse and patients with COVID-19.

## Discussion

Viruses are known to hijack cellular activities, including DDR, as a strategy to promote their replication[6,59]. This may have deleterious effects on the cell, potentially leading to genome instability[59]. SARS-CoV-2 infection has been reported to alter different host pathways[7,60,61] and to correlate with the activation of some DDR markers and senescence[7–10,33,62–64]. The observation that some DDR inhibitors reduce SARS-CoV-2 replication[65] further hints at a mutual interplay. However, when studied, DNA damage was correlated mainly with reactive oxygen species[10,33].

Here, we showed that SARS-CoV-2 infection causes DNA damage (Fig. 1e,f and Extended Data Fig. 1e,f), as observed in two immortal cell lines, in primary human cells and in vivo in mice and humans. We identified at least two mechanisms responsible for DNA damage accumulation: one impacting on cellular dNTP metabolism leading to DNA replication impairment; another impeding 53BP1 activation and reducing DNA repair (Fig. 8i). The DNA damage accumulated triggers DDR activation, but in an altered way (Fig. 1a,d and Extended Data Fig. 1a,d). For instance, CHK1, together with P53, decreases following SARS-CoV-2 infection (Figs. 1a and 2a–d and Extended Data Fig. 2c,d). Degradation of DDR factors is a strategy shared by different viruses to override host defences[59,66–68].

CHK1 is known to control the expression of E2F transcription factors, important regulators of cell-cycle progression[69], and consequently of the RRM2 gene, allowing DNA synthesis in S-phase[23,36,70].

We demonstrated that SARS-CoV-2 infection leads to CHK1 loss and consequent RRM2 decrease (Figs. 1a and 2a–d), causing dNTP shortage and prolonged S-phase (Fig. 2e–g), consistent with the generation of DNA replication stress and DNA damage. This cascade of events leads to the establishment of cellular senescence and activation of pro-inflammatory pathways (Extended Data Figs. 1g–o, 7e–g and 8).

CHK1 depletion is sufficient to recapitulate RRM2 reduction, DNA damage accumulation and cytokine expression (Extended Data Fig. 3). Importantly, the administration of dNs to SARS-CoV-2-infected cells reduced virally induced DNA damage, DDR activation and cytokine expression (Fig. 3), thus demonstrating the causative role of dNTP depletion in these events.

We propose that this is probably the unmeant consequence of the dire need for rNTPs of SARS-CoV-2. Staggering two-thirds of total RNA in SARS-CoV-2-infected cells is of viral origin[71]: thus infected cells need to triple their normal RNA synthesis capacity. Therefore, the virus has been under evolutionary pressure to boost rNTP levels. One way is to reduce CHK1 levels, causing decreased RRM2 activity and consequent accumulation of rNTPs at the expense of dNTPs. Interestingly, a similar yet opposite mechanism was observed in the DNA virus HPV31, which boosts RRM2 to increase dNTPs and favour its genome replication[72].

At least two SARS-CoV-2 products cause CHK1 degradation. ORF6, by associating with the nuclear pore complex, interferes with CHK1 nuclear import, leading to CHK1 cytoplasmic mis-localization and consequent proteasomal degradation. Notably, a point mutation that disrupts ORF6 binding to the nuclear pore complex prevented CHK1 poly-ubiquitination, degradation and DNA damage accumulation (Figs. 5a–d,i,j). In addition, proteasome inhibition with MG132 in ORF6-expressing cells was sufficient to rescue CHK1 levels (Fig. 5e–h). Differently, NSP13 leads to CHK1 depletion through the autophagic route, as indicated by the recovery of CHK1 levels upon treatment with autophagy inhibitors or with RNAi against key autophagy factors (Fig. 5k–p).

In addition to induce DNA damage, SARS-CoV-2 inhibits its repair. We observed a strikingly reduced ability of 53BP1 to form DDR foci, despite unaltered protein levels, in infected cells (Fig. 6a–d and Extended Data Fig. 6a–c). We propose that SARS-CoV-2 N, an avid RNA-binding protein, impairs 53BP1 condensation at DSB by competing for dilncRNA binding. Indeed, both 53BP1 and N-protein undergo LLPS in an RNA-dependent manner, and we demonstrate that N-protein, just like 53BP1 (ref. [16]), binds to dilncRNA (Fig. 6g–i and Supplementary Video 1).

These data suggest a nuclear role of SARS-CoV-2 N-protein. Although both SARS-CoV and SARS-CoV-2 N-proteins bear functional nuclear localization signals, they are only partly nuclear[73–75] (for example, Extended Data Fig. 6j,k), but phylogenetic studies have correlated the enhancement of motifs that promote nuclear localization of viral N-proteins with coronavirus pathogenicity and virulence[75].

Intriguingly, enoxacin, a molecule that we reported to boost RNA-mediated 53BP1 foci assembly and DNA repair[56], has been

**Fig. 8 | SARS-CoV-2 causes CHK1 and RRM2 loss and DDR activation in lungs and nasal mucosa of patients with COVID-19. a**, Immunofluorescence (IF) images of lungs of patients diagnosed (COVID, $n = 17$) or not (non-COVID, $n = 9$) with COVID-19, and nasal mucosa of patients with COVID-19 in which cells were detected as positive (FISH+, $n = 18$) and negative (FISH−, $n = 11$) for SARS-CoV-2. Tissues were stained for SARS-CoV-2 RNA, γH2AX and CK18 to label epithelial cells; nuclei were counter-stained with Hoechst. Scale bar, 10 μm. **b**, IF images of lung ($n = 4$) and nasal mucosa ($n = 5$) of patients with COVID-19 stained for SARS-CoV-2 N-protein, γH2AX and 53BP1. Sections ($n = 6$) of a lung treated ex vivo with 2 μM bleomycin were used as positive control for DDR activation. Nuclei were stained with Hoechst. Scale bar, 10 μm. **c**, Quantification of DDR determined in **a**. Histograms show the percentage of epithelial cells bearing γH2AX foci (>1). **d**, Quantification of 53BP1 recruitment at γH2AX sites determined in **b**. The histograms show the percentage of nuclei with 53BP1 foci (>1) among those positive for γH2AX. **e**, IHC images of tissues of patients with COVID-19 stained for the indicated markers. Virus presence was assessed by probing for N-protein. Nuclei were stained with haematoxylin (light blue). Conditions are as in **a**; scale bar, 50 μm. **f**, The histograms show the percentage of cells positive for the markers determined in **e**. pRPA[S4/8]: non-COVID ($n = 5$ patients); COVID ($n = 10$). CHK1: non-COVID ($n = 7$); COVID ($n = 3$); FISH− ($n = 4$); FISH+ ($n = 4$). Values are the mean ± s.e.m. **g**, IF images of tissues of patients with COVID-19 stained for RRM2; scale bar, 10 μm. **h**, The histograms show the average intensity of RRM2 signal as determined in **g**. Values are the mean ± s.e.m.; $n = 3$ individuals for lungs and $n = 5$ for nasal mucosa. **i**, Schematic model of the impact of SARS-CoV-2 infection on genome integrity and cellular senescence. Source numerical data are available in source data.

predicted in silico to reduce SARS-CoV-2 replication[76]. It is tempting to speculate that enoxacin, by enhancing 53BP1 activities, counteracts SARS-CoV-2 N-protein competition with 53BP1 for dilncRNAs.

Hyperactivation of inflammatory pathways is responsible for fatal COVID-19 cases[77]. DNA damage accumulation and chronic DDR activation are potent inducers of inflammation[78]. Consistent

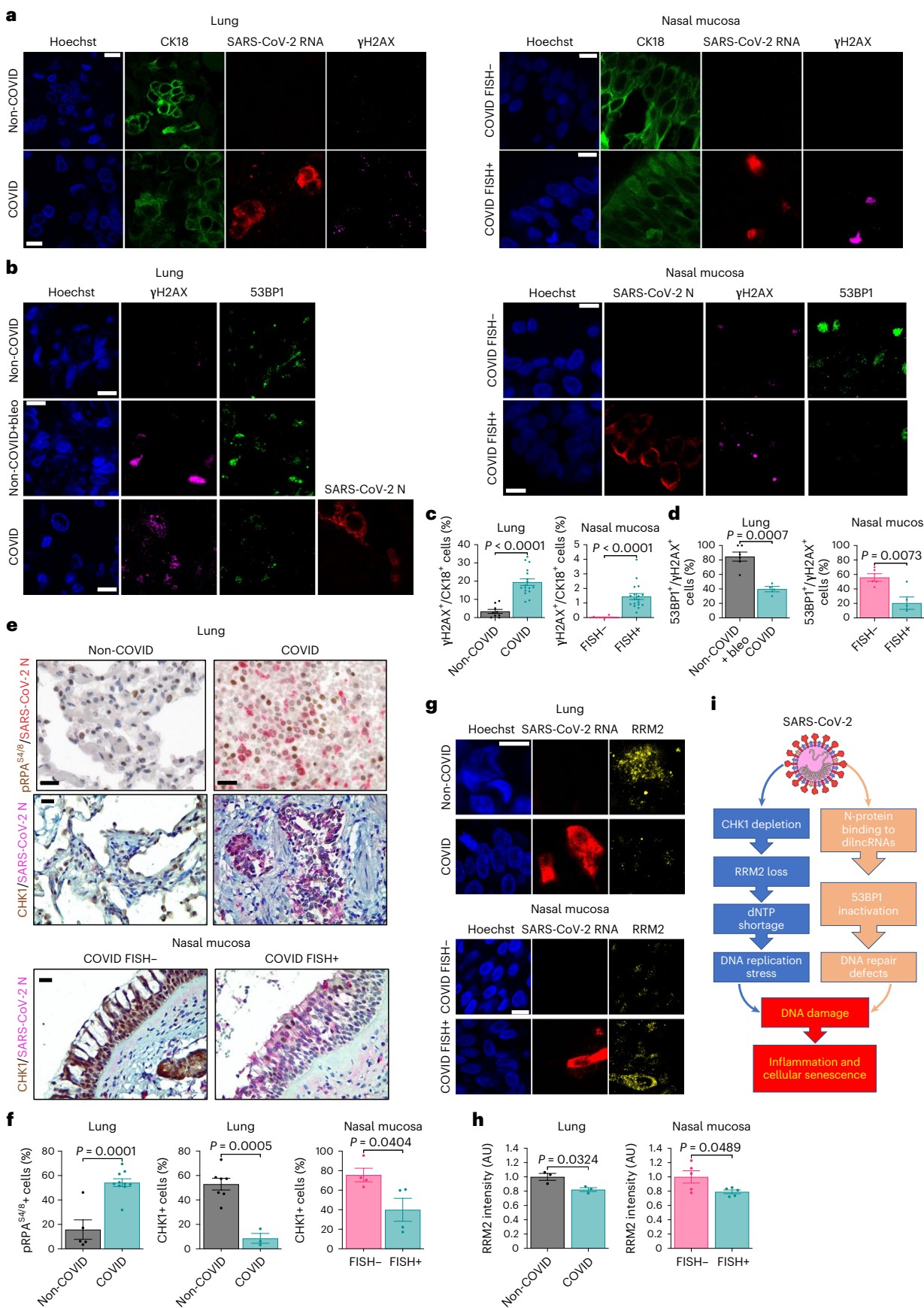

with previous studies[62,63], we observed that SARS-CoV-2 infection of cultured cells activates multiple pro-inflammatory signalling pathways, including cGAS/STING, STAT1 and p38/MAPK (Extended Data Fig. 1g–j), similar to CHK1 depletion (Extended Data Fig. 3e–j). Supported by reports that disruption of the CHK1–RRM2 pathway triggers cellular senescence[79] and our own evidence (Extended Data Figs. 1l–o, 7e–g and 8), we propose that SARS-CoV-2-mediated CHK1 loss promotes a pro-inflammatory programme akin to the senescence-associated secretory phenotype (Extended Data Fig. 3k–m).

A role for SARS-CoV-2-induced senescence in promoting macrophage infiltration and inflammation in vivo has been proposed[33]. We observed that SARS-CoV-2 infection causes DNA damage accumulation that correlates with markers of cellular senescence, in primary cells and in vivo (Extended Data Figs. 1l–o, 7e–g and 8). In particular, infected pneumocytes express high p21 levels, while polymorphonuclear and monocytoid inflammatory elements have elevated p16 (Extended Data Fig. 7g), reminiscent of a two-wave model of inflammatory response: an initial cell-intrinsic one and a second one triggered by the immune system[9,10].

Altogether, our results indicate that SARS-CoV-2-induced DNA damage triggers a cell-intrinsic pro-inflammatory programme that, in concert with the immune response, fuels the strong inflammatory response observed in patients with COVID-19. The observed ageing phenotypes recently reported in patients with severe COVID-19 (refs. [80,81]) are consistent with our observations.

Finally, by proposing a mechanism for the generation of DNA damage and the activation of DDR pathways and of a pro-inflammatory programme, we provide a model to improve our understanding of SARS-CoV-2-induced cellular senescence[9,33]. In this regard, it will also be interesting to determine if persistent DNA damage and DDR activation, features of cellular senescence[82,83], following SARS-CoV-2 infection, contribute to the chronic manifestations of the pathology known as long COVID[84].

## Online content

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

[1]IFOM ETS - The AIRC Institute of Molecular Oncology, Milan, Italy. [2]International Centre for Genetic Engineering and Biotechnology, Trieste, Italy. [3]University of Palermo, Palermo, Italy. [4]IRCCS San Raffaele Scientific Institute & University, Milan, Italy. [5]University of Trieste, Trieste, Italy. [6]University of Padova, Padova, Italy. [7]Fondazione IRCCS Istituto Neurologico Carlo Besta, Milan, Italy. [8]Cogentech Società Benefit srl, Milan, Italy. [9]Institute of Molecular Genetics (IGM), National Research Institute (CNR), Pavia, Italy. [10]Present address: Leibniz Institute for Experimental Virology (HPI), Hamburg, Germany. [11]These authors contributed equally: Ubaldo Gioia, Sara Tavella. ✉e-mail: fabrizio.dadda@ifom.eu

## Methods

### Mice and treatments

Experiments involving animals have been carried out in accordance with the Italian Laws (D.lgs. 26/2014), which enforce Directive 2010/63/EU (Directive 2010/63/EU of the European Parliament and of the Council of 22 September 2010 on the protection of animals used for scientific purposes). Accordingly, the project has been authorized by the Italian Competent Authority (Ministry of Health). B6.Cg-Tg(K18-ACE2)[2Prlmn]/J mice were purchased from The Jackson Laboratory. Mice were housed under specific-pathogen-free conditions as already described[85]. SARS-CoV-2 isolation and propagation for infection in mice has been carried out as shown before[85]. Virus infection of K18-hACE2 mice was performed via intranasal administration of $1 \times 10^5$ tissue culture infectious dose 50 per mouse under isoflurane 2% (# IsoVet250) anaesthesia, as described[85]. Mice were monitored to record body weight, clinical and respiratory parameters. Mice were killed by cervical dislocation. At the time of autopsy, mice were perfused through the right ventricle with phosphate-buffered saline (PBS). Lung tissues were collected in liquid nitrogen or in Zn-formalin and transferred into 70% ethanol 24 h later. For irradiation experiments, an 11-month-old C57BL/6J mouse was irradiated with 2 Gy TBI using GADGIL X-Ray irradiator. An age- and sex-matched mouse was used as control. Mice were killed by $CO_2$ inhalation at 1 h post-IR, and lungs were collected for fixation in 10% neutral buffered formalin overnight, washed in water and paraffin-embedded for histological analysis. RNA from mouse lungs was extracted and analysed as described[85].

### Patients diagnosed with COVID-19 and subjects negative for COVID-19

Histological analysis of lungs of patients with COVID-19 and subjects not diagnosed with the pathology (non-COVID)—but affected by viral pneumonia (influenza virus infection)—was performed by expert technicians and pathologists at the Pathology Unit of Trieste University Hospital. The same pathologists analysed all samples considered in this study, excluding operator-dependent biases. Viral presence in nasal mucosa of patients with COVID-19 was assessed by FISH against SARS-CoV-2 RNA. This study was approved by the Joint Ethical Committee of the Regione Friuli Venezia Giulia, Italy (re. 0019072/P/GEN/ARCS). All patients provided written informed consent to the use of their samples for research purposes at the time of hospital admission.

### Cell culture and treatments

Vero E6 (ATCC-1586), Huh7 (kindly provided by Ralf Bartenschlager, University of Heidelberg, Germany) and Calu-3 cell lines (ATCC HTB-55) were cultured in Dulbecco's modified Eagle medium (DMEM, ThermoFisher) supplemented with 10% foetal bovine serum (FBS, ThermoFisher) and 50 µg ml$^{-1}$ gentamicin. 53BP1-GFP U2OS, EJ5-GFP U2OS and NIH2/4 cell lines were cultured as already described[16,17,56]. Cell cultures were maintained at 37 °C under 5% $CO_2$. Cells were routinely tested for mycoplasma contamination. DNA damage was induced in cultured cells by 6 mM HU treatment for 4 h, or by exposure to IR (2 Gy) and analysed 15 or 60 min post-IR. Proteasome inhibition was conducted by incubating Huh7 cells for 6 h in growth medium containing 10 µM MG132 (M7449, Sigma). To inhibit autophagy, Huh7 cells were incubated with medium supplemented with either 100 nM Bafilomycin A1 (S1413, Selleckchem) or 50 µM CQ (C6628, Sigma) for 6 h, unless indicated otherwise.

### SARS-CoV-2 propagation and in vitro infection

Working stocks of SARS-CoV-2 ICGEB-FVG_5 isolated and sequenced in Trieste, Italy[86], were propagated on semiconfluent Vero E6 cells. Cultured Huh7 and Calu-3 cells were mock-infected or infected at a multiplicity of infection (MOI) of 0.1, in DMEM without supplements for 1 h at 37 °C and 5% $CO_2$. Then, the non-bound virus was rinsed off with PBS 1× and fresh DMEM containing 2% of heat-inactivated FBS was added to the cells. Uniformity of viral infection in all experiments was confirmed by viral titration, RNA detection and immunofluorescence.

### Isolation and SARS-CoV-2 infection of HNEpCs

HNEpCs were collected from healthy adult volunteers. Nasal cavities were anaesthetized using lidocaine, and nasal epithelial cells were collected by repeatedly scraping turbinates with a disposable bronchial cytology brush (CONMED). The tissue was resuspended in DMEM supplemented with 10% FBS and centrifuged at $300g$ for 5 min. The pellet was digested in a solution containing 1,000 U of Accutase (Sigma, A6964), 5,000 U of Dispase (Corning, 354235) and 1 mg ml$^{-1}$ DNAse II (Sigma, D8764) for 8 min at 37 °C. The digestion was stopped by adding an equal volume of Pneumacult medium (Stem Cell Technologies, 5050) supplemented with 10% FBS and filtered through a 100 µm cell strainer. The cell suspension was centrifuged at $300g$ for 5 min, and the pellet was resuspended in red blood cell lysis solution (150 mM $NH_4Cl$ and 10 mM $KHCO_3$) for 2 min. Cells were centrifuged at $300g$ for 5 min, resuspended in medium and seeded on eight-well chambers (Ibidi) at a concentration of $5 \times 10^5$ cells per well. After 48 h, cells were infected with SARS-CoV-2 (ref. [86]) at 1 MOI as described above. Cells were cultured for 48 or 72 h and fixed in 4% paraformaldehyde for 15 min at room temperature for staining.

### Immunofluorescence and FISH on HNEpCs

Fixed HNEpCs were permeabilized with 0.5% Triton X-100 (Sigma, T8787) in PBS for 15 min at room temperature. FISH to detect SARS-CoV-2 genome was performed using a kit by Molecular Instruments, following the manufacturer's instructions. After FISH, samples were incubated with a blocking solution (0.5% Triton X-100, 10% horse serum in PBS) for 30 min and incubated for 16 h at 4 °C with primary antibodies (Supplementary Table 1) diluted in the blocking solution. Samples were stained with the appropriate secondary antibody (Supplementary Table 1). Nuclei were counter-stained using Hoechst 33342 trihydrochloride (Invitrogen, h3570) and samples mounted using Mowiol (Sigma, 81381).

### Immunoblotting

Whole cell extracts were obtained by lysing Huh7 and Calu-3 in 1× Laemmli buffer (2% SDS, 10% glycerol and 60 mM Tris pH 6.8). Before fractionation on 4–12% gradient SDS–PAGE (ThermoFisher), whole extracts were boiled and sonicated for 15 s at low intensity using Bioruptor Next Gen (Diagenode) in a water bath at 4 °C. Proteins were then transferred onto nitrocellulose membrane and analysed as described before[56] with the antibodies listed in Supplementary Table 1. Bio-Rad Image Lab 6.1 was used for immunoblot data collection.

### Quantitative immunofluorescence analysis in cultured cells

Quantitative immunofluorescence assays were carried out in cultured cells as described[56], with minor modifications. Specifically, Calu-3 were fixed with 4% paraformaldehyde and permeabilized first with methanol/acetone (1:1) for 2 min at room temperature and then with Triton 0.2% in PBS 1× for 10 min at room temperature. The antibodies used are listed in Supplementary Table 1. Images at widefield microscope were acquired with MetaVue software or with Zen 2.0 Software (Zeiss); confocal microscope images were collected with Leica Application Suite X; data analyses were carried out by using CellProfiler 3.1.9 or ImageJ 1.53a.

### RNA extraction and RT–qPCR analysis

SARS-CoV-2-infected or mock-infected Huh7 and Calu-3 were collected in TriFast (EMR507100, Euroclone), and total RNA was extracted using RNeasy Kit (Qiagen). RNA from short interfering RNA (siRNA)-transfected Calu-3 and Huh7 was purified with Maxwell RSC simplyRNA Tissue Kit (Promega). DNase I was added during RNA purification, following the manufacturers' protocols. Purified total RNA was reverse transcribed with SuperScript VILO cDNA Synthesis

Kit (ThermoFisher), and complementary DNA was analysed by qPCR using SYBR Green I Master Mix (Roche) with the primers listed in Supplementary Table 2.

## Comet assay

Alkaline comet assay was performed on 48 h SARS-CoV-2-infected or mock-infected Huh7, or on 24 h infected or mock-infected Calu-3 using CometAssay Reagent Kit for Single Cell Gel Electrophoresis Assay (Trevigen, 4250-050-K), following the manufacturer's instructions. Tail moment was measured using CometScore 2.0 software.

## SA-β-gal assay

SA-β-gal activity was measured in 48 h SARS-CoV-2-infected or mock-infected Huh7 or in 72 h infected or mock-infected HNEpC as described[87]. Cells treated with 6 mM HU treatment for 4 h were used as positive controls of senescence induction.

## dNTP quantification

dNTP pools were extracted from 48 h SARS-CoV-2-infected or mock-infected Huh7 or from 24 h infected or mock-infected Calu-3. Cell plates were carefully washed free of medium with cold PBS and extracted with ice-cold 60% methanol. Methanolic extract was centrifuged, boiled for 3 min, brought to dryness by centrifugal evaporation (Savant, SC100 SpeedVac Concentrator and RT100A Refrigerated Condensation Trap) and stored at −80 °C until use. Cells left on the plate were dried and dissolved on 0.3 M NaOH, and the absorbance at 260 nm of the lysates was used as an index of cell mass, in turn an approximation for cell number[88]. The dry residue was dissolved in water and used to determine the size of dNTP pools by the DNA polymerase-based assay as described[89]. Two different aliquots of each pool extract were analysed, and pool sizes were normalized by the $A_{260nm}$ of the NaOH lysates.

## BrdU staining and flow cytometry analysis

For cell-cycle analysis, SARS-CoV-2-infected or mock-infected Huh7 were pulsed with 10 μM BrdU (Sigma-Aldrich) for 1 h. Then, cells were collected and fixed first in formaldehyde 2% and then in 75% ethanol. Cells were probed with anti-BrdU primary antibody (Supplementary Table 1) diluted in PBS 1% bovine serum albumin (BSA) at room temperature for 1 h. After washing, cells were incubated with the secondary antibody, diluted 1:400, at room temperature for 1 h in the same buffer. Finally, cells were stained with PI (Sigma-Aldrich, 50 μg ml⁻¹) in PBS 1% BSA and RNase A (Sigma-Aldrich, 250 μg ml⁻¹). Samples were acquired with Attune NxT (ThermoFisher) using a 561 nm laser and 695/40 filter for PI; 488 nm laser and 530/30 filter for BrdU. Analysis was carried out using FlowJo 10.7.1 (BD Biosciences). At least $10^4$ events were analysed for each sample.

## Plasmid and siRNA transfection

pLVX-EF1α-2xStrep-IRES-Puro vectors encoding for SARS-CoV-2 proteins are a kind gift from Professor Nevan J. Krogan[40]. pCAGGS-ORF6 and -ORF6[M58R] vectors were gently provided by Lisa Miorin[41]. pcDNA3.1 (+) Mammalian Expression Vector (ThermoFisher) was used as a control EV where indicated. Plasmids were transfected with Lipofectamine 2000 Transfection Reagent (ThermoFisher) in Opti-MEM (ThermoFisher). siRNAs were purchased from Dharmacon (Horizon) and transfected into Huh7 or Calu-3 with Lipofectamine RNAiMAX Transfection Reagent (ThermoFisher) in Opti-MEM. Cells were collected for analyses at 48 or 72 h post-transfection.

## Multiplex immunoassay

Bio-Plex Pro-Human Cytokine Immunoassay kits (Bio-Rad) were used to measure the concentration of pro-inflammatory cytokines and chemokines in cell supernatants, according to the manufacturer's guidelines. The magnetic-bead-based antibody detection kit allows simultaneous quantification of the analytes of interest. Each sample was tested undiluted and in duplicate. Positive and negative controls were included in the plate. The plate was read on the Bio-Plex 200 system (Bio-Rad), powered by the Luminex xMAP technology. The concentration of analyte bound to each bead was proportional to the median fluorescence intensity of the reporter signal, and was corrected by the standards provided in the kit (Bio-Rad). Data were expressed in pg ml⁻¹.

## dN supplementation

Following exposure to SARS-CoV-2 (0.1 MOI) for 1 h, Huh7 were washed once with PBS 1× and then incubated with fresh DMEM containing 2% of heat-inactivated FBS supplemented with 5 μM deoxy-adenosine (dA), deoxy-cytidine (dC) and deoxy-guanidine (dG)[90] (D8668, D0776 and D0901, Sigma) for 48 h before being collected for downstream analyses. dN supplementation in Calu-3 was carried out by incubating cells with medium containing 50 μM dA and dC, 5 μM dG and deoxy-thymidine (dT, T1895, Sigma) for 24 h.

## Ubiquitination assay

Forty-eight hours post plasmid transfection, Huh7 were lysed with IP buffer (150 mM KCl, 25 mM Tris–HCl pH 7.4, 5% glycerol, 0.5% NP40, 10 mM $MgCl_2$ and 1 mM $CaCl_2$), supplemented with 1× protease inhibitors (Roche) and 250 U ml⁻¹ Benzonase (E1014-25KU, Sigma) for 45 min at 4 °C. Lysates were cleared by centrifugation and supernatants incubated overnight at 4 °C with 1:200 of anti-CHK1 (Supplementary Table 1), previously coupled with Protein G Dynabeads (10004D, ThermoFisher). After five washes with IP buffer, proteins were eluted with 1× LDS buffer (B0007, ThermoFisher), separated by 4–12% SDS–PAGE (ThermoFisher) and transferred onto nitrocellulose membrane (Amersham). Membrane was autoclaved before overnight blocking at 4 °C in 5% BSA TBS-T. After blocking, membrane was incubated with anti-Ubiquitin antibody (Supplementary Table 1) 1:1,000 in 5% BSA TBS-T for 1 h at room temperature.

## SARS-CoV-2 N-protein expression and purification

pET28 plasmid bearing SARS-CoV-2 N-protein fused to a C-terminal His₆-tag (a kind gift from S. Pasqualato, Human Technopole, Milan, Italy) was expressed in *Escherichia coli* BL21-CodonPlus (DE3)-RP (Agilent) upon induction with 500 μM isopropyl-β-ᴅ-1-thiogalactopyranoside for 16 h at 18 °C. Cells were resuspended in resuspension buffer (25 mM Tris pH 8, 500 mM NaCl, 5% glycerol, 2 mM β-mercaptoethanol and 10 mM imidazole), supplemented with protease inhibitors and TurboNuclease and sonicated. After polyethyleneimine addition and centrifugation, the supernatant was applied onto Ni-NTA agarose beads (Qiagen) and the His₆-nucleocapsid protein was eluted in elution buffers containing 250 mM and 500 mM imidazole. The eluted protein was applied on a HiTrap Heparin HP column (Cytiva) and loaded on a Superdex 200 Increase 10/300 GL (Cytiva) pre-equilibrated in SEC buffer (25 mM Tris pH 8, 500 mM NaCl and 2 mM β-mercaptoethanol). Protein purity was assessed by Comassie blue SDS–PAGE as >90% pure.

## Micro-injection and live imaging analysis

Before micro-injection experiments, both purified N-protein and acetylated BSA (ThermoFisher) were dialysed overnight against micro-injection buffer (25 mM Tris pH 8 and 150 mM NaCl). Micro-injection experiments were performed as already described[17], with minor changes. An UltraVIEW VoX spinning-disk confocal system (PerkinElmer) with a motorized Luigs & Neumann SM7 micromanipulator was used. Glass borosilicate capillaries were pulled to a diameter of 0.7 μm using a Flaming/Brown micropipette puller Model P-1000 (Sutter Instrument) and loaded with 6 mg ml⁻¹ N-protein or BSA solution as control, in presence of Rhodamine B Dextran (ThermoFisher) as marker for micro-injected cells. Samples were injected into cell nuclei 30 min after irradiation (2 Gy) using a FemtoJet pump (Eppendorf) at a constant pressure of 20 hPa. A z-stack was acquired every minute for 100 min total using a 60× oil-immersion objective (Nikon Plan Apo VC,

1.4 numerical aperture). To excite GFP and Rhodamine B, 488 nm and 561 nm lasers were used, respectively. Images in micro-injection experiments were acquired with Volocity 6.4.0. Quantification of 53BP1-GFP foci per nucleus per single frame was performed using the software CellProfiler 3.1.9. Foci number per nucleus was plotted along time, and the corresponding curve was fitted to an exponential function to determine the decay rate ($k$) using Prism 9.3.0 software.

### Turbidity assay
Purified recombinant N-protein (5 mM) in micro-injection buffer (25 mM Tris pH 8, 150 mM NaCl) was mixed with increasing concentrations of RNA extracted from SARS-CoV-2-infected (48 h) or mock-infected Huh7. Turbidity of the solutions was measured at 350 nm using a NanoVue Plus Spectrophotometer (Cytiva) after 30 min, 2 h and 24 h of incubation at room temperature.

### SARS-CoV-2 N IP
Huh7 were transfected with the plasmid encoding for N-protein[40] or with an EV as control. Forty-eight hours post-transfection, cells were irradiated or not. One hour post-IR, cells were collected by trypsinization and washed twice in ice-cold 1× PBS. Cell pellets were lysed in IP buffer supplemented with 1× protease inhibitors (Roche), 0.5 mM dithiothreitol, 40 U ml$^{-1}$ RNaseOUT and 1,000 U ml$^{-1}$ DNase I (Roche), and incubated at room temperature for 15 min and at 4 °C for an extra 15 min with gentle rotation. Lysates were cleared by centrifugation at maximum speed for 20 min at 4 °C. After addition of 5 mM EDTA (pH 8), lysates were incubated overnight at 4 °C with 5 µg of anti-N-protein (40588-T62 Sino Biological) or with normal rabbit IgGs (Cell Signaling), which were previously bound to Dynabeads Protein G (ThermoFisher). After five washes with 1 ml of IP buffer, immunoprecipitated proteins were analysed by immunoblotting.

### RIP and analysis
RIP and dilncRNA analysis was carried out in NIH2/4 as previously described[16], with minor modifications. Briefly, I-SceI-GR-expressing NIH2/4 cells were transfected with the plasmid encoding for N-protein or with an EV as a control at 24 h before triamcinolone acetonide 0.1 µM (Sigma-Aldrich) administration. IP was performed using 5 µg of anti-N-protein, or 10 µg of anti-53BP1 (Supplementary Table 1), or with normal rabbit IgGs (Cell Signaling) as a mock IP.

### NHEJ repair assay
Evaluation of repair efficiency by NHEJ following viral N-protein expression was conducted as shown previously[56], with minor modification: HA-I-SceI expressing plasmid was transfected along with the plasmid encoding for SARS-CoV-2 N-protein.

### Immunolocalization, in situ RNA hybridization and analyses in mouse and human paraffin-embedded tissues
IHC staining was performed as previously reported[91]. Double IHC staining was conducted by applying SignalStainBoost IHC Detection (#18653; #31926, Cell Signaling) and Vulcan Fast Red as substrate chromogen.

For combined ISH and IHC staining, SARS-CoV-2 RNA probe hybridization (RNAscope Probe - V-nCoV2019-S; 848561; ACD) was performed using RNAscope 2.5 HD Detection Reagent-BROWN (ACD), following the manufacturer's protocol. After blocking, samples were incubated with primary antibodies (Supplementary Table 1) following by SignalStainBoost IHC Detection and Vulcan Fast Red. Nuclei were counter-stained with Gill's Hematoxylin n.1 (Bio-Optica). Quantitative analyses of IHC experiments were performed by calculating the average percentage of positive signals in five non-overlapping fields at medium-power magnification (×200) using the Nuclear v9 or Positive Pixel Count v9 ImageScope software (v12.3.2.8013, Leica Biosystems). Combined ISH/IHC images were analysed through the use

of a segmentation-based software (HALO v3.5.3577.140, Indica Labs), quantifying the percentage of positive cells.

Double-marker immunofluorescence was conducted as previously described[92].

### Statistics and reproducibility
All the data are shown as mean ± standard error of the mean (s.e.m.) of three independent experiments, unless stated otherwise in the figure legends. We did not use any criteria to determine the sample size. As much data as possible was collected depending on the nature of the experiments or to allow statistical analysis. Where single-cell analysis was performed, we scored at least 30 cells per condition in each biological replicate. Throughout the manuscript, no data were excluded. Only in rare occasions were individual data points removed following unbiased criteria of outlier identification using Prism 9.3.0 software. In addition, in the case of IHC staining in infected mouse lungs, we excluded the tissue sections that show low rates of SARS-CoV-2 infection. The number of animals and replicates are indicated in each figure legend. Animals were randomized to the experimental groups. For in vitro experiments, wells were randomly assigned into each group and all cells were analysed equally. No blinding method was applied. No statistical method was used to pre-determine sample sizes. Ordinary one-way analysis of variance (ANOVA) with Dunnet's post hoc test was applied in Figs. 1b and 2b and Extended Data Figs. 1b, 2d–f, 4d and 5c. Unpaired two-tailed $t$-test was applied in Figs. 1d,f, 2d, 6b,d,g, 7b and 8b,d,f,h and Extended Data Figs. 1d,f,h,o, 2b,n,o,p,r, 5b, 6d,f and 7b. Multiple paired two-tailed $t$-test was applied in Extended Data Fig. 1j. Ordinary one-way ANOVA with Fisher's least significant difference test was applied in Figs. 2j,l,n,o, 3d, 4h and 5b,f and Extended Data Figs. 1m and 2q. Ordinary one-way ANOVA with Šidák's post hoc test was applied in Figs. 2e, 4j and 6f,i,j. Ordinary two-way ANOVA with Šidák's post hoc test was applied in Fig. 2g and Extended Data Fig. 5d. Ratio paired two-tailed $t$-test was applied in Figs. 2h and 6h. Ordinary two-way ANOVA with Bonferroni's post hoc test was applied in Extended Data Fig. 2k. Repeated measures one-way ANOVA with Fisher's least significant difference test was applied in Extended Data Fig. 2s. Ordinary one-way ANOVA with Tukey's post hoc test was applied in Figs. 3b, 4b,d,f and 5d and Extended Data Figs. 4b and 5k.

### Reporting summary
Further information on research design is available in the Nature Portfolio Reporting Summary linked to this article.

## Data availability
Source data are provided with this paper, available online for Figs. 1–8 and Extended Data Figs. 1–8. All other data that support the findings of this study are available from the corresponding author on reasonable request.

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

## Acknowledgements

We thank N. J. Krogan (UCSC, San Francisco, CA, USA), S. Pasqualato (Human Technopole, Milan, Italy) and L. Miorin (Icahn School of Medicine at Mount Sinai, New York, NY, USA) for reagents; S. Martone and M. G. Totaro (IFOM, Milan, Italy) for technical support with flow cytometry analyses; S. Magni, E. Martini and D. Parazzoli (IFOM, Milan, Italy) for support with imaging analyses; L. Falbo (IFOM, Milan, Italy) for providing antibodies; E. Maspero and S. Polo (IFOM, Milan, Italy) for suggestions for ubiquitination analyses; S. Martin and S. Santaguida (IEO, Milan, Italy) for support with autophagy studies and reagents; V. Cianfanelli, V. Nazio and F. Cecconi for support with autophagy studies; F. Giavazzi (Università degli Studi di Milano, Segrate, Italy) for helpful advice with micro-injection analyses. M.I. is supported by the European Research Council (ERC) Consolidator Grant (725038), ERC Proof of Concept Grant (957502), Italian Association for Cancer Research (AIRC) Grants (19891, 22737), Italian Ministry of Health Grant (RF-2018-12365801) and Funded Research Agreements from Gilead Sciences, Asher Bio, Takis Biotech and Vir Biotechnology. V.F. is supported by a donation from FONDAZIONE PROSSIMO MIO and from AIRC Fellowship for Italy (26813-2021). C.T. laboratory is supported by AIRC 5 × 1000 2019 (22759). The work in A.M. laboratory related to SARS-CoV-2 is supported by grants from SNAM Foundation, Generali SpA, Beneficentia Stiftung, CARIPLO (INNATE-CoV) and the #FarmaCovid crowdfunding initiative. S.Z. laboratory is supported by AIRC IG (24529). F.d'A.d.F laboratory is supported by: ERC advanced grant (TELORNAGING—835103); AIRC IG (21762); Telethon (GGP17111); AIRC 5 × 1000 (21091); ERC PoC grant (FIREQUENCER—875139); Progetti di Ricerca di Interesse Nazionale (PRIN) 2015 'ATR and ATM-mediated control of chromosome integrity and cell plasticity'; Progetti di Ricerca di Interesse Nazionale (PRIN) 2017 'RNA and genome Instability'; Progetto AriSLA 2021 'DDR & ALS'; POR FESR 2014–2020 Regione Lombardia (InterSLA project); FRRB—Fondazione Regionale per la Ricerca Biomedica—under the frame of EJP RD, the European Joint Programme on Rare Diseases with funding from the European Union's Horizon 2020 research and innovation programme under the EJP RD COFUND-EJP NO 825575. S.T. was a PhD student within the European School of Molecular Medicine (SEMM).

## Author contributions

U.G. conceived, conducted and analysed RIP, co-IP, comet, ubiquitination and NHEJ repair assays; performed immunoblot analyses; expressed individual SARS-CoV-2 proteins in cultured cells and analysed their impact on DDR activation by immunoblot; designed, carried out and analysed the experiments upon CHK1 knockdown by RNAi; analysed all the remaining data; wrote and edited the manuscript. S.T. performed immunofluorescence staining of infected and transfected cultured cells, carried out the immunostaining for flow-cytometry analyses, generated cDNA from infected cultured cells, conceived the autophagy experiments, contributed to design the multiplex immunoassays and the dN supplementation experiments, and wrote and edited the manuscript. P.M.-O. and S.R. performed all the SARS-CoV-2 infection assays, measured virus by immunoblot, immunofluorescence, plaque assay and RT–PCR and prepared samples for analyses. P.M.-O. conducted SA-β-gal assay. G.C. performed IHC analyses on mouse and human specimens. A.C. and P.M.-O. performed the analyses of DDR activation in infected HNEpC. A.C. and A.P. carried out FISH in infected HNEpC, IHC and immunofluorescence analyses in human lung and nasal mucosa tissues. M.Ce. produced recombinant SARS-CoV-2 N and performed micro-injection experiments and turbidity assays. M.Ca. purified and analysed by RT–qPCR the RNA of infected cultured cells and CHK1-depleted cells. A.C.H. provided technical support with immunoblots and immunofluorescence. V.F. propagated SARS-CoV-2, infected *hACE2* mice and generated lung tissues from such animals. E.P. performed the analysis of dNTP levels in infected cultured cells. N.I. performed Bio-Plex multiplex immunoassays. F.P. contributed to set up IHC in mouse specimens. V.M. contributed to the analyses of RNA of infected cultured cells by RT–qPCR. S.S. generated lung samples from irradiated or control mice. M.I.C. analysed flow cytometry experiments. S.B. and Z.L. assisted M.Ce. in conducting and analysing micro-injection experiments. T.C. assisted in the management of the BSL3 facility and virus stock preparation. M.C.V. prepared and characterized nasal primary cells. P.C. designed Bio-Plex multiplex immunoassays, supervised N.I. and edited the manuscript. M.I. supervised V.F. C.R. designed the quantification of dNTP levels, the dN supplementation experiments, supervised E.P. and edited the manuscript. R.B. performed post-mortem pathology on human lungs. C.T. supervised G.C. and edited the manuscript. S.Z. coordinated the analysis of human tissues, supervised A.C., A.P. and M.C.V., and edited the manuscript. A.M. coordinated the virology aspects of the work, supervised P.M.-O., S.R. and T.C., and edited the manuscript. F.d'A.d.F. conceived the study, coordinated experimental activities, and assembled and revised the manuscript.

## Competing interests

M.I. participates in advisory boards/consultancies for Gilead Sciences, Roche, Third Rock Ventures, Antios Therapeutics, Asher Bio, Amgen, Allovir. All the other authors declare no competing interests.

## Additional information

**Extended data** is available for this paper at https://doi.org/10.1038/s41556-023-01096-x.

**Correspondence and requests for materials** should be addressed to Fabrizio d'Adda di Fagagna.

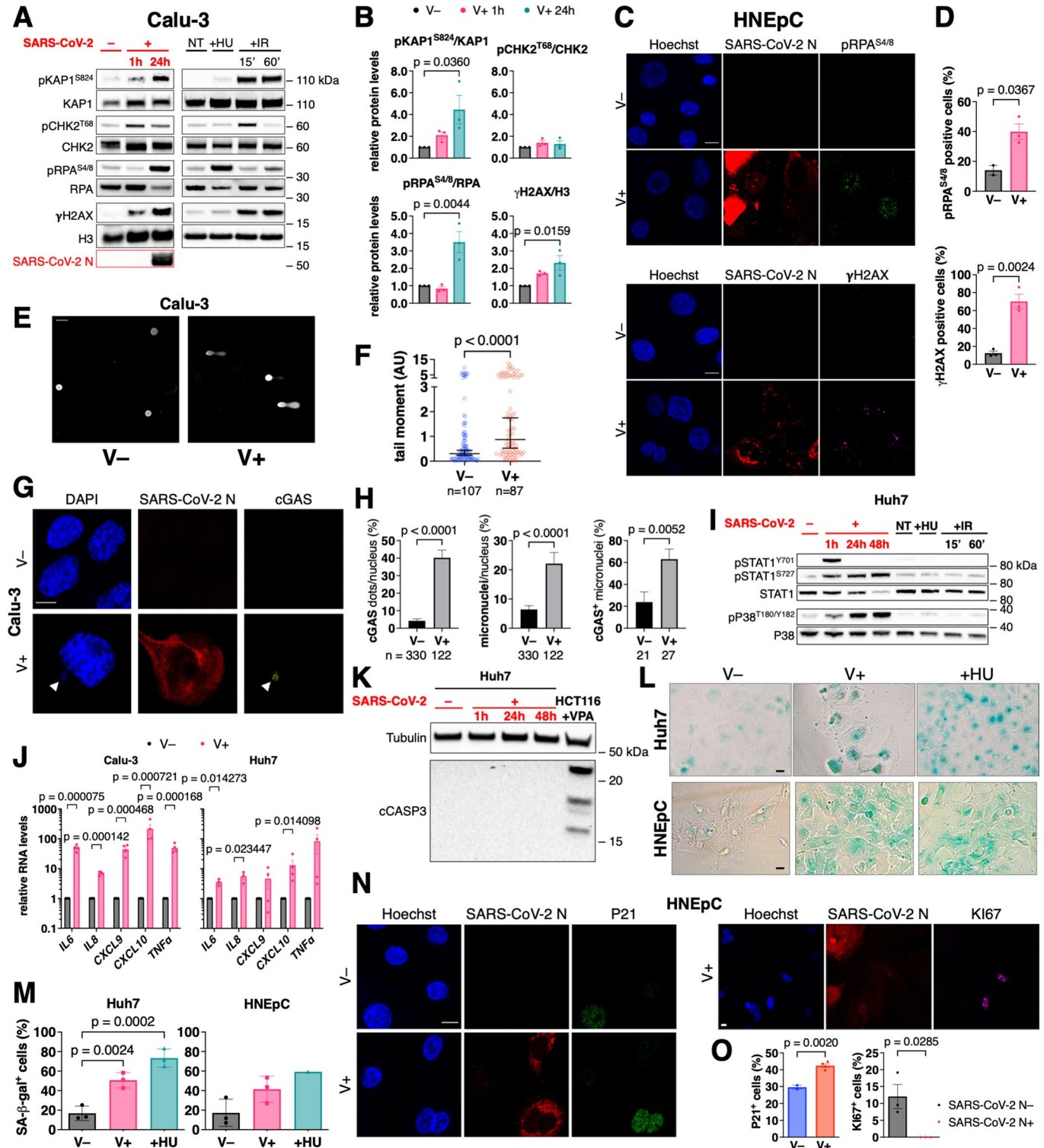

**Extended Data Fig. 1 | SARS-CoV-2 induces DNA damage, inflammation and cellular senescence. A**) Immunoblotting of Calu-3, infected (V+) or not (V−) with SARS-CoV-2. Calu-3 exposed or not (NT) to HU or IR were used as controls. Infection was monitored by probing for N-protein. **B**) Quantification of protein levels shown in A; values are relative to V−. **C**) IF images of HNEpC; nuclei were stained with Hoechst. **D**) Quantification of the percentage of γH2AX or pRPA^S4/8 positive cells shown in C. **E**) Images of comet assays of Calu-3. Scale bar, 50 μm. **F**) Quantification of comet tail moment shown in **E**. Horizontal bars represent the median values ± 95% CI of three independent infections. **G**) IF images of Calu-3; nuclei were stained with DAPI; arrows point to cGAS⁺ micronuclei. **H**) Percentage of cells positive for cGAS dots, micronuclei and cGAS⁺ micronuclei shown in G. Values are the means ± s.e.m. **I**) Immunoblots from samples described in Fig. 1A.

The experiment was repeated three times with similar results. **J**) RT–qPCR for pro-inflammatory cytokines expression in Calu-3 and Huh7, 24 h and 48 h after infection, respectively. Values are the means ± s.e.m. of four independent experiments, except for *IL6* and *IL8* in Huh7 (*n* = 3) and shown as relative to V−. **K**) Immunoblotting for cleaved Caspase-3 (cCASP3) from Huh7 lysates. Valproic acid (VPA)-treated HCT116 were used as positive control. Tubulin was used as loading control. The experiment was repeated three times with similar results. **L**) Images of Huh7 and HNEpC stained for SA-β-gal; HU-treated cells were used as positive control. **M**) Quantification of positive cells shown in L. **N**) IF images of HNEpC. **O**) Percentage of P21- or KI67-positive cells. Scale bar, 10 μm for panels **C**, **G**, **L**, **N**. Source numerical data and unprocessed blots are available in source data.

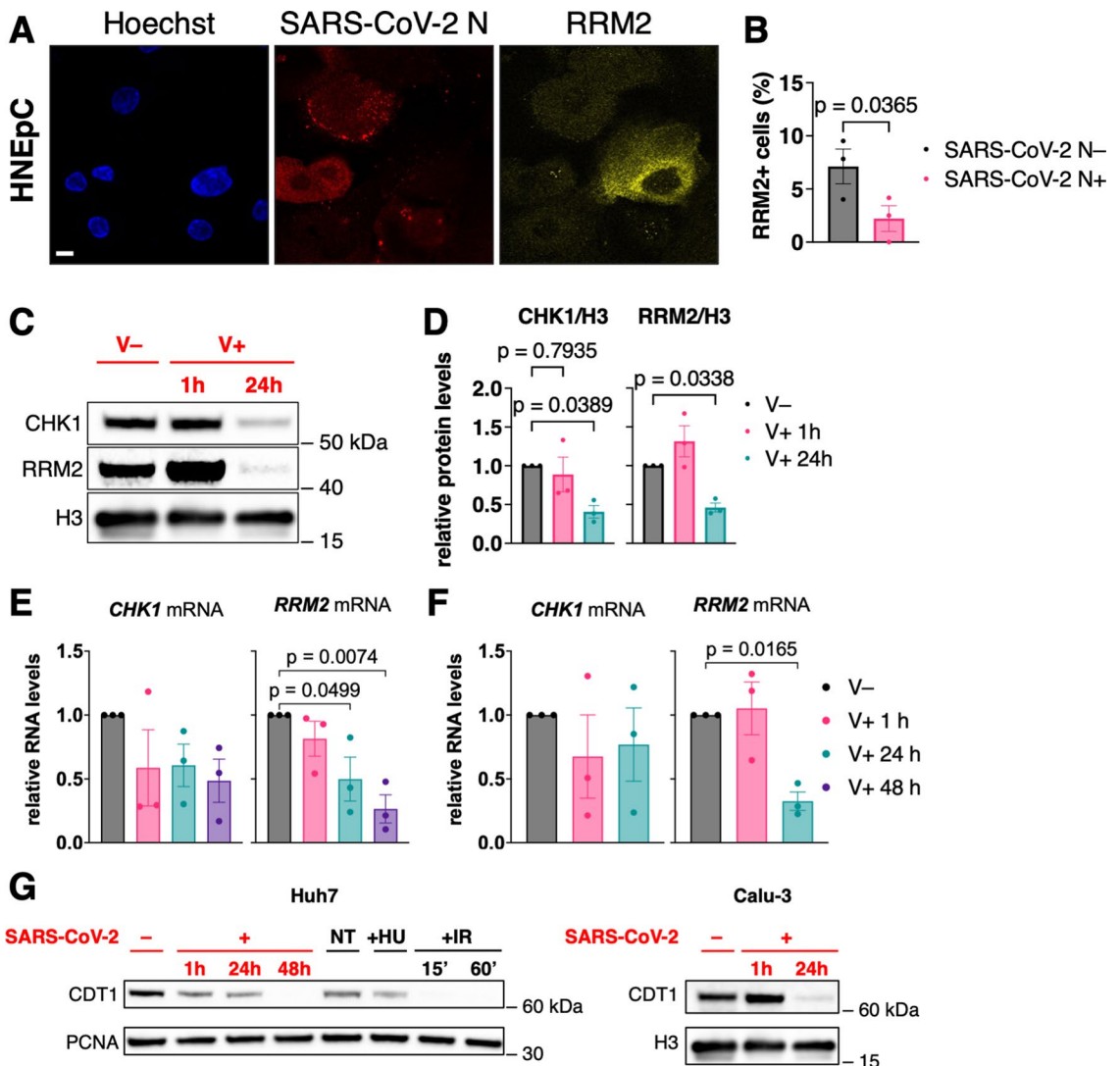

**Extended Data Fig. 2 | SARS-CoV-2 causes CHK1 and RRM2 reduction leading to cell cycle progression impairment. A**) IF images of infected HNEpC; nuclei were stained with Hoechst. Scale bar, 10 μm. **B**) Quantification of the percentage of RRM2-expressing cells in infected (SARS-CoV-2 N + ) or not (SARS-CoV-2 N-) HNEpC. **C**) Immunoblotting of whole cell lysates of Calu-3 infected, or not, with SARS-CoV-2 and analyzed at different time points post-infection.

**D**) Quantification of protein levels shown in **C**; values are shown as relative to mock-infected samples. **E,F**) RT–qPCR of *CHK1* and *RRM2* mRNA expression in infected (V+ ) or mock-infected (V−) Huh7 and Calu-3 cells, respectively. **G**) Immunoblotting of CDT1 in Huh7 and Calu-3 cells treated as indicated. The experiment was repeated three times with similar results. Source numerical data and unprocessed blots are available in source data.

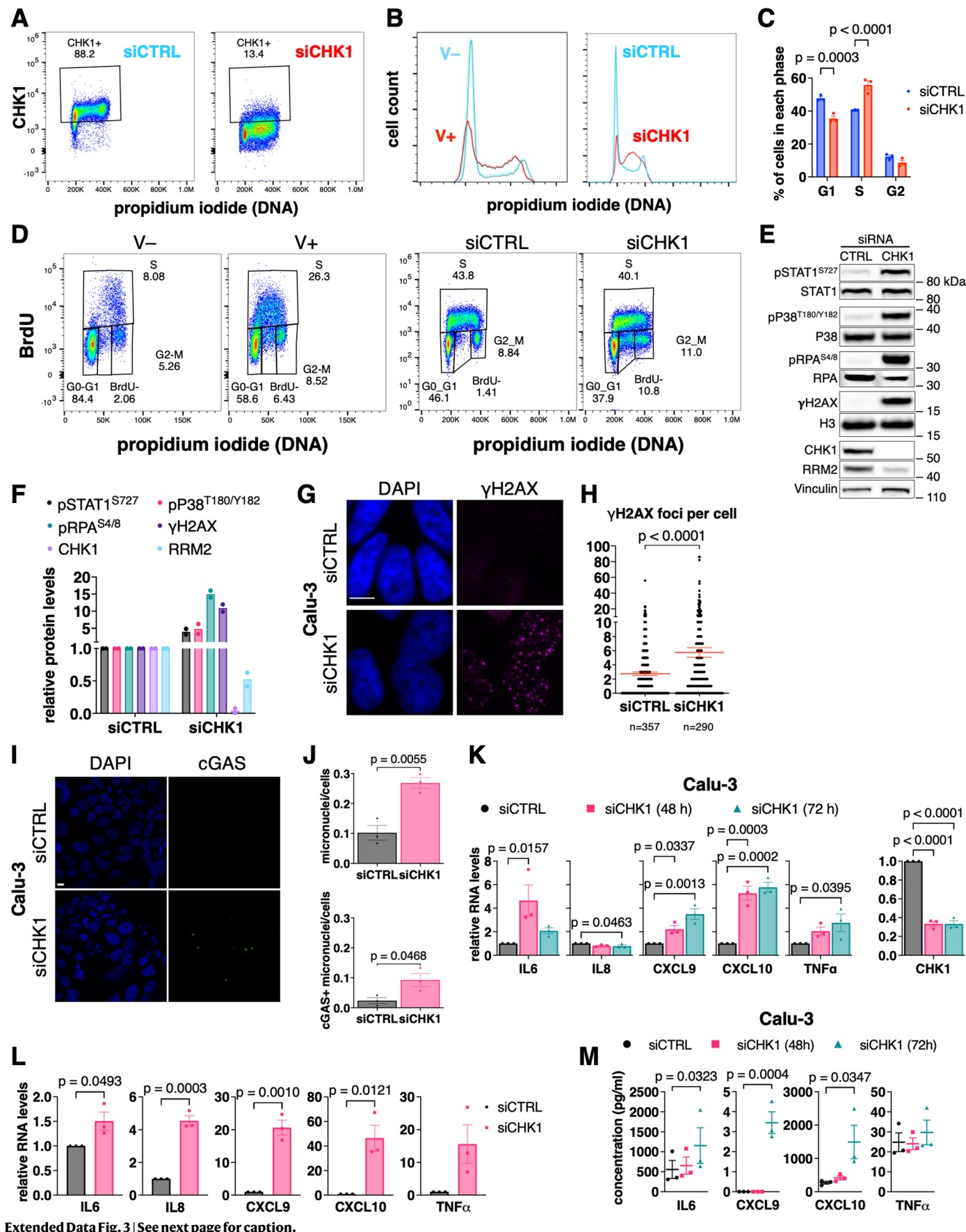

**Extended Data Fig. 3 | See next page for caption.**

**Extended Data Fig. 3 | CHK1 depletion is sufficient to recapitulate the effects of SARS-CoV-2 infection. A**) Huh7 transfected with siRNAs against *CHK1* mRNA (siCHK1) or siCTRL were stained for CHK1 and propidium iodide (PI) prior to flow cytometry. **B**) DNA content analysis of V− or V+ Huh7 fixed 48 h post-infection and siCHK1- or siCTRL-transfected Huh7. **C**) Histograms show the percentage of cells in each phase of the cell cycle upon siCHK1 or siCTRL treatment. **D**) Bivariate plot showing DNA content (PI) and BrdU incorporation measured by flow cytometry of V− or V+ Huh7 fixed 48 h post-infection and siCHK1- or siCTRL-transfected Huh7. **E**) Immunoblots of siCHK1- or siCTRL-transfected Huh7. **F**) Quantification of protein levels shown in **E**; values are the means ± s.e.m. of two independent experiments and shown as relative to the siCTRL-transfected sample. **G**) IF images of Calu-3 transfected with the indicated siRNAs; nuclei were stained with DAPI. Scale bar, 10 μm. **H**) Quantification of γH2AX foci per cell shown in G. **I**) IF images of cGAS staining in samples as in **G**; nuclei were stained with DAPI. Scale bar, 10 μm. **J**) Micronuclei and cGAS⁺ micronuclei quantifications on total cell number; at least 300 nuclei were scored for each sample. **K,L**) RT−qPCR for pro-inflammatory cytokines and *CHK1* mRNA expression in siCHK1-treated Calu-3 and Huh7 cells, respectively. Values are shown as relative to siCTRL-transfected samples. **M**) Quantification of the amounts of secreted cytokines and chemokines from siCHK1- or siCTRL-transfected Calu-3 by Bio-Plex multiplex immunoassays. Source numerical data and unprocessed blots are available in source data.

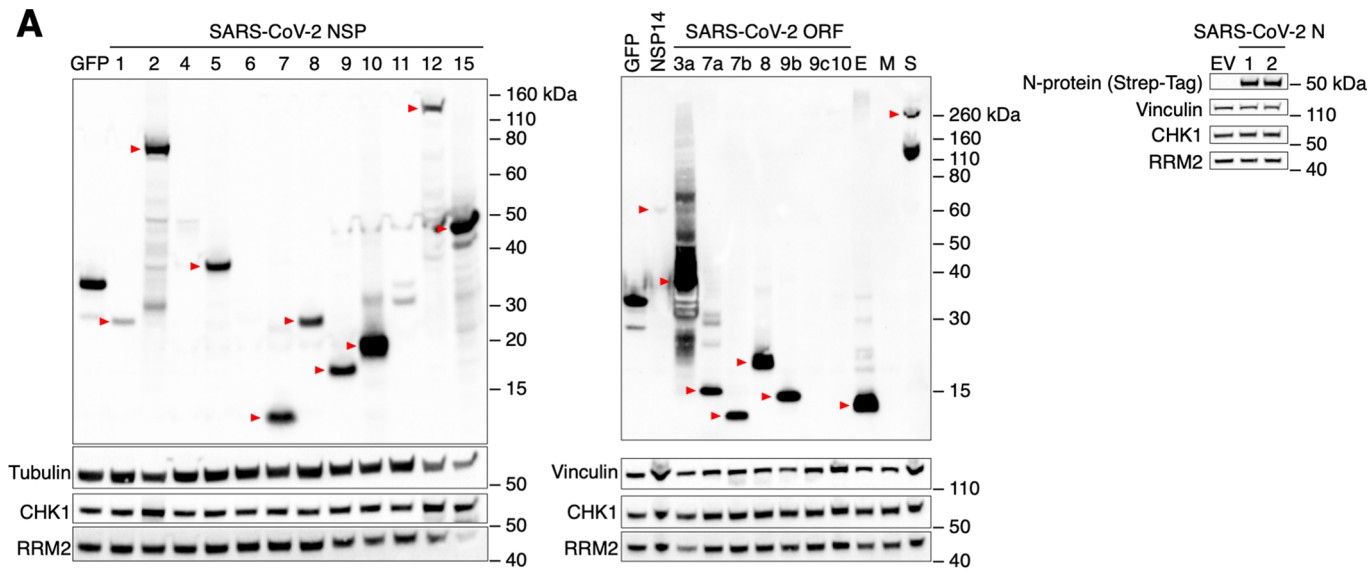

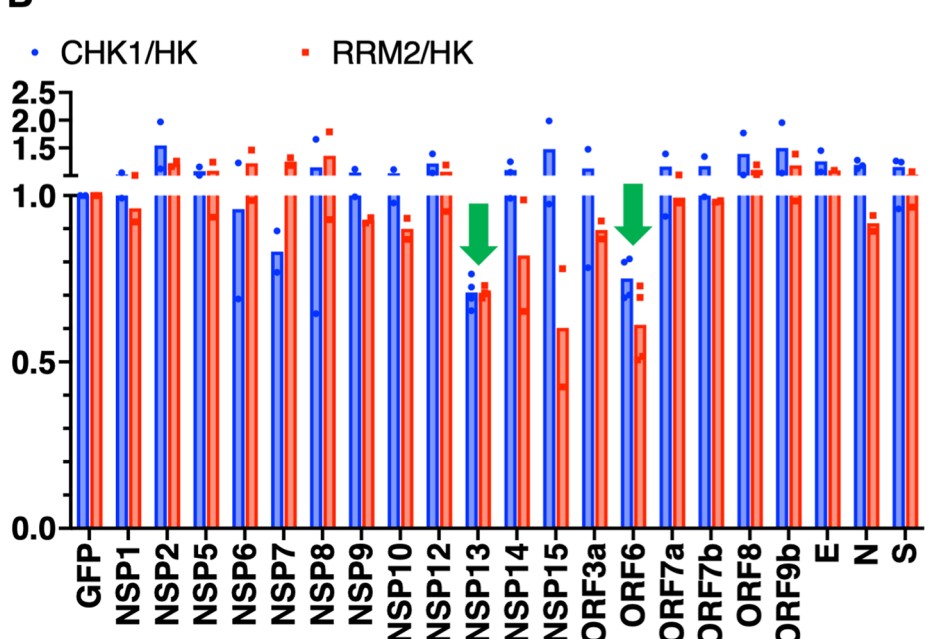

**Extended Data Fig. 4 | Impact of SARS-CoV-2 proteins on CHK1 and RRM2 expression. A**) Immunoblotting of whole cell lysates from Huh7 cells transfected with plasmids encoding for the indicated SARS-CoV-2 proteins and stained for CHK1 and RRM2. Predominant bands at the expected molecular weights are indicated by red arrows. **B**) Quantification of CHK1 and RRM2 protein levels shown in **A**; values are the means of two independent experiments, except for NSP13 and ORF6 ($n = 4$); protein levels are normalized to either vinculin or tubulin protein amounts (house-keeper, HK) and shown as relative to the control-sample expressing GFP. Arrows indicate the viral proteins that most affected both CHK1 and RRM2 levels. Huh7 cells did not express detectable levels of the following viral proteins: NSP4, NSP11, ORF9c, ORF10, M and therefore CHK1 and RRM2 protein levels where not quantified. Source numerical data and unprocessed blots are available in source data.

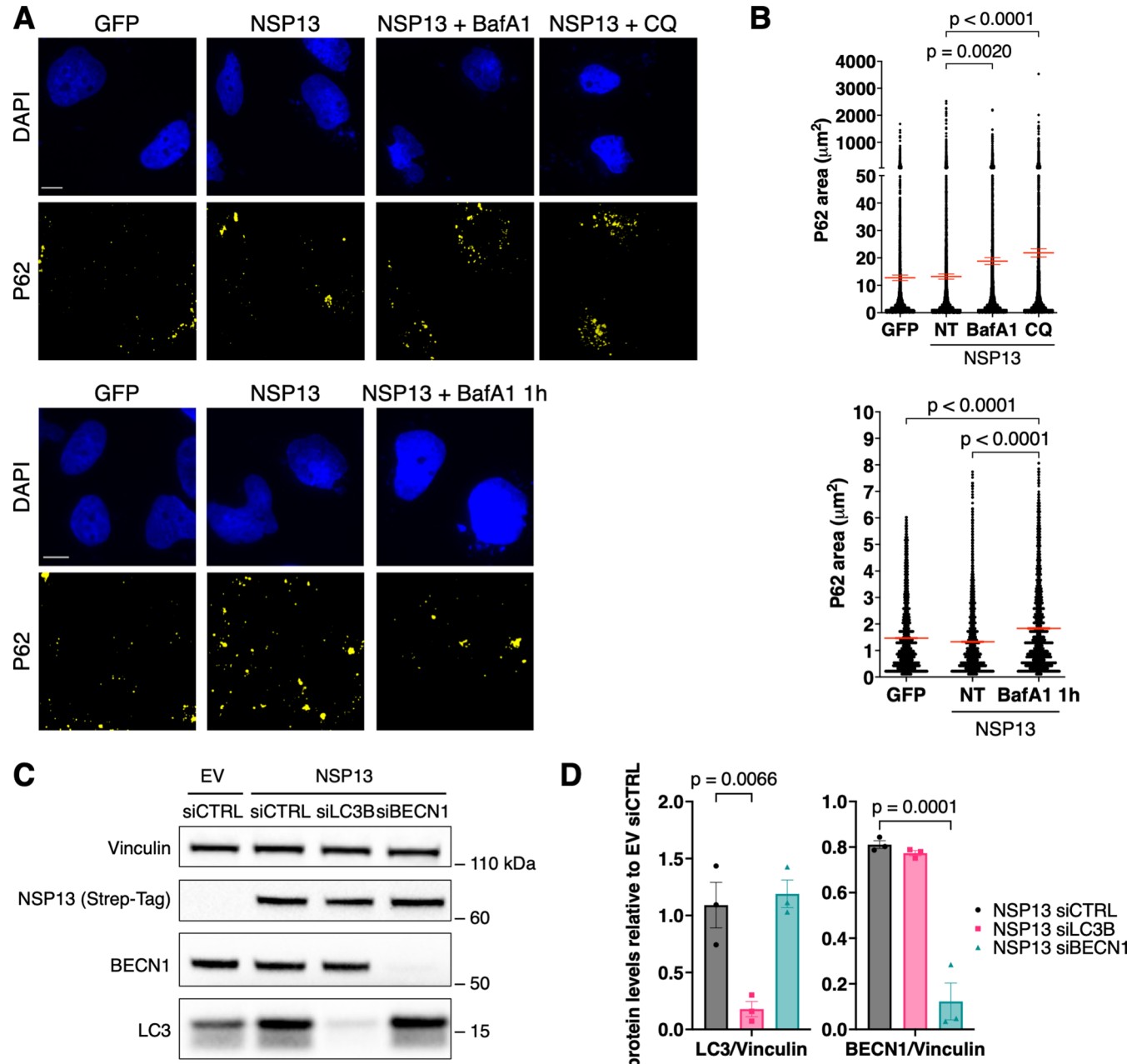

**Extended Data Fig. 5 | Controls for autophagy experiments. A)** Representative IF images of autophagosome accumulation in Huh7 cells following GFP or viral NSP13 expression and treatment with the autophagy inhibitors BafA1 and CQ. Autophagosomes were visualized by probing for P62; nuclei were stained with DAPI. Scale bar, 10 μm. **B)** Quantification of the area (μm²) of cytoplasmic P62⁺ aggregates shown in A. At least 100 cells were scored for each sample.

**C)** Representative immunoblots showing the efficiency of BECN1 and LC3B knock-down in Huh7 cells expressing NSP13; as control, cells were transfected with an EV. **D)** Quantification of LC3 and BECN1 protein levels shown in **C.** Values are shown as relative to samples transfected with a non-targeting siRNA and with the EV siCTRL. Vinculin was used as a loading control. Source numerical data and unprocessed blots are available in source data.

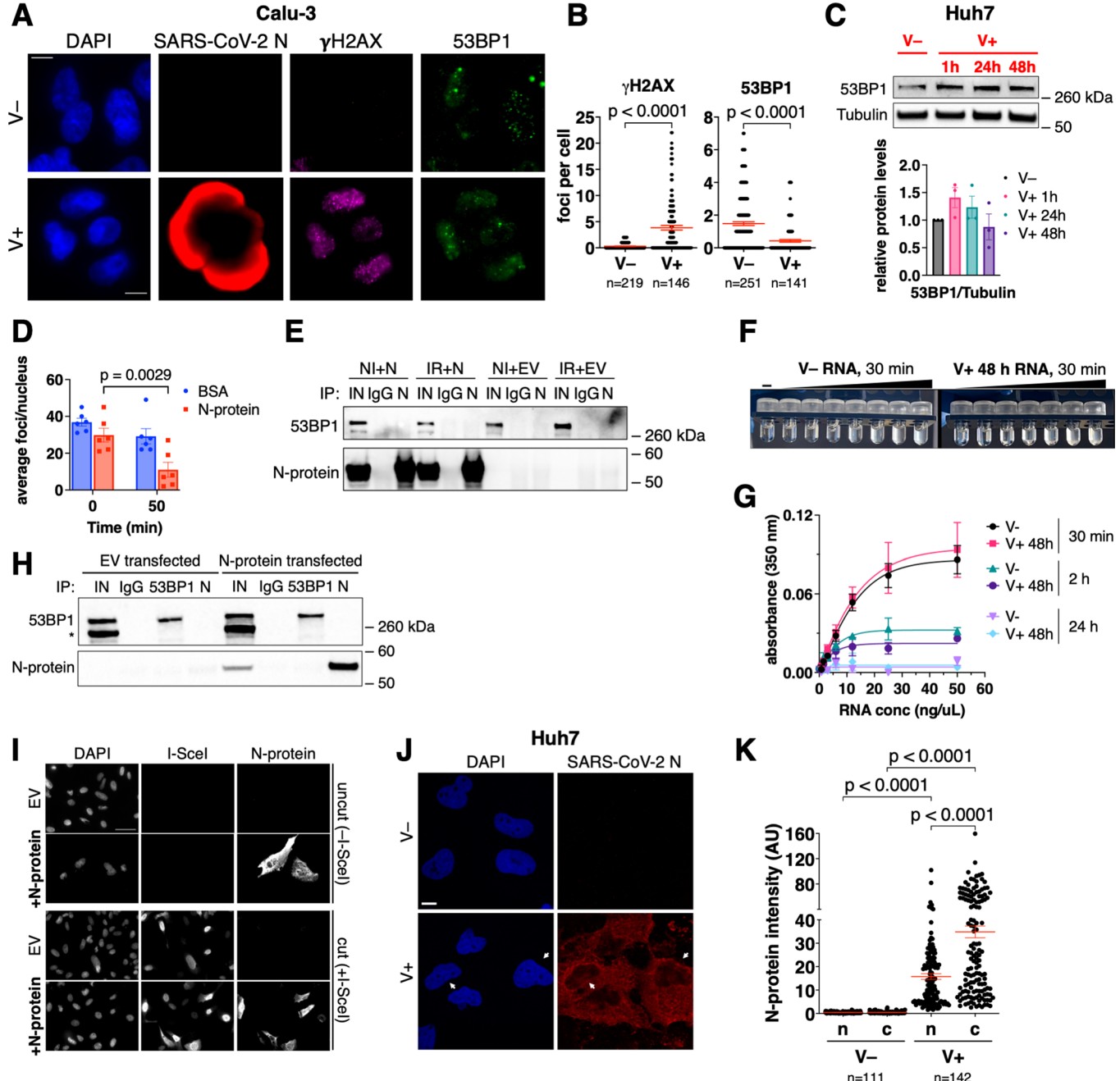

**Extended Data Fig. 6 | SARS-CoV-2 N-protein impairs 53BP1 foci formation in a RNA-dependent manner. A**) IF images of Calu-3 cells fixed 24 h post-infection; nuclei were stained with DAPI. Scale bar, 10 μm. **B**) Quantification of DDR foci shown in A; the dot-plots show the number of γH2AX or 53BP1 foci per nucleus; horizontal bars are the means ± s.e.m. of four independent infections. **C**) Immunoblot of infected Huh7 shown in Fig. 1A; tubulin was used as loading control. The histograms show the quantification of total 53BP1 levels; values are shown as relative to mock-infected samples. **D**) Quantification of 53BP1 foci per nucleus in irradiated 53BP1-GFP after micro-injection of purified recombinant N-protein or BSA as control (*n* = 6). **E**) Protein lysates of Huh7 cells transfected with N-protein or EV, irradiated (IR) or not (NI), were incubated with an anti-N-protein antibody (IP: N) or with normal rabbit IgGs (IgG) and analyzed by immunoblotting for the presence of 53BP1. The experiment was repeated three times with similar results. **F**) Representative pictures of turbidity assays. **G**) Analysis of turbidity represented in **F**; *n* = 4 independent experiments. **H**) Immunoprecipitation efficiency of the RIP experiments shown in Fig. 6H,I. Asterisk marks unspecific signal. The experiment was repeated three times with similar results. **I**) IF images of samples described in Fig. 6J. The experiment was repeated three times with similar results. Scale bar, 50 μm. **J**) IF images of Huh7 fixed 48 h post-infection and stained for N-protein. Arrows indicate cells with nuclear N-protein signal. Scale bar, 10 μm. **K**) Quantification of N-protein levels in the nucleus (n) and cytoplasm (c) of the samples described in J. Horizontal bars represent the means ± s.e.m. of a representative experiment of infection. Source numerical data and unprocessed blots are available in source data.

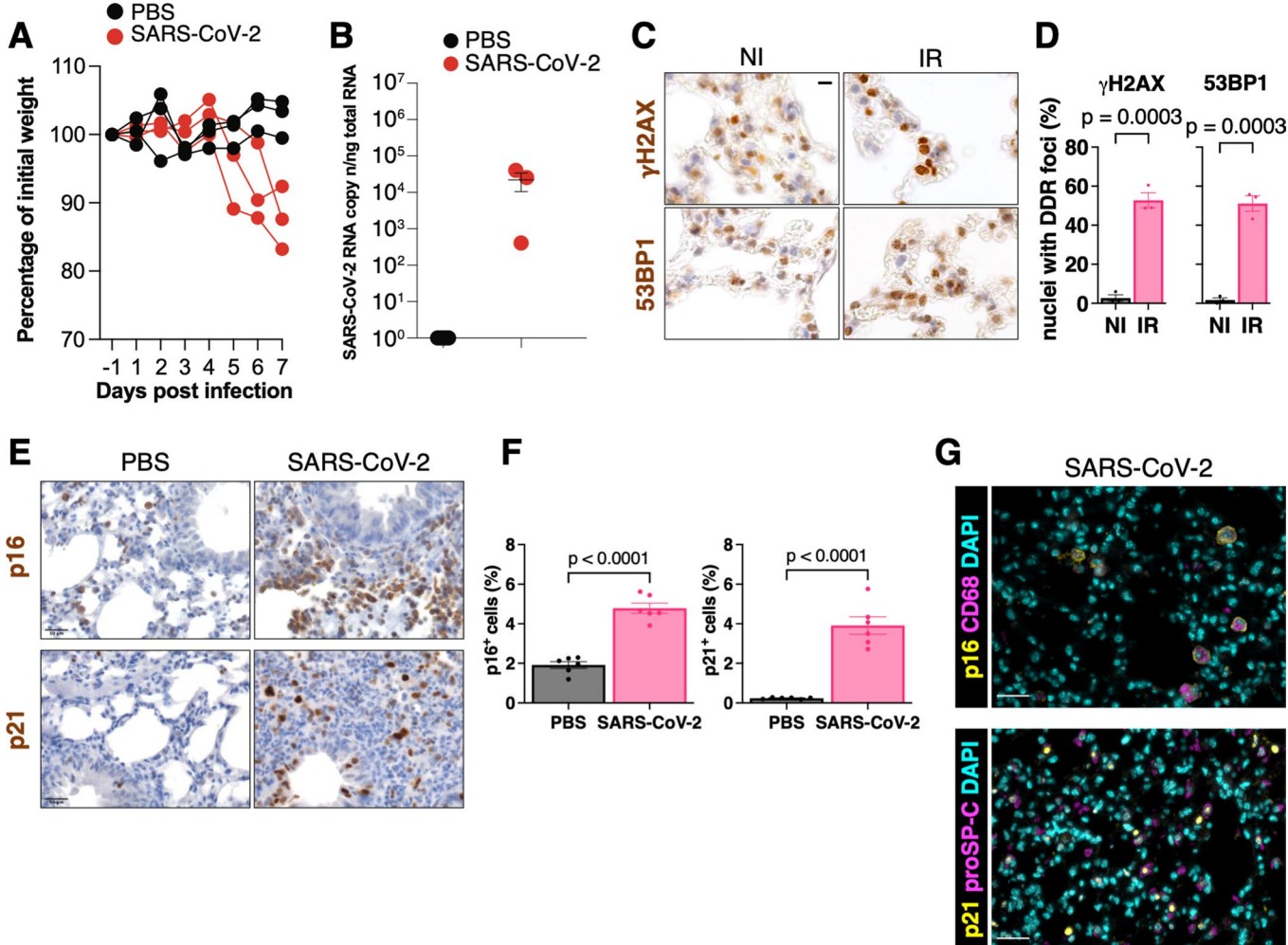

**Extended Data Fig. 7 | SARS-CoV-2 infection induces senescence in *hACE2*-mouse lungs. A**) Mouse body weight was monitored daily for 7 days and expressed as the percentage relative to day -1. Single values of PBS-control mice (black dots) and infected mice (red dots) are represented. **B**) Quantification of SARS-CoV-2 RNA in the lung of PBS-treated control (*n* = 3) and SARS-CoV-2 infected mice (*n* = 3) measured at 6 days post-infection. Values are expressed as copy number per ng of total RNA. **C**) IHC images of sections (*n* = 3) of the lungs from irradiated (IR, *n* = 1) or not irradiated (NI, *n* = 1) wild-type mice analyzed as controls of DDR activation. Scale bar, 20 μm. **D**) Quantification of DDR activation represented in C, each dot corresponds to a section. **E**) IHC images of lungs from SARS-CoV-2 infected *hACE2*-mice or mock-infected (PBS) wild-type mice probed for p16 or p21. Scale bar, 50 μm. **F**) Quantification of the percentage of cells positive for the markers shown in **E**, each dot corresponds to a mouse. **G**) IF images of a representative experiment of lungs, from SARS-CoV-2 infected *hACE2*-mice probed for the indicated proteins. CD68 and prosurfactant protein C (proSP-C) markers were used to label macrophages and pulmonary alveolar type II cells, respectively. Scale bar, 50 μm. The experiment was repeated three times with similar results. Source numerical data are available in source data.

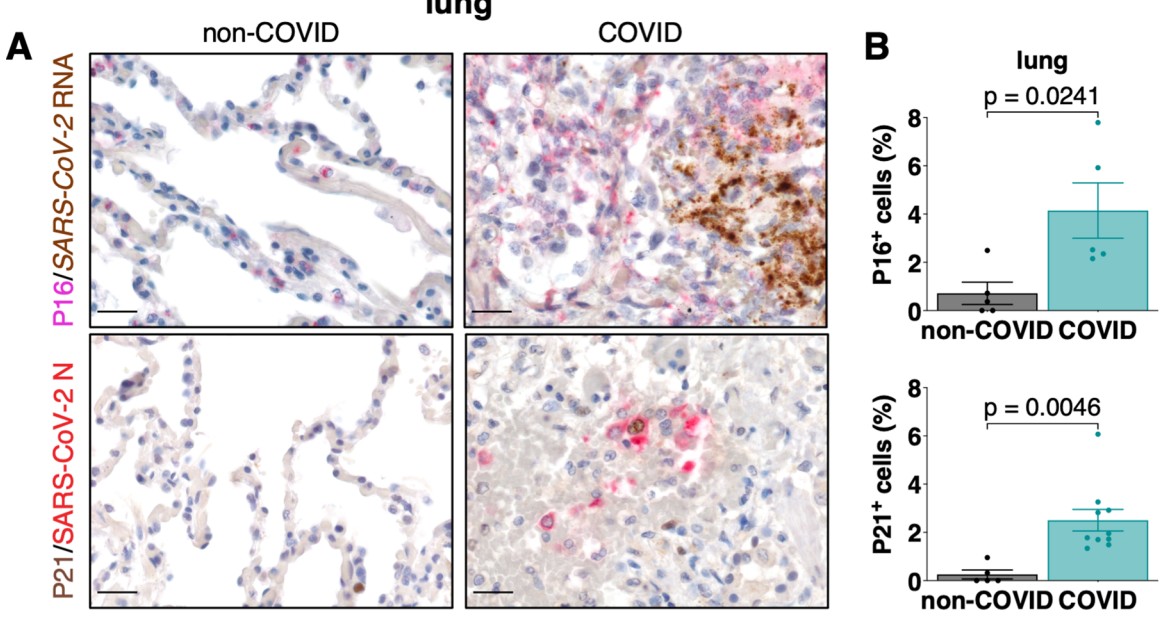

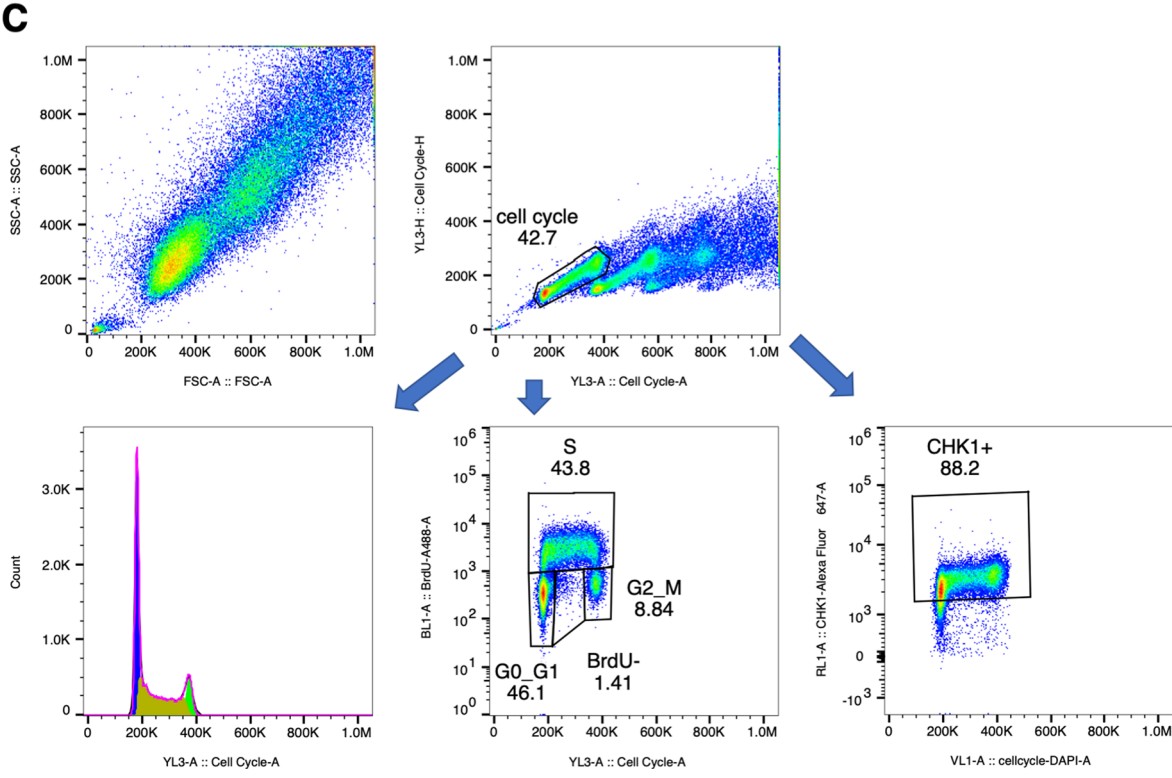

**Extended Data Fig. 8 | SARS-CoV-2 infection induces senescence in COVID-19 patients' lungs. A)** Representative IHC images of lung tissues stained for the senescence markers P16 (*n* = 5) or P21 (non-COVID individuals: *n* = 5; COVID-19 patients: *n* = 10); the presence of the virus was assessed by staining for SARS-CoV-2 N or by probing the viral genome through RNA ISH; nuclei were stained with hematoxylin (light blue); scale bar, 50 μm. Conditions are as in Fig. 8A. **B)** Quantification of the expression of the senescence markers determined in **A**. Values are the means ± s.e.m.; each dot corresponds to one individual. Source numerical data are available in source data. **C)** Gating strategy for flow cytometry.

# Reporting Summary

## Statistics

For all statistical analyses, confirm that the following items are present in the figure legend, table legend, main text, or Methods section.

| n/a | Confirmed | |
|---|---|---|
| ☐ | ☒ | The exact sample size (*n*) for each experimental group/condition, given as a discrete number and unit of measurement |
| ☐ | ☒ | A statement on whether measurements were taken from distinct samples or whether the same sample was measured repeatedly |
| ☐ | ☒ | The statistical test(s) used AND whether they are one- or two-sided *Only common tests should be described solely by name; describe more complex techniques in the Methods section.* |
| ☐ | ☒ | A description of all covariates tested |
| ☐ | ☒ | A description of any assumptions or corrections, such as tests of normality and adjustment for multiple comparisons |
| ☐ | ☒ | A full description of the statistical parameters including central tendency (e.g. means) or other basic estimates (e.g. regression coefficient) AND variation (e.g. standard deviation) or associated estimates of uncertainty (e.g. confidence intervals) |
| ☐ | ☒ | For null hypothesis testing, the test statistic (e.g. *F*, *t*, *r*) with confidence intervals, effect sizes, degrees of freedom and *P* value noted *Give P values as exact values whenever suitable.* |
| ☒ | ☐ | For Bayesian analysis, information on the choice of priors and Markov chain Monte Carlo settings |
| ☒ | ☐ | For hierarchical and complex designs, identification of the appropriate level for tests and full reporting of outcomes |
| ☒ | ☐ | Estimates of effect sizes (e.g. Cohen's *d*, Pearson's *r*), indicating how they were calculated |

*Our web collection on statistics for biologists contains articles on many of the points above.*

## Software and code

Policy information about availability of computer code

Data collection: Bio-Rad Image Lab 6.1 was used for immunoblot data collection; images at widefield microscope were acquired with MetaVue software or with Zen 2.0 Software (Zeiss); images at confocal microscope were collected with Leica Application Suite X; images in micro-injection experiments were acquired with Volocity 6.4.0; BioPlex 200 system (Bio-Rad), powered by the Luminex xMAP technology was used for multiplex immunoassays; for flow cytometry studies, see the related section.

Data analysis: Bio-Rad Image Lab 6.1 was used for densitometric analysis in immunoblot experiments; CellProfiler 3.1.9 was used to measure DDR activation in immunofluorescence analyses and in micro-injection experiments; tail moment in comet assays was measured using CometScore 2.0; ImageJ 1.53a was used to quantify protein sub-cellular distribution in immunofluorescence experiments; GraphPad Prism 9.3.0 was used for statistics; quantitative analyses of IHC experiments were performed using the Nuclear v9 or Positive Pixel Count v9 ImageScope software (v12.3.2.8013, Leica Biosystems); combined ISH/IHC images were analyzed with HALO v3.5.3577.140 (Indica Labs); for flow cytometry studies, we used FlowJo 10.7.1 (BD Biosciences).

For manuscripts utilizing custom algorithms or software that are central to the research but not yet described in published literature, software must be made available to editors and reviewers. We strongly encourage code deposition in a community repository (e.g. GitHub). See the Nature Portfolio guidelines for submitting code & software for further information.

## Data

Policy information about **availability of data**

All manuscripts must include a **data availability statement**. This statement should provide the following information, where applicable:
- Accession codes, unique identifiers, or web links for publicly available datasets
- A description of any restrictions on data availability
- For clinical datasets or third party data, please ensure that the statement adheres to our **policy**

> No datasets were generated or analysed during the current study. All raw data associated to Figures in manuscript are provided as supplementary material or available upon reasonable request.

## Human research participants

Policy information about **studies involving human research participants and Sex and Gender in Research.**

| | |
|---|---|
| Reporting on sex and gender | Sex and gender analysis was not necessary for this study |
| Population characteristics | We analysed lung parenchyma from 17 COVID-19 patients and nasal mucosa from 18 COVID-19 patients. Detailed information on patient's characteristics and treatment are provided in (Bussani et al., 10.1016/j.ebiom.2020.103104). The same number of lung samples was analysed from non-COVID patients, who died with viral pneumonia of different etiologies. It was not possible to analyze COVID-19-negative mucosae, as the harvesting procedure of the nasal mucosa is highly invasive and it destroys the appearance of the face; thus it was justified only during the pandemic in COVID-19 patients. |
| Recruitment | Patients were not recruited specifically for this study. All samples were previously collected from autopsy cases of COVID-19 and non-COVID patients, and then grouped according to their positivity for SARS-CoV-2. |
| Ethics oversight | This study was approved by the competent Joint Ethics Committee of the Regione Friuli Venezia Giulia, Italy. All patients provided their written informed consent to the use of their samples for research purposes at the time of hospital admission. |

Note that full information on the approval of the study protocol must also be provided in the manuscript.

# Field-specific reporting

Please select the one below that is the best fit for your research. If you are not sure, read the appropriate sections before making your selection.

☒ Life sciences ☐ Behavioural & social sciences ☐ Ecological, evolutionary & environmental sciences

For a reference copy of the document with all sections, see nature.com/documents/nr-reporting-summary-flat.pdf

# Life sciences study design

All studies must disclose on these points even when the disclosure is negative.

| | |
|---|---|
| Sample size | We did not use any criteria to determine the sample size. As much data as possible was collected depending on the nature of the experiments or in order to have statistical analysis |
| Data exclusions | Throughout the manuscript no data was excluded. Only in rare occasions, individual values were removed following unbiased criteria of outlier identification using Prism 9 software. In addition, in the case of IHC staining in infected mouse lungs, we excluded the tissue sections that show low rates of SARS-CoV-2 infection. |
| Replication | For all the experiments at least 3 independent replicates were performed unless differently stated in the figure legends. |
| Randomization | For in vivo experiments in mice, animals were randomized to the experimental groups. For in vitro experiments, wells were randomly assigned into each group and all cells were analysed equally. |
| Blinding | No blinding method was applied, as we used unbiased software for data collection and analysis. |

# Reporting for specific materials, systems and methods

We require information from authors about some types of materials, experimental systems and methods used in many studies. Here, indicate whether each material, system or method listed is relevant to your study. If you are not sure if a list item applies to your research, read the appropriate section before selecting a response.

## Materials & experimental systems

| n/a | Involved in the study |
|---|---|
| ☐ | ☒ Antibodies |
| ☐ | ☒ Eukaryotic cell lines |
| ☒ | ☐ Palaeontology and archaeology |
| ☐ | ☒ Animals and other organisms |
| ☒ | ☐ Clinical data |
| ☒ | ☐ Dual use research of concern |

## Methods

| n/a | Involved in the study |
|---|---|
| ☒ | ☐ ChIP-seq |
| ☐ | ☒ Flow cytometry |
| ☒ | ☐ MRI-based neuroimaging |

## Antibodies

Antibodies used

gH2AX (Ser139) Abcam ab11174
gH2AX (Ser139) Millipore 05-636
53BP1 Bethyl A303-906A
53BP1 Novus NB100-304
ACE2 Abcam ab15348
ATM Abcam ab32420
ATR Santa Cruz sc-1887
Beclin 1 Bethyl A302-566A-T
Beta-actin Sigma-Aldrich A2228
BrdU BDbioscience 347580
CD68 Abcam ab125212
CDT1 Cell Signaling #8064
cGAS Cell Signaling #15102
CHK1 Novus NB100-46
CHK1 (2G1D5) Cell Signaling #2360
CHK1 (ST57-09) ThermoFisher MA532180
CHK2 Millipore 05-649
Cleaved Caspase-3 (Asp 175) Cell Signaling 9661
DNA-PK Abcam ab32566
HA-tag Abcam ab236632
Histone H3 Abcam ab10799
Human Cytokeratin 8/18 (EP17/EP30) Dako M3652
ISceI (FL-86) Santa Cruz sc-98269
KAP1 Abcam ab10484
LC3B Sigma-Aldrich L7543
p16 Abcam ab51243
p21 Abcam ab188224
P21 Cell Signaling #2946
P38 MAPK Cell Signaling #9212
P53 Abcam ab1101
p62 Abcam ab240635
pATM (Ser1981) Rockland 200-301-400
pATR (T1989) Abcam ab223258
pCHK1 (S317) Cell Signaling #2344
pCHK2 (Thr68) Cell Signaling #2661
PCNA Bio Rad MCA1558
pDNA-PK (Ser2056) (EPR5670) Abcam ab124918
pKAP1 (S824) Bethyl A300-767A
pP38 MAPK (Thr180/Tyr182) Cell Signaling #9211
pP53 (Ser15) Cell Signaling #9284
proSP-C Abcam ab3786
pRPA (S4/S8) Bethyl A300-245A
pSTAT1 (Ser 727) Cell Signaling #9177
pSTAT1 (Tyr 701) (58D6) Cell Signaling #9167
RPA Calbiochem NA18-100UG
RRM1 Santa Cruz sc-11733
RRM2 Novus NBP1-31661
RRM2 Santa Cruz sc-10844
SARS-CoV2 nucleocapsid Sino Biological 40588-T62
SARS-CoV2 nucleocapsid Sino Biological 40143-R019
SARS-CoV2 nucleocapsid (1A6) ThermoFisher MA5-35941
STAT1 (9H2) Cell Signaling #9176
Strep-tag II epitope Qiagen 34850
Tubulin Sigma-Aldrich T5168
Ubiquitin (P4D1) Santa Cruz sc-8017
Vinculin Sigma-Aldrich V9131
Cy3 D/M Jackson 715-165-150
Cy3 D/R Jackson 711-165-152
Cy3 D/G Jackson 705-165-147

A488 D/M ThermoFisher A21202
A488 D/R ThermoFisher A21206
A488 D/G ThermoFisher A11055
A647 D/M ThermoFisher A31571
A647 D/R ThermoFisher A31573
A647 D/G ThermoFisher A21447

| Validation | All antibodies were validated by the manufacturer and were previously used in peer reviewed works. Methods of validation and references to published application for all antibodies are all present into manufacturer dedicated website page of each indicated product. |
|---|---|

## Eukaryotic cell lines

Policy information about cell lines and Sex and Gender in Research

| Cell line source(s) | Vero E6 cells (ATCC-1586); human hepatocarcinoma Huh7 cells (JCRB0403, JCRB cell bank of Okayama University) were kindly provided by Ralf Bartenschlager, University of Heidelberg, Germany; lung adenocarcinoma Calu-3 (ATCC HTB-55); U2OS 53BP1-GFP (Bekker-Jensen et al. 2005); U2OS EJ5-GFP (Gunn & Stark 2012); NIH2/4 (Soutoglou et al. 2007) |
|---|---|
| Authentication | Cell lines were authenticated by STR profiling (GenePrint system, Promega) |
| Mycoplasma contamination | All cell lines were tested negative for mycoplasma |
| Commonly misidentified lines (See ICLAC register) | No commonly misidentified lines were used |

## Animals and other research organisms

Policy information about studies involving animals; ARRIVE guidelines recommended for reporting animal research, and Sex and Gender in Research

| Laboratory animals | Ten B6.Cg-Tg(K18-ACE2)2Prlmn/J mice (M. musculus; two 13-week-old females; two 11-week-old males, four 10-week-old males, one 8.5-week-old male and one 8.5-week-old female); four C57BL/6 J mice (M. musculus; two 11-month-old males and two 10-week-old males). |
|---|---|
| Wild animals | This study did not involve wild animals. |
| Reporting on sex | Sex analysis was not necessary for this study. |
| Field-collected samples | This study did not involve samples collected from the field. |
| Ethics oversight | Experiments involving animals have been carried out in accordance with the Italian Laws (D.lgs. 26/2014), which enforce Directive 2010/63/EU (Directive 2010/63/EU of the European Parliament and of the Council of 22 September 2010 on the protection of animals used for scientific purposes). Accordingly, the project has been authorized by the Italian Competent Authority (Ministry of Health). |

Note that full information on the approval of the study protocol must also be provided in the manuscript.

## Flow Cytometry

### Plots

Confirm that:

☒ The axis labels state the marker and fluorochrome used (e.g. CD4-FITC).

☒ The axis scales are clearly visible. Include numbers along axes only for bottom left plot of group (a 'group' is an analysis of identical markers).

☒ All plots are contour plots with outliers or pseudocolor plots.

☒ A numerical value for number of cells or percentage (with statistics) is provided.

### Methodology

| Sample preparation | Experiments were carried out in cultured cells fixed first in formaldehyde 2% and then in 75% ethanol |
|---|---|
| Instrument | Samples were acquired with Attune NxT (Thermofisher) |
| Software | Analysis was carried out using FlowJo 10.7.1 (BD Biosciences) |
| Cell population abundance | 30,000 events were analyzed for each sample in each individual experiment. |
| Gating strategy | Cell doublets were removed and living cells selection was based on forward and side scatter. Single cells were gated based on |

Gating strategy | their SSC-A vs. FSC-A and SSC-A vs. SSC-H parameters. 561 nm laser and 695/40 filter were used for propidium iodide detection; 488 nm laser and 530/30 filter were used for BrdU and CHK1 detection.

☒ Tick this box to confirm that a figure exemplifying the gating strategy is provided in the Supplementary Information.

