## [Peer Review File · Nature Cell Biology]

Peer Review Information

Journal: Nature Cell Biology

Manuscript Title: SARS-CoV-2 infection causes DNA damage through multiple mechanisms

Corresponding author name(s): Dr Fabrizio d'Adda di Fagagna

Editorial Notes:

-
-
-
-

Reviewer Comments & Decisions:

Decision Letter, initial version:
--

Dear Dr d'Adda di Fagagna,

Your manuscript "SARS-CoV-2 infection causes DNA damage and an altered DNA damage response", has now been seen by 3 referees, who are experts in viral infection and immunity (referee 1); senescence and DNA repair (referee 2); and DNA repair (referee 3), and whose comments are pasted below. In light of their advice, we regret that we cannot offer to publish the study in Nature Cell Biology.

As you will see, although the reviewers find this work interesting, they raise serious concerns that question the strength of the data and of the novel conclusions that can be drawn at this stage.

We are very sorry that we could not be more positive on this occasion, but we thank you for the opportunity to consider this work.

With kind regards,
Jie

Jie Wang, PhD
Senior Editor
Nature Cell Biology

email: jie.wang@nature.com

Reviewers' comments:

Reviewer #1 (Remarks to the Author):

In this study, the authors described a non-canonical activation of DNA damage response caused by SARS-CoV-2 infection which led to DNA damage. They observed the reduction of CHK1 and RRM2 caused by SARS-CoV-2 infection. These impacts caused the shortage of dNTPs and the accumulation of DNA damage. In addition, the authors reported an important role of SARS-CoV-2 N-protein, which can reduce DNA repair by non-homologous end joining by competing with 53BP1 for the association with damage-induced long non-coding RNAs. However, there are obvious shortcomings in current version that need to be improved. Particularly considering the role of CHK1-RRM2 on DNA damage has been reported, the authors need to explore the underlying mechanism for the reduction of CHK1 caused by SARS-COV-2. In addition, more validations are required by using either mouse and human samples.

Major points:

1. In figure 1A and 1B, SARS-CoV-2 infection triggered the autophosphorylation, and thus activation, of the master kinases DNAPK (pDNA-PKS2056 – involved in DNA repair) and ATM (pATMS1981), why the phosphorylation CHK2, the direct downstream target of ATM kinase activity, is not affected?
2. Since SARS-CoV-2 caused the robust phosphorylation of H2AX (H2AX) and RPA (pRPAS4/8), markers of DSB and SSB, respectively (Fig. 1A, B), were RAD51 foci and RAD51 protein level also investigated? The author only examine the H2AX phenotype in the Fig 4-6, was the phosphorylation of RPA (pRPAS4/8) also affected in the same assay in Fig 4-6?
3. Since the Huh7 cell line does not express cGAS or STING, could SARS-CoV-2 induce the transcription of pro-inflammatory response genes in Hun7 cells? Whether CHK1 overexpression in Hun7 cells can rescue the SARS-CoV-2-induced the pro-inflammatory cytokines expression, RRM2 reduction and impaired S-phase progression? It would be an added value to the manuscript.
4. In Figure 2J, add the immunofluorescence panel like Figure 2H which would be a powerful experiment to show that cells expressing ORF6 and NSP13 reduced RRM2 levels and caused increased

H2AX and RPA phosphorylation.

5. In Figure S2G, please show the expression of the indicated SARS-CoV-2 proteins, as well as RRM2 proteins, by western blot.

6. In Figure 2J, how the SARS-CoV-2 gene products ORF6 and NSP13 reduce the CHK1 protein levels? Can the authors provide mechanistic explanation?

7. It was known that, upon DNA damage, Chk1 activation up-regulates RRM2 expression through the E2F1 transcription factor. CHEK1 down-regulation by siRNA and small molecule inhibitors of Chk1 blocked RRM2 induction by CPT (J Biol Chem. 2009. PMID: 19416980). Another study also provided evidence for RNR2 regulation by Chk1 for S-phase progression as the author has cited (Biochem. Biophys. Res. Commun. 2008). Therefore, the figure 3A-D is in lack of innovation and therefore they should not be placed in the main figures.

8. The figure 3F, Please add ELISA experiments for those cytokines such as IL1b, IL6, CXCL9, CXCL10, TNFa. What is the major mechanism by which CHK1 loss leads to increased mRNA expression of inflammatory factors?

9. SARS-CoV-2 N protein should undergo Liquid-liquid phase separation and perform high efficient condensation in cytoplasm in cells, why the N proteins in figure 4A, 4C were largely diffused in cytoplasm? In figure 4E, why the SARS-CoV-2 N proteins mainly located in the nucleus, quite different from figure 4A, 4C? Please add positive control experiments: immunofluorescence experiments of 53BP1 and γ H2AX foci after DNA damage.

10. In the article, authors observed that SARS-CoV-2 N-protein can reduce 53BP1 foci (Fig 4A-D). However, in experiment using Calu-3 cells and in vivo experiment, SARS-CoV-2 infection cannot cause the reduction of 53BP1 foci (Fig S4A-B and Fig 5). This conclusion is inconsistent with their previous results (Fig 4A-D).

11. Was the SARS-CoV-2 infection-induced damage repair primarily accomplished by NHEJ rather than homologous recombination? Have the author examine the HR efficiency in cells expressing SARS-CoV-2 N-protein using HR repair assay?

12. Please verify the activation of CHK1 caused by SARS-CoV-2 infection in mice model by immunohistochemistry or immunofluorescence in figure 5A.

13. In figure 5A, an experiment using SARS-CoV-2 infected hACE2-mouse with irradiated (IR) or not irradiated (NI) treatment to demonstrate, consistent with their observations in cultured cells (Fig. 4A-D), that SARS-CoV-2 rescues suppresses 53BP1 activation would be an added value to the manuscript.

14. The authors suggest that SARS-CoV-2 can reduce 53BP1 foci. Is 53BP1 foci disrupted in lung and nasal mucosal tissues of COVID-19 patients, compared to patients that stained negative for the presence of SARS-CoV-2, or to subjects not diagnosed for the pathology (non-COVID)?

15. The authors identify two candidate mechanisms whereby SARS-CoV-2 infection causes DNA damage and an altered DNA damage response: (i) through the degradation of the DDR kinase CHK1

and the loss of RRM2, impaired S-phase progression, DNA damage accumulation, induction of inflammatory cytokines and (ii) impairs 53BP1 recruitment at DSB by competing with diIncRNAs binding, reducing DNA repair by NHEJ. Are there any connections between those two mechanisms? Will there be a compensatory effect?

16. Are there more important upstream proteins in the DNA damage pathway that are affected? Was the type of DNA damage caused by SARS-CoV-2 mainly double-strand breaks?

Minor points:

1. Add the RT-qPCR analysis of the mRNA levels of pro-inflammatory cytokines (IL1b, IL6, CXCL9, CXCL10 and TNF) in CHK1-depleted Hun7 cells.

2. In Fig. S4F, please show the photos corresponding to the turbidity assays.

3. Please specify the n value for the figure 3F.

4. In figure 6A and 6SA, please show the representative images of lung and nasal mucosal tissues from patients that stained negative for the presence of SARS-CoV-2, or to subjects not diagnosed for the pathology (non-COVID).

5. The representative images of lung and nasal mucosal tissues in figure 6C and 6E are quite different from those in figure S6A-B. Please provide an explanation.

6. In figure 6C and 6E, it is better to show the representative images (γ H2AX and RRM2) of lung and nasal mucosal tissues from patients that stained negative for the presence of SARS-CoV-2, or to subjects not diagnosed for the pathology (non-COVID).

Reviewer #2 (Remarks to the Author):

In the present manuscript Gioia and collaborators explore the ability of SARS-CoV-2 to induce DNA damage upon infection. They suggest that the observed DNA damage is induced by a non canonical DNA damage response pathway based on interrogation of human cell lines infections (Huh-7 and Calu-3), mice infections and post mortem human sample from COVID-19 patients. Actually, they suggest a double route that generates genomic instability. The first concerns CHK1-mediated down-regulation of RRM2 that in turns leads to dNTPs shortage followed by replication stress, DNA damage and inflammatory cytokines production. The second, describes a competitive action of SARS-CoV2 N protein over 53BP1 by binding DNA damage induced long non coding RNAs (diIncRNAs - a previous discovery from the group) leading to impairment of the NHEJ repair process. Although this is a very interesting story with great potential in the field of molecular virology, many findings are not clear while mechanistic aspects remain unaddressed or confusing. These issues should be addressed in order to achieve publication level.

Major comments

1. It is not clear to me why two cancer cell lines (Huh-7 and Calu-3) are the main cellular settings employed. As cancer cells, their genetic constitution is severely impaired (see following notes), eg both cell lines harbor mutant p53, and thus many homeostatic circuitry are altered and thus their

function/reaction upon viral infection may not reflect that of the COVID-19 patients. The authors make a very limited use of primary nasal epithelial cells, which are a more ideal cellular model. I would favor the use of cellular system(s) that are normal and do not carry the mutational burden of full blown tumor cells that can severely skew the findings.

Also, additional time points beyond 48h should be included in the analysis, given that SARS-CoV-2 infection can persist more than 48h. In this way a broader picture will be gained on the status of DDR components, as some exhibit a weird pattern (Figure 1A). For example: i) why p-CHK1 is increased at 1 and 24 hrs and then decreases at 48h (including total levels), ii) why total ATR levels (and consequently basal p-ATR) gradually increase over the time spectrum analyzed (compared against H3 levels used as control loading), given that DDR apical kinases (ATM, ATR, DNA-PK) are activated mainly at post-transcriptional level? Similar pattern appears to be for ATR in hydroxyurea treatment.

Notes: HUH7 cells harbor

i) 108 genes with mutations

https://maayanlab.cloud/Harmonizome/gene_set/HUH7/CCL+Cell+Line+Gene+Mutation+Profiles

ii) 1079 genes with low or high copy number

CALU3 cells harbor

i) 79 genes with mutations

https://maayanlab.cloud/Harmonizome/gene_set/CALU3/CCL+Cell+Line+Gene+Mutation+Profiles

ii) 1013 genes with low or high copy number

https://maayanlab.cloud/Harmonizome/gene_set/CALU3/CCL+Cell+Line+Gene+CNV+Profiles

2. The authors should provide evidence on the molecular mechanism(s) responsible for the CHK1 decreased levels during SARS-CoV-2 infection by ORF6 and NSP13. The immunofluorescent and immunoblot analyses in Figure 2H,J are just correlative while Figure S2C supports a (post-)translational impact with no other evidence on how ORF6 and NSP13 decrease CHK1 protein levels. Do ORF6 and NSP13 affect directly CHK1, or other mediator(s) intervene? Previous studies, quoted also in the manuscript (refs 21 and 28), have described the axis ATR/ATM-CHK1-E2F1-RRM2. This molecular pathway should also be interrogated if it is indeed the one responsible in the examined settings. If ORF6 and NSP13 directly suppress CHK1, in which subcellular compartment does this take place, given that the virus replicates in the cytoplasm? Moreover, in an actual infection all the viral proteins, that the authors examine separately applying expression vectors, are expressed simultaneously. Thus an important complementary/rescue experiment that would simulate what takes place in vivo would be to infect cells with the virus and subsequently silence the particular protein (ORF6 or NSP13) and examine if the outcome copes with the results of the transfection assays.

3. Authors showed that reduction of dNTPs levels leads to DNA replication defects. Rescue experiments in the form of supplementing dNTPs and CHK1 reconstitution should be performed to confirm their hypothesis. Moreover, rescue experiments should be accompanied by additional experimental verifications like assessing DDR markers expression, comet assay, production of inflammatory cytokines.

4. In Huh-7 cells, despite the activation of the DNA damage response pathway (Figure 1A), there is no evidence of apoptosis induction (Figure S1A). Given that markers indicative of senescence are observed in infected mice, the authors should examine for the presence of this phenomenon in the employed cell lines and in the sections from the animal and human tissues (see point 9 below) using the recently recommended guidelines (Nat Protoc PMID 33911261).

5. The authors explore cytokine production only in Calu-3, based on the rationale that only these cells express sGAS/STING, while Huh-7 do not. Considering the previous point, the authors are advised to examine for cytokine production in Huh-7 cells also, particularly if these cells enter senescence upon

virus infection.

6. The authors should check in the CHK1 knocked-down cells (Figure 3) the DNA integrity and the expression of cGAS/STING for its role in the inflammatory response.
7. The authors support that the N protein interferes with 53BP1 and impairs DNA repair by NHEJ. What is the alternative mechanism(s) by which cells repair the DNA damage, since no apoptosis is observed? Given that SARS-CoV-2 replicates in the cytoplasm, while 53BP1 is a nuclear localized protein, does N protein enter the nucleus of the cells upon SARS-CoV-2 infection? Analysis of cytoplasmic and nuclear fractions, further to total lysates, could help define if nuclear localization of N protein takes place. In such a case what is the mechanism of nuclear translocation? Moreover, rescue experiments consisting of viral infection followed by N-protein silencing should be also performed, as described in point 2. Finally, it would be very solid for the manuscript to find in vivo evidence (eg in tissue sections) of nuclear localization of N protein and possibly co-localization with diIncRNAs.
8. To support their hypothesis the authors investigate in vivo mouse models and human COVID-19 samples. SARS-CoV-2 presence should be confirmed by qRT-PCR and IHC/IF analysis in all materials from in vivo experiments. Especially in Figure 5 (A and B) it should be added an IHC figure for SARS-CoV-2 detection in a parallel serial section analysis for the detection of double positive cells (SARS-CoV2 and γ H2AX/53BP1) in the four different conditions (PBS/virus infection, NI, IR). Alternatively double or triple IF analysis would be also fine. Moreover, to support their hypothesis the authors should interrogate CHK1 and RRM2 expression in SARS-CoV-2 infected mouse tissues.
9. A consequence of SARS-CoV-2 infection is the induction of cellular senescence, as previously reported and quoted in the manuscript. According to a previously published multi-marker algorithm, SA-b-gal activity and/or lipofuscin accumulation (GL13 staining) and co-staining with other markers frequently observed (p16INK4A, p21WAF1/Cip1) or absent (proliferation markers, Ki67) in senescence should be applied in mouse and human tissue sections (Nat Protoc PMID 33911261). The multi-marker workflow should be confirmed by serial section analysis or with double/Triple staining by IHC or IF analysis in the four different conditions (PBS/virus infection, NI, IR).
10. Staining should be provided for ACE2 and markers, such as against type II pneumonocytes, to identify in vivo the type of cells that the virus infects as well as the presence of the virus, by co-staining or serial section analysis.
11. Many IF photos taken in the examined cellular systems are not convincing, particularly for SARS-CoV-2 proteins. Merging of IF photos should be done to better delineate spatial distribution of contents depicted. Also tissue IF photos in Figure 6C,E are similarly not so convincing. Magnification in Figure S6B is blurred.

Minor comments

1. What is the nature of the mock condition applied for SARS-CoV-2 infection in Figure 1A?
2. In Figure 1, CHK1 protein expression is examined at both total- and phospho-levels. In subsequent figures that depict CHK1 immunoblot analyses, this is not defined. In case total levels are estimated why did the authors choose to examine this form instead of the active, phosphorylated CHK1?
3. Figures S1 D,E,F could be included in main Figure 1.
4. Since the Huh7 cell line does not express cGAS or STING, the authors employ the Calu-3 cells in order to detect the damaged nuclear DNA in cytosol and the induction of pro-inflammatory response gene (Figure S1 D). Authors should carry out the comet assay also in the Calu-3 cell line since the DNA fragmentation, the cytosolic DNA and the inflammatory phenotype are detected in different cell lines.
5. Please quantify the results for IF experiments in Figure 2C.
6. Please add the name of the cell line at the top side of each image/graphs.
7. In Figure S4A is the IF for N protein staining specific? The staining pattern is very peculiar,

considering also the other markers assessed in this panel (DAPI, γ H2AX, 53BP1) and in comparison with panels of Figure 4.

8. Please provide an IHC Figure from mice lung for ACE2 immunostaining to confirm its expression.

9. Figure 7 should be updated to include the mechanism of inflammatory response.

10. What were the criteria that authors relied on to apply a 0,1 MOI for cellular infection?

11. Include the quantification method applied for the IHC analysis.

Reviewer #3 (Remarks to the Author):

This manuscript proposes that SARS-CoV-2 infection causes DNA damage and an altered DNA damage response (DDR).

The conclusion that the viral infection causes DNA damage is well documented. However, the claim that the DDR is altered is not supported well by the data and may not be true. Thus, the only conclusion that is well validated is that SARS-CoV-2 infection induces DNA damage, whereas all the other data are very hard to interpret.

Specific Comments

1. Fig. 1 shows that SARS-CoV-2 infection of human hepatocarcinoma Huh7 cells induces activation of DNA-PK and ATM, but not ATR. KAP1 a substrate of ATM was also phosphorylated after infection, but Chk2, also an ATM substrate, and p53 were not phosphorylated. A comet assay further revealed DNA fragmentation after infection. In a different carcinoma cell line (Calu-3), SARS-CoV-2 infection activated cGAS-STING, although these data probably need to be complemented by additional experiments and currently are only presented in the supplementary section. The authors conclude that viral infection activates the DDR pathway in a non-canonical manner. I agree with the conclusion that the DDR is activated, but not with the conclusion that the activation is non-canonical. The authors are basing this conclusion on one cancer cell line, whose DDR may be compromised simply because these are cancer cells. Most likely, the viral infection does not induce DNA DSBs directly, so the DDR after viral infection cannot be compared to the DDR after exposure to IR. Indeed, ATM phosphorylation after viral infection was much weaker than after IR (Fig. 1A) and surprisingly, Chk2 was not activated 60 minutes after IR, even though ATM was fully activated at that time point (Fig. 1A). The authors also did not examine Chk2 phosphorylation in Calu-3 cells, which they absolutely should have done, if they want to claim that the DDR after viral infection is non-canonical.

2. Fig. 2 shows that the protein levels of Chk1 and RRM2 decrease in infected cells 48 h after infection. It is hard to interpret these data. Are the cells undergoing apoptosis or senescence? dNTP levels also decreased. Changes like this are somewhat expected (especially the decrease in dNTP levels), because the virus is using the resources of the cells for its own proliferation. In Fig. 2H,I,J,K the authors attribute the decrease in Chk1 and RRM2 protein levels to expression of viral proteins ORF6 and NSP13. This conclusion was based on expressing individually each viral protein in Huh7 cells and monitoring Chk1 and RRM2 levels by immunoblotting. In Fig. 2K left panel, the magnitude of reduction of RRM2 levels after ORF6 and NSP13 expression is not great. Also the data in Fig. S2G hide the variability in Chk1 levels, because the part of the graph between values 1.0 and 2.0 is drawn at a smaller scale than the part of the graph between values 0.0 and 1.0. Also there are no standard

deviations shown. I presume that the authors want to conclude that the viral infection decreases Chk1 and RRM2 levels due to expression of ORF6 and NSP13, but the results do not support this conclusion. Also no mechanism is provided to explain how ORF6 and NSP13 might reduce Chk1 and RRM2 levels.

3. Fig. 3 shows that depleting Chk1 induces cell cycle arrest in S phase, DNA damage and moderately increased expression of certain interleukins. These experiments are meaningful, if we accept the premise of the authors that the viral proteins ORF6 and NSP13 suppress Chk1 expression and that the phenotypes observed after viral infection are only due to the decrease in Chk1 expression. I have a hard time believing this, because a virus replicating in cells will likely have many effects on the cell.

4. Fig. 4 shows that 53BP1 does not localize with gH2AX foci in virus infected cells. Further, the authors showed that the SARS-CoV-2 N protein associates with dilncRNAs, preventing 53BP1 from interacting with these RNAs and therefore preventing 53BP1 focus formation. Expression of the N protein also suppressed NHEJ, as monitored in EJ5-GFP reporter cells. However, the magnitude of the effect is very small (Fig 4J) and a control reporter cell line (eg DR-GFP) was not examined. Another control would be to examine MDC1 foci, as these, in contrast to 53BP1 foci, should not be affected by the N protein.

5. Fig. 5 examines gH2AX and 53BP1 foci in lung tissues of virus infected mice and mice exposed to IR. 53BP1 foci were not induced in the virus infected cells, whereas gH2AX foci were weakly induced. There is a problem again with the y axes, which makes it hard to compare the foci induced by viral infection vs IR. Nevertheless, this experiment shows in a physiological setting that viral infection induces DNA damage.

6. Fig. 6 shows gH2AX signal in postmortem lung biopsies of COVID-19 patients. p21 expression was also induced in these cells, while RRM2 levels were decreased. The authors attribute the decrease of RRM2 to the same mechanism as in cell lines; however, the decrease in RRM2 may be secondary to cell senescence and a reduced number of cells in S phase. Moreover, the magnitude of the decrease was very small (Fig. 6F).

**Although we cannot publish your paper, it may be appropriate for another journal in the Nature Portfolio. If you wish to explore the journals and transfer your manuscript please use our manuscript transfer portal. If you transfer to Nature journals or the Communications journals, you will not have to re-supply manuscript metadata and files. This link can only be used once and remains active until used.

All Nature Portfolio journals are editorially independent, and the decision on your manuscript will be taken by their editors. For more information, please see our manuscript transfer FAQ page.

Note that any decision to opt in to In Review at the original journal is not sent to the receiving journal on transfer. You can opt in to In Review at receiving journals that support this service by choosing to modify your manuscript on transfer. In Review is available for primary research manuscript types

only.

**For Nature Research general information and news for authors, see <http://npg.nature.com/authors>.

Author Rebuttal to Initial comments

We would like to thank the reviewers and the Editor for the constructive comments. Here below, we provide a point-by-point response to the concerns raised.

Reviewer #1 (Remarks to the Author):

In this study, the authors described a non-canonical activation of DNA damage response caused by SARS-CoV-2 infection which led to DNA damage. They observed the reduction of CHK1 and RRM2 caused by SARS-CoV-2 infection. These impacts caused the shortage of dNTPs and the accumulation of DNA damage. In addition, the authors reported an important role of SARS-CoV-2 N-protein, which can reduce DNA repair by non-homologous end joining by competing with 53BP1 for the association with damage-induced long non-coding RNAs. However, there are obvious shortcomings in current version that need to be improved. Particularly considering the role of CHK1-RRM2 on DNA damage has been reported, the authors need to explore the underlying mechanism for the reduction of CHK1 caused by SARS-COV-2. In addition, more validations are required by using either mouse and human samples.

Major points:

1. In figure 1A and 1B, SARS-CoV-2 infection triggered the autophosphorylation, and thus activation, of the master kinases DNAPK (pDNA-PKS2056 – involved in DNA repair) and ATM (pATMS1981), why the phosphorylation CHK2, the direct downstream target of ATM kinase activity, is not affected?

In Fig. 1A and S1A, we show that in cells exposed to IR CHK2 is promptly but transiently phosphorylated on threonine 68 – compare 15' post-IR and 60' post-IR. This is expected and in agreement with the published literature (e.g.: <https://pubmed.ncbi.nlm.nih.gov/12805407/>; <https://pubmed.ncbi.nlm.nih.gov/25404613/>).

Upon viral infection, at the 1-hour time point studied, no DDR marker is activated, indicating that no DNA damage has yet been generated, thus explaining also the lack of CHK2 phosphorylation. At the 24 and 48 h time points studied, several DDR markers are activated but CHK2 is not phosphorylated, consistent with its kinetic of phosphorylation also observed upon irradiation.

2. Since SARS-CoV-2 caused the robust phosphorylation of H2AX (H2AX) and RPA (pRPAS4/8), markers of DSB and SSB, respectively (Fig. 1A, B), were RAD51 foci and RAD51 protein level also investigated?

Following this reviewer's suggestion, we investigated the levels of RAD51 and its activation in foci. We observed that SARS-CoV-2 infection causes a reduction in RAD51 foci formation. When we probed this further, we observed also reduced RAD51 protein levels (see data below).

Although potentially interesting, upon editorial approval, we are prone to retain this observation for referees only and not include it in the present manuscript which already unveils two distinct mechanisms threatening genome stability: CHK1 downregulation and

53BP1 foci impairment, encompassing 9 main and 7 supplementary (quite crowded) figures.

The author only examine the H2AX phenotype in the Fig 4-6, was the phosphorylation of RPA (pRPAS4/8) also affected in the same assay in Fig 4-6?

As requested, we have now extended our analyses to pRPA^{S4/8} in HNEpC and *in vivo* in mouse and in COVID-19 patients' lungs, and we can confirm that SARS-CoV-2 infection causes the phosphorylation of RPA (Fig. S1C,D, 7 and 8E,F), thus strengthening our conclusions and extending them to physiological settings.

3. Since the Huh7 cell line does not express cGAS or STING, could SARS-CoV-2 induce the transcription of pro-inflammatory response genes in Hun7 cells?

To address this, we extended our RT-qPCR-based transcriptional analyses of pro-inflammatory cytokine and chemokine expression to infected Huh7 cells. Although we observed that also here cytokines and chemokines are induced, their induction was generally reduced and significantly impaired for some of them – see examples below and new Fig. S1J.

Stimulated by this reviewer, to probe the events underlying the cGAS/STING-independent cytokines induction observed in Huh7 cells, we analyzed the engagement of additional pathways commonly associated with cytokines and chemokines expression. Interestingly, we observed that both STAT1 and p38/MAPK pathways are activated upon SARS-CoV-2 infection (new Fig. S1I), thus providing a likely explanation for their expression.

Notably, the induction of pro-inflammatory genes is consistent with SARS-CoV-2-induced cellular senescence, observed also in Huh7 cells (new Fig. S1L,M) which is characterized by the activation of the so-called senescence-associated secretory phenotype (SASP).

Whether CHK1 overexpression in Huh7 cells can rescue the SARS-CoV-2-induced the pro-inflammatory cytokines expression, RRM2 reduction and impaired S-phase progression? It would be an added value to the manuscript.

We agree with this reviewer and we pursued this experiment with significant effort. However, we soon realized this is challenging and we share here our findings. We observed that transfection of a plasmid expressing Flag-CHK1 – but not its empty vector (EV) control – is sufficient to cause γ H2AX and pRPA accumulation as detected by immunoblotting and immunofluorescence (see figures below). This is consistent with a previous report showing accumulation of γ H2AX upon CHK1 ectopic expression in cultured cells and an apparent negative feedback loop leading to CHK1 cleavage

(<https://pubmed.ncbi.nlm.nih.gov/18550533/>).

Thus, unfortunately these limitations prevent the execution of the experimental design proposed by this reviewer.

Nevertheless, we are happy to highlight that in the revised manuscript we show that deoxy-nucleoside (dN) supplementation significantly prevents SARS-CoV-2 infection-induced DNA damage accumulation, DDR activation, and pro-inflammatory cytokine transcription (Fig. 2I-N). Since we propose that the ultimate impact of CHK1 downregulation is a reduced supply of dNTPs, these results nicely support our model and indirectly address the request of this reviewer.

4. In Figure 2J, add the immunofluorescence panel like Figure 2H which would be a powerful experiment to show that cells expressing ORF6 and NSP13 reduced RRM2 levels and caused increased H2AX and RPA phosphorylation.

We are happy to include these additional powerful results (new Fig. 3A,B).

5. In Figure S2G, please show the expression of the indicated SARS-CoV-2 proteins, as well as RRM2 proteins, by western blot.

We have now included the requested immunoblots and the relative quantifications in Fig. S3.

6. In Figure 2J, how the SARS-CoV-2 gene products ORF6 and NSP13 reduce the CHK1 protein levels? Can the authors provide mechanistic explanation?

This is an important point that we have addressed in the revised version of the manuscript. Here below we summarize our insights on the mechanisms of ORF6- and NSP13-mediated CHK1 degradation.

It has been published that ORF6 disrupts the nuclear import of several proteins by interacting with the nuclear pore complex

(<https://www.pnas.org/doi/10.1073/pnas.2016650117>); here we show that it prevents also CHK1 translocation from the cytoplasm to the nucleus, causing its degradation via the proteasome. Indeed, treatment with the proteasome inhibitor MG132 leads to the accumulation of poly-ubiquitinated CHK1, ultimately rescuing its levels in ORF6-expressing cells. In addition, a point mutation in ORF6 abrogating its interaction with the nuclear pore complex (<https://www.pnas.org/doi/10.1073/pnas.2016650117>) is sufficient to abolish ORF6 impact on CHK1 levels and on DNA damage generation. This is more extensively described in a dedicated portion of the revised text and in Figure 4.

Instead, the viral protein NSP13 exploits another mechanism to downregulate CHK1 protein levels. We discovered that it causes CHK1 degradation through the autophagic route. Indeed, autophagy inhibition, either through a set of specific pharmacological inhibitors or RNAi-mediated knock down of key autophagy components, impedes CHK1 loss caused by NSP13 expression (new Figure 5).

We are quite excited to have significantly improved our manuscript, prompted by our reviewers.

7. It was known that, upon DNA damage, Chk1 activation up-regulates RRM2 expression through the E2F1 transcription factor. CHEK1 down-regulation by siRNA and small molecule inhibitors of

Chk1 blocked RRM2 induction by CPT (J Biol Chem. 2009. PMID: 19416980). Another study also provided evidence for RNR2 regulation by Chk1 for S-phase progression as the author has cited (Biochem. Biophys. Res. Commun. 2008). Therefore, the figure 3A-D is in lack of innovation and therefore they should not be placed in the main figures.

We agree with the referee and we have now moved these results, generated to confirm the aforementioned observations in our experimental settings, in the supplementary section.

8. The figure 3F, Please add ELISA experiments for those cytokines such as IL1b, IL6, CXCL9, CXCL10, TNFa. What is the major mechanism by which CHK1 loss leads to increased mRNA expression of inflammatory factors?

We have now performed the requested ELISA experiments to detect pro-inflammatory cytokines and chemokines released by CHK1-depleted Calu-3 cells. As predicted, we observed that CHK1 depletion through RNAi leads to increased secretion of IL6, CXCL9 and CXCL10 (Fig. S2S).

In regard to the mechanisms underlying inflammatory cytokine and chemokine expression upon CHK1 loss, in our manuscript we provide evidence for at least three of them (likely acting in concert): DNA damage generation (Fig. S2M-O), cGAS/STING engagement by micronuclei accumulation (Fig. S2P) and STAT1/p38 activation (Fig. S2M,N).

9. SARS-CoV-2 N protein should undergo Liquid-liquid phase separation and perform high efficient condensation in cytoplasm in cells, why the N proteins in figure 4A, 4C were largely diffused in cytoplasm?

We performed additional staining for SARS-CoV-2 N both in infected cultured cells and *in vivo* that confirmed the presence of viral N-protein also in the form of cytoplasmic aggregates (e.g., Fig. S1C,N, S5J and 8C). This is in agreement to the published literature (e.g., <https://www.nature.com/articles/s41556-021-00710-0>).

In figure 4E, why the SARS-CoV-2 N proteins mainly located in the nucleus, quite different from figure 4A, 4C?

We extensively probed for SARS-CoV-2 N protein localization by confocal microscopy analyses and we could confirm that N-protein, although mainly cytoplasmic, also localizes to the nucleus of infected cells (Fig. S5J,K). This is indeed consistent with the most recent reports:

<https://www.nature.com/articles/s43587-022-00170-7>

<https://www.embopress.org/doi/full/10.15252/msb.202110396>

<https://journals.plos.org/plosbiology/article?id=10.1371/journal.pbio.3001158>. We have

now replaced Figure 4E with a revised one that is more representative of N protein subcellular distribution (see new Fig. 6E).

Please add positive control experiments: immunofluorescence experiments of 53BP1 and γ H2AX foci after DNA damage.

The requested control immunofluorescence experiments have been included in Fig. 6E.

10. In the article, authors observed that SARS-CoV-2 N-protein can reduce 53BP1 foci (Fig 4A-D). However, in experiment using Calu-3 cells and *in vivo* experiment, SARS-CoV-2 infection cannot cause the reduction of 53BP1 foci (Fig S4A-B and Fig 5). This conclusion is inconsistent with their previous results (Fig 4A-D). **Our results are consistent and we take this opportunity to better explain them. It is known that γ H2AX and 53BP1 foci colocalize (e.g., Fig. 6E). We discovered that viral infection or N protein expression uncouples this, reducing 53BP1 foci formation at γ H2AX signals. This is consistently observed in all settings studied in our work (Fig. 6 and S5A,B). However, to further strengthen this observation in Calu-3 cells, we performed an additional experiment of infection that, combined with the previous ones, showed a statistically significant downregulation of 53BP1 foci in infected Calu-3 (Fig. S5A,B). Such results are even more dramatic if considering the increase of γ H2AX caused by viral infection (Fig. S5A,B). To confirm such findings also *in vivo*, we extended the analysis of 53BP1 and γ H2AX levels in mouse lungs (Fig. 7) and in human specimens (Fig. 8C,D). The results obtained *in vivo* consistently showed a reduced ability of 53BP1 to accumulate at γ H2AX foci in SARS-CoV-2 infected tissues. We have now revised the text to make this point clearer.**

11. Was the SARS-CoV-2 infection-induced damage repair primarily accomplished by NHEJ rather than homologous recombination? Have the author examine the HR efficiency in cells expressing SARS-CoV-2 N-protein using HR repair assay?

We have now carried out the experiment suggested by the referee and observed no significant impact of SARS-CoV-2 N-protein on the repair through HR (see the histogram below).

The reduced RAD51 levels observed upon SARS-CoV-2 infection (see point 2 raised by this referee) are therefore likely independent of N-protein.

Upon Editor and reviewers' approval, we propose (also for space reasons) not to include these results in the manuscript.

12. Please verify the activation of CHK1 caused by SARS-CoV-2 infection in mice model by immunohistochemistry or immunofluorescence in figure 5A.

We have now probed for CHK1 protein level by IHC in mock- and SARS-CoV-2-infected mice and confirmed its downregulation also in this *in vivo* setting (see new Fig. 7).

13. In figure 5A, an experiment using SARS-CoV-2 infected hACE2-mouse with irradiated (IR) or not irradiated (NI) treatment to demonstrate, consistent with their observations in cultured cells (Fig. 4A-D), that SARS-CoV-2 rescues suppresses 53BP1 activation would be an added value to the manuscript.

We agree with this suggestion and we thought hard about this. However, feasibility was an insurmountable obstacle. Very few mouse facilities have a BSL-3 (biosafety level 3) designation that allow them to handle mice infected with SARS-CoV-2 – we were lucky to find one for our experiments. Among those we identified and we could access, none had an irradiator inside the BSL-3 area.

14. The authors suggest that SARS-CoV-2 can reduce 53BP1 foci. Is 53BP1 foci disrupted in lung and nasal mucosal tissues of COVID-19 patients, compared to patients that stained negative for the presence of SARS-CoV-2, or to subjects not diagnosed for the pathology (non-COVID)?

Following this reviewer's advice, we analyzed the lung of both COVID-19 patients and subjects not diagnosed for the pathology (non-COVID) but with viral pneumonia (influenza virus infection). By staining for γ H2AX and 53BP1 we observed that DNA damage was massively detected in COVID lungs, whereas very few DNA damage foci were observed in non-COVID lungs (Fig. 8A-D). As positive controls of DDR activation, we *ex vivo* treated healthy sections of lungs from a cancer patient with bleomycin, a known DNA damaging agent. Upon bleomycin administration, 80% of cells that stained positive for γ H2AX also displayed signals of 53BP1 (Fig. 8C,D). In contrast, a significant lower number of cells scored positive for both γ H2AX and 53BP1 in COVID-19 patients, consistent with impaired 53BP1 recruitment also *in vivo* in human lungs infected by SARS-CoV-2 (Fig. 8C,D).

We also performed the same analysis on nasal mucosae. In this case, we could not have access to COVID-19-negative mucosae, due to ethical limitations (surgery would have disfigured patients and families did not allow that, it was possible only during the pandemic). Still, among the available mucosae, we managed to discriminate between cells that scored positive for the presence of SARS-CoV-2 by FISH (FISH+) and those that did not (FISH-). Also in this case, γ H2AX signal was exclusively present in FISH+ cells (Fig. 8A-D). In addition, the number of double positive 53BP1+ γ H2AX+ cells was significantly lower in FISH+ cells, again consistent with the capacity of the virus to impair the recruitment of 53BP1 (Fig. 8C,D).

15. The authors identify two candidate mechanisms whereby SARS-CoV-2 infection causes DNA damage and an altered DNA damage response: (i) through the degradation of the DDR kinase CHK1 and the loss of RRM2, impaired S-phase progression, DNA damage accumulation, induction of inflammatory cytokines and (ii) impairs 53BP1 recruitment at DSB by competing with diIncRNAs binding, reducing DNA repair by NHEJ. Are there any connections between those two mechanisms? Will there be a compensatory effect?

They are likely independent events as CHK1 loss does not affect 53BP1 foci formation and the other way around. However, they coexist in infected cells.

We don't see room for "compensatory effects", since both mechanisms cause DNA damage.

16. Are there more important upstream proteins in the DNA damage pathway that are affected?

We performed a fairly thorough analysis of the whole DDR cascade. This comprises the most upstream DDR kinases including ATM, ATR and DNA-PK.

Was the type of DNA damage caused by SARS-COV-2 mainly double-strand breaks?
We focused on SSB and DSB, as indicated also by comet assays.

Minor points:

1. Add the RT-qPCR analysis of the mRNA levels of pro-inflammatory cytokines (IL1b, IL6, CXCL9, CXCL10 and TNF) in CHK1-depleted Huh7 cells.

Thank to this reviewer, we analysed the expression of pro-inflammatory cytokines and chemokines following CHK1 knockdown in Huh7 cells by RT-qPCR and found a significant upregulation of *IL6*, *IL8*, *CXCL9* and *CXCL10* mRNA levels in CHK1-depleted cells (Fig. S2R). This is indeed consistent with the activation of STAT1 and P38 observed in Huh7 cells upon CHK1 loss (Fig. S2M,N). IL1b, initially mentioned in our manuscript, has not been included in the revised version since it was rarely significantly affected in all our settings, including infection.

2. In Fig. S4F, please show the photos corresponding to the turbidity assays.

We have added these pictures in the new Figure S5G.

3. Please specify the n value for the figure 3F.

This is now specified (new Fig. S2Q).

4. In figure 6A and 6SA, please show the representative images of lung and nasal mucosal tissues from patients that stained negative for the presence of SARS-CoV-2, or to subjects not diagnosed for the pathology (non-COVID).

We have included the required images in the new Figures 8 and S7.

5. The representative images of lung and nasal mucosal tissues in figure 6C and 6E are quite different from those in figure S6A-B. Please provide an explanation.

We replaced images with better ones.

6. In figure 6C and 6E, it is better to show the representative images (γ H2AX and RRM2) of lung and nasal mucosal tissues from patients that stained negative for the presence of SARS-CoV-2, or to subjects not diagnosed for the pathology (non-COVID).

We have now added in Figures 8 and S7 the representative images of all the markers analyzed also in non-COVID tissues.

Reviewer #2 (Remarks to the Author):

In the present manuscript Gioia and collaborators explore the ability of SARS-CoV-2 to induce DNA damage upon infection. They suggest that the observed DNA damage is induced by a non canonical DNA damage response pathway based on interrogation of human cell lines infections (Huh-7 and Calu-3), mice infections and post mortem human sample from COVID-19 patients. Actually, they suggest a double route that generates genomic instability. The first concerns CHK1-mediated down-regulation of RRM2 that in turns leads to dNTPs shortage followed by replication stress, DNA damage and inflammatory cytokines production. The second, describes a competitive action of SARS-CoV2 N protein over 53BP1 by binding DNA damage induced long non coding RNAs (dilncRNAs - a previous discovery from the group) leading to impairment of the NHEJ repair process. Although this is a very interesting story with great potential in the field of molecular virology, many findings are not clear while mechanistic aspects remain unaddressed or confusing. These issues should be addressed in order to achieve publication level.

Major comments:

1. It is not clear to me why two cancer cell lines (Huh-7 and Calu-3) are the main cellular settings employed. As cancer cells, their genetic constitution is severely impaired (see following notes), eg both cell lines harbor mutant p53, and thus many homeostatic circuitry are altered and thus their function/reaction upon viral infection may not reflect that of the COVID-19 patients. The authors make a very limited use of primary nasal epithelial cells, which are a more ideal cellular model. I would favor the use of cellular system(s) that are normal and do not carry the mutational burden of full blown tumor cells that can severely skew the findings.

We take here the opportunity to better explain the choice of the cell lines used and how we extensively validated them for our studies.

Huh7 and Calu-3 cells are the standard in virology studies and they are widely used. Huh7 cells are naturally permissive to coronavirus infection

<https://www.nature.com/articles/s41564-020-00846-z>;

<https://pubmed.ncbi.nlm.nih.gov/14766227/>;

<https://www.sciencedirect.com/science/article/pii/S0006291X04015566>;

https://wwwnc.cdc.gov/eid/article/26/6/20-0516_article) and they are commonly employed in anti-viral screenings, including for SARS-CoV-2

(<https://journals.plos.org/plosbiology/article?id=10.1371/journal.pbio.3001490>). The same applies to Calu-3

(<https://www.nature.com/articles/s41467-020-20457-w>;
<https://www.sciencedirect.com/science/article/pii/S0092867420302294>)

which are relevant also because they are of lung origin and can be easily infected without the need for ACE2 overexpression.

However, we agree with this referee that each cell line comes with its own idiosyncrasies. For this reason, in all our DDR studies we validated the ability of these cell lines to mount a DDR by exposing them to two independent genotoxic treatments: ionizing radiations (IR) and hydroxyurea (HU). Under both conditions a robust DDR is detected in both cell lines as determined by comprehensive analyses summarized in Figure 1 and extended in Calu-3 in this revised version (Fig. S1A). Therefore, we believe that the consistent conclusions drawn from these two independent cell lines are solid and biologically sound.

Of course, we agree with this referee on the great value of the use of human primary nasal epithelial cells. For this reason we validated key conclusions with them and, following this referee recommendation, we extended their use as shown in Figures S1C,D, S1L-O and S2A,B. Of note, we would like to highlight that the use of such primary cells is not routinary, including in high profile publications on SARS-CoV-2 and both their generation (they were obtained by “vigorous brushing” of lab members’ noses) and their culture is not trivial.

Also, additional time points beyond 48h should be included in the analysis, given that SARS-CoV-2 infection can persist more than 48h. In this way a broader picture will be gained on the status of DDR components, as some exhibit a weird pattern (Figure 1A). For example: i) why p-CHK1 is increased at 1 and 24 hrs and then decreases at 48h (including total levels), ii) why total ATR levels (and consequently basal p-ATR) gradually increase over the time spectrum analyzed (compared against H3 levels used as control loading), given that DDR apical kinases (ATM, ATR, DNA-PK) are activated mainly at post-transcriptional level? Similar pattern appears to be for ATR in hydroxyurea treatment.

Virus-induced cytotoxicity is a well-known concept in virology and correlates with virus strains and infected cell types and it often culminates with the lysis of the host cell

(<https://www.sciencedirect.com/science/article/pii/B978012374984000989X>). In the case of Huh7 infection with SARS-CoV-2, this effect occurs with MOI 0.1 at 48 hours post infection, preventing meaningful analyses beyond that period. For this reason, we did not advance our analysis beyond 48 hours, when viral cytotoxicity becomes widespread and make any result hard to interpret.

It is important to bear in mind that our conclusions drawn from the study of cultured cells are robustly confirmed *in vivo* well beyond 48 hours post infection in two distinct settings: infected hACE2 mouse lungs, and in human lungs and human nasal epithelia.

In regard to the mechanism of CHK1 downregulation, the revised manuscript now has an entire section dedicated to it (see point n. 2 raised by this reviewer).

In regard to ATR, we show here below the quantification of three independent biological replicates and one representative figure from the original manuscript: no significant upregulation of ATR compared to H3 used as loading control is observed (see below).

Notes: HUH7 cells harbor

i) 108 genes with mutations

https://maayanlab.cloud/Harmonizome/gene_set/HUH7/CCL+Cell+Line+Gene+Mutation+Profiles

ii) 1079 genes with low or high copy number

CALU3 cells harbor

i) 79 genes with mutations

https://maayanlab.cloud/Harmonizome/gene_set/CALU3/CCL+Cell+Line+Gene+Mutation+Profiles

ii) 1013 genes with low or high copy number

https://maayanlab.cloud/Harmonizome/gene_set/CALU3/CCL+Cell+Line+Gene+CNV+Profiles

As explained above, our results are based on the analyses of two independent cultured cell lines (Huh7, Calu-3 – standards in the virology field) that we extensively validated for DDR proficiency. We additionally strengthened key conclusions in human primary nasal epithelial cells in culture, and further confirmed them *in vivo* in mouse lungs and in human lungs and nose epithelia.

2. The authors should provide evidence on the molecular mechanism(s) responsible for the CHK1 decreased levels during SARS-CoV-2 infection by ORF6 and NSP13. The immunofluorescent and immunoblot analyses in Figure 2H,J are just correlative while Figure S2C supports a (post-)translational impact with no other evidence on how ORF6 and NSP13 decrease CHK1 protein levels. Do ORF6 and NSP13 affect directly CHK1, or other mediator(s) intervene? Previous studies, quoted also in the manuscript (refs 21 and 28), have described the axis ATR/ATM-CHK1-E2F1-RRM2. This molecular pathway should also be interrogated if it is indeed the one responsible in the examined settings. If ORF6 and NSP13 directly suppress CHK1, in which subcellular compartment does this take place, given that the virus replicates in the cytoplasm?

We thank the reviewer for the suggestion. Our revised manuscript includes a dedicated section describing the mechanisms controlling ORF6/NSP13-mediated CHK1 loss.

Briefly, we discovered that ORF6, a protein that interacts with the nuclear pore complex and disrupts protein trafficking

(<https://www.pnas.org/doi/10.1073/pnas.2016650117>),

prevents CHK1 translocation from the cytoplasm to the nucleus, causing its degradation via the proteasome. Indeed, treatment with the proteasome inhibitor MG132 leads to the accumulation of poly-ubiquitinated-CHK1, ultimately rescuing CHK1 levels in ORF6 expressing cells. In addition, a point mutation in ORF6 disrupting its interaction with the nuclear pore complex (<https://www.pnas.org/doi/10.1073/pnas.2016650117>) is sufficient to abolish ORF6 impact on CHK1 levels and on DNA damage generation. This is more extensively described in a dedicated portion of the revised text and in Figure 4.

Instead, the viral protein NSP13 exploits another mechanism to downregulate CHK1 protein levels. We discovered that it causes CHK1 degradation through the autophagic route. Indeed, autophagy inhibition, either through a set of specific pharmacological inhibitors or RNAi-mediated knock down of key autophagy components, impedes CHK1 loss caused by NSP13 expression (new Figure 5).

Moreover, in an actual infection all the viral proteins, that the authors examine separately applying expression vectors, are expressed simultaneously. Thus an important complementary/rescue experiment that would simulate what takes place in vivo would be to infect cells with the virus and subsequently silence the particular protein (ORF6 or NSP13) and examine if the outcome copes with the results of the transfection assays.

This is an interesting experiment that we considered ourselves. However, one has to consider that, like most viruses, SARS-CoV-2 expresses its proteins through overlapping open reading frames (ORFs). Thus, designing a siRNA that effectively knocks down one ORF only, without interfering with the expression of others, is unfeasible.

3. Authors showed that reduction of dNTPs levels leads to DNA replication defects. Rescue experiments in the form of supplementing dNTPs and CHK1 reconstitution should be performed to confirm their hypothesis. Moreover, rescue experiments should be accompanied by additional experimental verifications like assessing DDR markers expression, comet assay, production of inflammatory cytokines.

We very much agree with this referee about the value of the experiments suggested. For this reason, we carried out both of them.

However, we soon realized this is challenging and we share here our findings. We observed that transfection of a plasmid expressing Flag-CHK1 – but not its empty vector (EV) control – is sufficient to cause γ H2AX and pRPA accumulation as detected by immunoblotting and immunofluorescence (see figures below). This is consistent with a previous report showing accumulation of γ H2AX upon CHK1 ectopic expression in cultured cells and an apparent negative feedback loop leading to CHK1 cleavage (<https://pubmed.ncbi.nlm.nih.gov/18550533/>).

Thus, unfortunately these limitations prevent the execution of the experimental design proposed by this reviewer.

Differently, deoxy-nucleoside (dN) supplementation was more informative. Indeed, we administered dNs to SARS-CoV-2-infected Huh7 or Calu3 cells and observed that their supplementation was sufficient to significantly reduce DNA damage accumulation, DDR activation, and pro-inflammatory cytokine transcription (Fig. 2I-N). Since we propose that the ultimate impact of CHK1 downregulation is a reduced supply of dNTPs, these results nicely support our model and address the request of this reviewer.

4. In Huh-7 cells, despite the activation of the DNA damage response pathway (Figure 1A), there is no evidence of apoptosis induction (Figure S1A). Given that markers indicative of senescence are observed in infected mice, the authors should examine for the presence of this phenomenon in the employed cell lines and in the sections from the animal and human tissues (see point 9 below) using the recently recommended guidelines (Nat Protoc PMID 33911261).

After this manuscript was initially submitted, a number of articles provided excellent evidence of SARS-CoV-2-induced cellular senescence, including reports from the Schmitt lab

(<https://www.nature.com/articles/s41586-021-03995-1>)

and Gorgoulis lab

(<https://erj.ersjournals.com/content/early/2022/01/20/13993003.02951-2021>) among others

**(<https://www.nature.com/articles/s43587-022-00170-7>;
<https://www.atsjournals.org/doi/full/10.1165/rcmb.2021-0205LE>).**

We believe the value of our findings is to contribute to provide a mechanistic insight on the causative events driving the cellular senescence now widely reported.

In any case, we did our best to perform most of the analyses recommended in the above-cited guidelines and generated too our own evidence of cellular senescence in infected cells, including non-malignant HNEpC (see SA- β -gal in Fig. S1L,M, p21 and Ki67 in Fig. S1N,O), and in tissues (see p16, p21 and DDR in Figures 7, 8, S6 and S7). In addition, we provided evidence of SASP induction upon SARS-CoV-2 infection by RT-qPCR in two different cell lines (Fig. S1J).

5. The authors explore cytokine production only in Calu-3, based on the rationale that only these cells express sGAS/STING, while Huh-7 do not. Considering the previous point, the authors are advised to examine for cytokine production in Huh-7 cells also, particularly if these cells enter senescence upon virus infection.

We thank the reviewer for this suggestion. We thus probed by RT-qPCR the expression of inflammatory cytokines and chemokines also in Huh7 cell line. Although we observed that also here they are upregulated, the induction of some of them was more attenuated – see examples below and new Fig. S1J.

To further probe the events underlying the cGAS/STING-independent cytokine induction observed in Huh7 cells, we analyzed the engagement of additional pathways commonly associated with cytokine and chemokine expression. Interestingly, we observed that both STAT1 and p38/MAPK pathways are activated upon SARS-CoV-2 infection (new Fig. S1I), thus providing a likely explanation for their expression.

Notably, the induction of pro-inflammatory genes is consistent with SARS-CoV-2-induced cellular senescence, observed also in Huh7 cells (new Fig. S1L,M) which is characterized by the activation of the so-called senescence-associated secretory phenotype (SASP).

6. The authors should check in the CHK1 knocked-down cells (Figure 3) the DNA integrity and the expression of cGAS/STING for its role in the inflammatory response.

Prompted by this reviewer, we depleted CHK1 in Calu-3 cells and studied its impact on cGAS/STING engagement and DNA integrity at the single-cell level by immunofluorescence. We observed that loss of CHK1 was accompanied by an increase of DNA damage as indicated by augmented number of γ H2AX foci (Fig. S2O) and the formation of micronuclei that stained positive for cGAS (Fig. S2P).

7. The authors support that the N protein interferes with 53BP1 and impairs DNA repair by NHEJ. What is the alternative mechanism(s) by which cells repair the DNA damage, since no apoptosis is observed?

In fact, our hypothesis is that viral N-protein, by hampering NHEJ, leads to accumulation of DNA damage and consequent induction of senescence. Although N-protein has no impact on repair by HR (see below), we cannot exclude that other viral proteins may hinder this repair pathway and therefore contribute to accumulation of unrepaired DNA damage.

Given that SARS-CoV-2 replicates in the cytoplasm, while 53BP1 is a nuclear localized protein, does N protein enter the nucleus of the cells upon SARS-CoV-2 infection? Analysis of cytoplasmic and nuclear fractions, further to total lysates, could help define if nuclear localization of N protein takes place. In such a case what is the mechanism of nuclear translocation?

This is a valid point that is now being addressed by the most recent literature. Indeed, it is emerging that N-protein also localizes to the nucleus of infected cells, see:

<https://www.nature.com/articles/s43587-022-00170-7>

<https://www.embopress.org/doi/full/10.15252/msb.202110396>

<https://journals.plos.org/plosbiology/article?id=10.1371/journal.pbio.3001158>

To generate our own independent evidence, we have additionally performed quantitative analyses using confocal microscopy and we can confirm that N-protein also localizes to the nucleus of infected cells (Fig. S5J,K).

Moreover, rescue experiments consisting of viral infection followed by N-protein silencing should be also performed, as described in point 2.

As discussed above (point 2 by this referee), overlapping ORFs make this approach virtually impossible.

Finally, it would be very solid for the manuscript to find in vivo evidence (eg in tissue sections) of nuclear localization of N protein and possibly co-localization with dilncRNAs.

We are happy to report that by staining mouse and human tissues for N-protein we can provide evidence of N-protein localization also in the nucleus (see Fig. 7 and 8E), as also reported by others

<https://www.nature.com/articles/s43587-022-00170-7>.

For what dilncRNA colocalization with N-protein concerns, since we cannot predict the chromosomal sites of DNA damage induced by viral infection, we cannot design probes to perform a combined immunoFISH detection experiment.

8. To support their hypothesis the authors investigate *in vivo* mouse models and human COVID-19 samples. SARS-CoV-2 presence should be confirmed by qRT-PCR and IHC/IF analysis in all materials from *in vivo* experiments. Especially in Figure 5 (A and B) it should be added an IHC figure for SARS-CoV-2 detection in a parallel serial section analysis for the detection of double positive cells (SARS-CoV2 and γ H2AX/53BP1) in the four different conditions (PBS/virus infection, NI, IR). Alternatively double or triple IF analysis would be also fine.

We performed the requested double- and triple-staining in *in vivo* models to confirm the presence of SARS-CoV-2 along with the markers tested. Thus, now all samples studied have been validated for the presence of the virus by RT-qPCR, FISH/ISH and IF/IHC techniques, confirming and extending the results initially proposed (see revised Fig. 7, 8, S6 and S7).

Moreover, to support their hypothesis the authors should interrogate CHK1 and RRM2 expression in SARS-CoV-2 infected mouse tissues.

We stained mouse lungs for both CHK1 and RRM2 along with SARS-CoV-2 N-protein or its RNA. As observed in cultured cells, both CHK1 and RRM2 signals were strongly reduced in cells that stained positive for SARS-CoV-2. See new Figure 7.

9. A consequence of SARS-CoV-2 infection is the induction of cellular senescence, as previously reported and quoted in the manuscript. According to a previously published multi-marker algorithm, SA- β -gal activity and/or lipofuscin accumulation (GL13 staining) and co-staining with other markers frequently observed (p16INK4A, p21WAF1/Cip1) or absent (proliferation markers, Ki67) in senescence should be applied in mouse and human tissue sections (Nat Protoc PMID 33911261). The multi-marker workflow should be confirmed by serial section analysis or with double/Triple staining by IHC or IF analysis in the four different conditions (PBS/virus infection, NI, IR).

In addition to a number of articles that provided compelling evidence of SARS-CoV-2-induced cellular senescence, including reports from the Schmitt lab (<https://www.nature.com/articles/s41586-021-03995-1>) and Gorgoulis lab (<https://erj.ersjournals.com/content/early/2022/01/20/13993003.02951-2021>) among others (<https://www.nature.com/articles/s43587-022-00170-7>; <https://www.atsjournals.org/doi/full/10.1165/rcmb.2021-0205LE>), we also generated our own evidence of cellular senescence in infected cells (e.g. see SA- β -gal in Fig. S1L,M, p21 and Ki67 in Fig. S1N,O) and in tissues (see p16, p21 and DDR in Figures 7, 8, S6 and S7).

10. Staining should be provided for ACE2 and markers, such as against type II pneumocytes, to identify in vivo the type of cells that the virus infects as well as the presence of the virus, by co-staining or serial section analysis.

Following this suggestion, we performed the requested analyses and co-stained mouse lungs for SARS-CoV-2 genome and for ACE2, or proSP-C (prosurfactant protein C, a marker of AT2 cells). Consistently with what reported by us and others

(e.g., <https://erj.ersjournals.com/content/early/2022/01/20/13993003.02951-2021>; <https://www.embopress.org/doi/full/10.15252/embr.202153658>), we confirmed that SARS-CoV-2 mainly targeted type II pneumocytes in mouse lungs (see the images attached below).

Given the large number of panels of the revised manuscript, we leave the decision to the Editor whether to include also these results or leave them for referees only.

11. Many IF photos taken in the examined cellular systems are not convincing, particularly for SARS-CoV-2 proteins. Merging of IF photos should be done to better delineate spatial distribution of contents depicted. Also tissue IF photos in Figure 6C,E are similarly not so convincing. Magnification in Figure S6B is blurred.

We have now improved the quality of all the images of *in vivo* models and replaced them with new ones.

Minor comments:

1. What is the nature of the mock condition applied for SARS-CoV-2 infection in Figure 1A?

The mock is an identical parallel treatment, in which medium of the non-infected cells was replaced with a fresh one without supplementary FBS, at the same volume as infected cells. Cells were then incubated for 1 h at 37 °C and 5% CO₂, in parallel with infected cells. Afterwards, mock medium was removed, cells were washed once with PBS 1x and fresh medium +2% of inactivated FBS was added to the cell culture, as well as in infected cells. We have included this information in the revised method section.

Notably, Alessandro Marcello is a virologist with 17 years of viral infection studies and all infections were carried out in his lab and by his experienced team.

2. In Figure 1, CHK1 protein expression is examined at both total- and phospho-levels. In subsequent figures that depict CHK1 immunoblot analyses, this is not defined. In case total levels are estimated why did the authors choose to examine this form instead of the active, phosphorylated CHK1?

We initially studied how SARS-CoV-2 infection affected DDR activation and thus we probed for both phosphorylated and total DDR marker levels. When we observed that viral infection has an impact on the total protein levels of some DDR proteins (like CHK1), and since, with no CHK1 protein no CHK1 protein phosphorylation events were to be studied, we focused our analyses on the mechanisms of total CHK1 reduction following SARS-CoV-2 infection.

3. Figures S1 D,E,F could be included in main Figure 1.

We displayed these results in the supplementary section purely for space reasons. We are happy to include them among the main figures upon advice from the Editor.

4. Since the Huh7 cell line does not express cGAS or STING, the authors employ the Calu-3 cells in order to detect the damaged nuclear DNA in cytosol and the induction of pro-inflammatory response gene (Figure S1 D). Authors should carry out the comet assay also in the Calu-3 cell line since the DNA fragmentation, the cytosolic DNA and the inflammatory phenotype are detected in different cell lines.

We thank the reviewer for the suggestion and thus we performed the comet assay in Calu-3 cells as well. We are happy to confirm that we observed increased comet tail moment (more fragmented DNA) in infected Calu-3 compared to mock-infected cells (Fig. S1E,F).

5. Please quantify the results for IF experiments in Figure 2C.

Quantifications are now shown in Figure 2D.

6. Please add the name of the cell line at the top side of each image/graphs.

Where possible we have included this information.

7. In Figure S4A is the IF for N protein staining specific? The staining pattern is very peculiar, considering also the other markers assessed in this panel (DAPI, γ H2AX, 53BP1) and in comparison with panels of Figure 4.

The peculiar aspect of N protein in this image can be easily explained by the fact that Calu-3 cells are prone to form syncytia: this, together with the high levels of expression of this viral protein, produces such signal. This has also been observed by others and, as a comparison, we paste here below an image from the literature with these cells: <https://doi.org/10.1038/s41392-021-00800-3>.

8. Please provide an IHC Figure from mice lung for ACE2 immunostaining to confirm its expression.

Here below we provide the requested staining that confirmed the expression of ACE2 in mouse lungs.

Given the large number of panels of the revised manuscript, we leave the decision to the Editor whether to include also these results or leave them for referees only.

9. Figure 7 should be updated to include the mechanism of inflammatory response.

We have updated the scheme.

10. What were the criteria that authors relied on to apply a 0,1 MOI for cellular infection?

According to our previous experience with SARS-CoV-2 and others viruses, the criteria to apply a specific MOI is obtained by testing different MOIs and timepoints for each cell line (<https://pubmed.ncbi.nlm.nih.gov/33556379/>

<https://pubmed.ncbi.nlm.nih.gov/32835326/> <https://pubmed.ncbi.nlm.nih.gov/26355085/>).

The results obtained help us to find a balance within a proper infection (and replication of the virus) and the cytotoxic effect produced by the virus in a timeline adjusted to the aim of the experiments, which in this case is the activation of the DDR. So, we infected Calu-3 (that are highly susceptible to SARS-CoV-2 infection) with 0.1 MOI and collected them at 24 h. On the other hand, Huh7 (which are less susceptible to SARS-CoV-2 infection) showed the same pattern observed in Calu-3 at 48 h with same MOI. Finally, primary cells need a higher MOI (i.e., 1) and prolonged timepoint (i.e., 72 h) due to the low susceptibility to infection.

11. Include the quantification method applied for the IHC analysis.

We have described the quantification method for the IHC analysis in the revised manuscript.

Reviewer #3 (Remarks to the Author):

This manuscript proposes that SARS-CoV-2 infection causes DNA damage and an altered DNA damage response (DDR).

The conclusion that the viral infection causes DNA damage is well documented. However, the claim that the DDR is altered is not supported well by the data and may not be true. Thus, the only conclusion that is well validated is that SARS-CoV-2 infection induces DNA damage, whereas all the other data are very hard to interpret.

Specific Comments:

1. Fig. 1 shows that SARS-CoV-2 infection of human hepatocarcinoma Huh7 cells induces activation of DNA-PK and ATM, but not ATR. KAP1 a substrate of ATM was also phosphorylated after infection, but Chk2, also an ATM substrate, and p53 were not phosphorylated. A comet assay further revealed DNA fragmentation after infection. In a different carcinoma cell line (Calu-3), SARS-CoV-2 infection activated cGAS-STING, although these data probably need to be complemented by additional experiments and currently are only presented in the supplementary section. The authors conclude that viral infection activates the DDR pathway in a non-canonical manner. I agree with the conclusion that the DDR is activated, but not with the conclusion that the activation is non-canonical. The authors are basing this conclusion on one cancer cell line, whose DDR may be compromised simply because these are cancer cells.

We take here the opportunity to better explain the bases for our conclusions.

We studied the impact of SARS-CoV-2 infection on two, not one, established cell lines: Huh7 and Calu-3. These are the standard in virology and widely exploited in SARS-CoV-2 studies

**(<https://doi.org/10.1038/s41564-020-00846-z>;
<https://www.sciencedirect.com/science/article/pii/S0006291X04015566>;
https://wwwnc.cdc.gov/eid/article/26/6/20-0516_article;
<https://doi.org/10.1371/journal.pbio.3001490>;
<https://doi.org/10.1038/s41467-020-20457-w>;
<https://www.sciencedirect.com/science/article/pii/S0092867420302294>).**

Independent analyses of DDR activation in these two distinct cell lines generated consistent results.

However, as we agree with this referee that established cell lines may have potential limitations, we included positive controls for all individual DDR markers tested. That is to say that all immunoblots include two independent DNA damaging treatments: hydroxyurea (HU), that induces DNA replication stress by reducing dNTP pools – similarly to SARS-CoV-2 – and ionizing radiations (IR). In both cell lines, for all markers tested, full DDR proficiency was observed (Fig. 1A and S1A). Thus, we have no concerns on the strength of the conclusions reported in our manuscript.

In addition, to rule out any concern on this point, we confirmed our main conclusions in human primary nasal epithelial cells infected by SARS-CoV-2 (Fig. S1C,D, S2A,B and 6C,D). Of note, their use is not routinary and they seldom appear in publications including in high profile journals.

Finally, all key conclusions on altered DDR activation observed in established and primary cultured cells were independently validated in two *in vivo* settings: mouse lungs and human lungs and nasal epithelia.

Therefore, to summarize, our evidence of a non-canonical DDR activation is based on the independent study of two distinct well established cell lines (Huh7 and Calu-3), freshly generated human primary nasal epithelial cells, mouse lungs, human lungs and nose epithelia.

The statement on the “non canonical activation of DDR” stems from the observation that, differently from HU and IR, SARS-CoV-2 infected cells show CHK1 downregulation and reduction of 53BP1 foci formation. Nonetheless, we agree with this referee that the word “non-canonical” could be unclear and misleading and therefore we changed it with the term “altered”.

Most likely, the viral infection does not induce DNA DSBs directly, so the DDR after viral infection cannot be compared to the DDR after exposure to IR.

Indeed, this is our very same conclusion: the virus reduces dNTP levels and thus impairs DNA replication. This is the reason why, in addition to ionizing radiations, a common source of DNA damage, we used hydroxyurea (HU), an agent that also “does not induce DNA DSBs directly” because depletes cells of dNTPs: just like SARS-CoV-2 does.

Indeed, ATM phosphorylation after viral infection was much weaker than after IR (Fig. 1A) and surprisingly, Chk2 was not activated 60 minutes after IR, even though ATM was fully activated at that time point (Fig. 1A).

In Fig. 1A and S1A, we show that in cells exposed to IR CHK2 is promptly but transiently phosphorylated on threonine 68 – compare 15' post-IR and 60' post-IR. This is expected

and in agreement with the published literature (e.g.: <https://pubmed.ncbi.nlm.nih.gov/12805407/>; <https://pubmed.ncbi.nlm.nih.gov/25404613/>).

Upon viral infection, at the 1-hour time point studied, no DDR marker is activated, indicating that no DNA damage has yet been generated, thus explaining also the lack of CHK2 phosphorylation. At the 24 and 48 hours time points studied, several DDR markers are activated but Chk2 is not phosphorylated, consistent with its kinetic of phosphorylation also observed upon irradiation.

The authors also did not examine Chk2 phosphorylation in Calu-3 cells, which they absolutely should have done, if they want to claim that the DDR after viral infection is non-canonical.

We agree with this reviewer about this shortcoming and in the revised manuscript we now show our analysis of CHK2 phosphorylation in Calu-3 as well and, similarly to what observed in Huh7, we found no significant accumulation of pCHK2 upon SARS-CoV-2 infection (Fig. S1A,B).

2. Fig. 2 shows that the protein levels of Chk1 and RRM2 decrease in infected cells 48 h after infection. It is hard to interpret these data. Are the cells undergoing apoptosis or senescence? dNTP levels also decreased. Changes like this are somewhat expected (especially the decrease in dNTP levels), because the virus is using the resources of the cells for its own proliferation.

Our results indicate that individual viral products (ORF6 and NSP13) are sufficient to downregulate CHK1 and thus RRM2. The consequences of that are a demonstrated reduction of dNTPs, causing impaired DNA replication and DNA damage, ultimately leading to cellular senescence (Fig. 1, 2, S1 and S2). The causality of reduced dNTP levels (at the bottom of the cascade of events just mentioned) is demonstrated by the supplementation of deoxy-nucleosides to infected cells and the reduction in DDR activation, DNA damage accumulation and inflammatory cytokine expression observed (Fig. 2I-N). Thus, dNTP reduction is the cause, not the consequence of the events observed.

Therefore, we agree with this reviewer that the “virus is using the resources of the cells for its own proliferation” and we propose this as the evolutionary explanation for its demonstrated ability to reduce dNTP levels. As it is sometimes the case, novel findings may seem obvious when discovered, but in our case, they were, and still are, unreported.

In Fig. 2H,I,J,K the authors attribute the decrease in Chk1 and RRM2 protein levels to expression of viral proteins ORF6 and NSP13. This conclusion was based on expressing individually each viral protein in Huh7 cells and monitoring Chk1 and RRM2 levels by immunoblotting. In Fig. 2K left panel, the magnitude of reduction of RRM2 levels after ORF6 and NSP13 expression is not great. Also the data in Fig. S2G hide the variability in Chk1 levels, because the part of the graph between values 1.0 and 2.0 is drawn at a smaller scale than the part of the graph between values

0.0 and 1.0. Also there are no standard deviations shown. I presume that the authors want to conclude that the viral infection decreases Chk1 and RRM2 levels due to expression of ORF6 and NSP13, but the results do not support this conclusion.

We have performed additional and optimized experiments of ORF6/NSP13 transfection to maximize the impact of their expression on CHK1, RRM2, γ H2AX and pRPA levels. We indeed confirmed and extended our observation (i.e.: CHK1/RRM2 loss and concomitant γ H2AX/pRPA upregulation) both in bulk by immunoblots and at single-cell resolution by immunofluorescence (see the revised Figure 3).

Also no mechanism is provided to explain how ORF6 and NSP13 might reduce Chk1 and RRM2 levels.

We thank the reviewer for the suggestion. Our revised manuscript includes a dedicated section describing the mechanisms controlling ORF6/NSP13-mediated CHK1 loss.

Briefly, we discovered that ORF6, a protein that interacts with the nuclear pore complex and disrupts protein trafficking

(<https://www.pnas.org/doi/10.1073/pnas.2016650117>),

prevents CHK1 translocation from the cytoplasm to the nucleus, causing its degradation via the proteasome. Indeed, treatment with the proteasome inhibitor MG132 leads to the accumulation of poly-ubiquitinated-CHK1, ultimately rescuing CHK1 levels in ORF6 expressing cells. In addition, a point mutation in ORF6 disrupting its interaction with the nuclear pore complex

(<https://www.pnas.org/doi/10.1073/pnas.2016650117>) is sufficient to abolish ORF6 impact on CHK1 levels and on DNA damage generation. This is more extensively described in a dedicated portion of the revised text and in Figure 4.

Instead, the viral protein NSP13 exploits another mechanism to downregulate CHK1 protein levels. We discovered that it causes CHK1 degradation through the autophagic route. Indeed, autophagy inhibition, either through a set of specific pharmacological inhibitors or RNAi-mediated knock down of key autophagy components, impedes CHK1 loss caused by NSP13 expression (new Figure 5).

3. Fig. 3 shows that depleting Chk1 induces cell cycle arrest in S phase, DNA damage and moderately increased expression of certain interleukins. These experiments are meaningful, if we accept the premise of the authors that the viral proteins ORF6 and NSP13 suppress Chk1 expression and that the phenotypes observed after viral infection are only due to the decrease in Chk1 expression. I have a hard time believing this, because a virus replicating in cells will likely have many effects on the cell.

We would like to stress that we never claimed that ALL effects exerted by a complex virus like SARS-CoV-2 in the cell are mediated by CHK1 loss. But we demonstrated that CHK1 depletion is sufficient to recapitulate many of the phenotypes that we uncovered associated with SARS-CoV-2 infection, namely arrest in S-phase, accumulation of DNA damage, DDR activation and activation of pro-inflammatory pathways (Fig. S2I-S).

4. Fig. 4 shows that 53BP1 does not localize with gH2AX foci in virus infected cells. Further, the authors showed that the SARS-CoV-2 N protein associates with diIncRNAs, preventing 53BP1 from interacting with these RNAs and therefore preventing 53BP1 focus formation. Expression of the N protein also suppressed NHEJ, as monitored in EJ5-GFP reporter cells. However, the magnitude of the effect is very small (Fig 4J) and a control reporter cell line (eg DR-GFP) was not examined.

Here below we show in parallel the impact of N-protein expression or 53BP1 knockdown tested in the same assay executed by the same researcher (the first author of this manuscript) as published in Scientific Reports (10.1038/s41598-019-42892-6), as well as in a report published by another group in Nature Communications (10.1038/ncomms13785). It is apparent that the reduction is quantitatively nearly identical in both instances, suggesting that N-protein expression impacts on NHEJ to an extent similar to 53BP1 loss.

In addition, following the advice of this referee, we tested the effect of SARS-CoV-2 N-protein on repair through HR and we observed no significant impact (see the image below).

If the Editor and the reviewers agree, we wish, also for space reasons, to leave this last result for referees only.

Another control would be to examine MDC1 foci, as these, in contrast to 53BP1 foci, should not be affected by the N protein.

We thank the reviewer for this suggestion. We thus monitored MDC1 activation in cells expressing SARS-CoV-2 N-protein upon DNA damage generation using IR. Our data showed that viral N-protein did not affect MDC1 activation, as also anticipated by this referee (see the image below). However, purely for lack of space and upon Editorial approval, we are prone not to include such results in the current version of the manuscript.

5. Fig. 5 examines gH2AX and 53BP1 foci in lung tissues of virus infected mice and mice exposed to IR. 53BP1 foci were not induced in the virus infected cells, whereas gH2AX foci were weakly induced. There is a problem again with the y axes, which makes it hard to compare the foci induced by viral infection vs IR. Nevertheless, this experiment shows in a physiological setting that viral infection induces DNA damage.

We performed a more thorough and automatized analysis of both 53BP1 and γ H2AX activation in infected mice which confirmed the results obtained previously; in addition, we have fixed the issues with the “y” axis. These data are now part of the new Figures 7 and S6.

6. Fig. 6 shows gH2AX signal in postmortem lung biopsies of COVID-19 patients. p21 expression was also induced in these cells, while RRM2 levels were decreased. The authors attribute the decrease of RRM2 to the same mechanism as in cell lines; however, the decrease in RRM2 may be secondary to cell senescence and a reduced number of cells in S phase. Moreover, the magnitude of the decrease was very small (Fig. 6F).

In this manuscript we provide causal evidence of viral products-mediated CHK1 loss, consequent RRM2 decrease, reduced dNTP levels, DNA damage accumulation, DDR activation and cellular senescence. The fact that CHK1 knockdown decreases RRM2 levels and that dN supplementation reduces DNA damage accumulation support a causative link in such events. This is nicely recapitulated *in vivo* where, of course, it is impossible to establish causality.

Therefore, in addition to the results mentioned above, it is worth reminding we have shown that infected cells although accumulating in S-phase (Fig. 2F,G) where RRM2 amounts peak, display reduced RRM2 levels. This result uncouples RRM2 levels from cell cycle phases.

Decision Letter, first revision:

Dear Dr d'Adda di Fagagna,

Thank you very much for your email asking us to reconsider our decision on your manuscript, "SARS-CoV-2 infection causes DNA damage through multiple mechanisms". We are always willing to hear the authors' perspective, but we must first prioritize decisions on new submissions. We appreciate your patience while we considered this appeal.

I have now discussed your manuscript and the referees' comments and your rebuttal, in detail with my colleagues, and we are willing to consider the revised manuscript, provided that nothing similar is accepted for publication at Nature Cell Biology or published elsewhere in the meantime.

For re-review, please submit the Reporting Summary, Editorial Policy Checklist, and source data files (details below) along with your rebuttal to the full reviews (verbatim) and your revised manuscript. Please pay close attention to our guidelines on statistical and methodological reporting (listed below) as failure to do so may delay the reconsideration of the revised manuscript. In particular please provide:

- a Supplementary Figure including unprocessed images of all gels/blots in the form of a multi-page pdf file. Please ensure that blots/gels are labeled and the sections presented in the figures are clearly indicated.
- a Supplementary Table including all numerical source data in Excel format, with data for different figures provided as different sheets within a single Excel file. The file should include source data giving rise to graphical representations and statistical descriptions in the paper and for all instances where the figures present representative experiments of multiple independent repeats, the source data of all repeats should be provided.

On resubmission, please provide the completed Editorial Policy Checklist (found here <https://www.nature.com/documents/nr-editorial-policy-checklist.pdf>), and Reporting Summary (found here <https://www.nature.com/documents/nr-reporting-summary.pdf>). This is essential for reconsideration of the manuscript and these documents will be available to editors and referees in the event of peer review. For more information see below. Please also ensure that the presentation of statistical information in the revised submission complies with Nature Cell Biology's statistical guidelines (see below).

Please use the link below to submit the complete manuscript files, and include a point-by-point

response to the complete reviewer comments, verbatim as provided in their reports.

[REDACTED]

Please let us know how you wish to proceed and when we can expect your revised manuscript. Thank you for thinking of NCB for this work,

With kind regards,

Melina

Melina Casadio, PhD
Senior Editor, Nature Cell Biology
ORCID ID: <https://orcid.org/0000-0003-2389-2243>

GUIDELINES FOR EXPERIMENTAL AND STATISTICAL REPORTING

REPORTING REQUIREMENTS – To improve the quality of methods and statistics reporting in our papers we have recently revised the reporting checklist we introduced in 2013. We are now asking all life sciences authors to complete two items: an Editorial Policy Checklist (found here <https://www.nature.com/documents/nr-editorial-policy-checklist.pdf>) that verifies compliance with all required editorial policies and a reporting summary (found here <https://www.nature.com/documents/nr-reporting-summary.pdf>) that collects information on experimental design and reagents. These documents are available to referees to aid the evaluation of the manuscript. Please note that these forms are dynamic 'smart pdfs' and must therefore be downloaded and completed in Adobe Reader. We will then flatten them for ease of use by the reviewers. If you would like to reference the guidance text as you complete the template, please access these flattened versions at <http://www.nature.com/authors/policies/availability.html>.

We strongly recommend the presentation of source data for graphical and statistical analyses as a separate Supplementary Table, and request that source data for all independent repeats are provided

when representative experiments of multiple independent repeats, or averages of two independent experiments are presented. This supplementary table should be in Excel format, with data for different figures provided as different sheets within a single Excel file. It should be labelled and numbered as one of the supplementary tables, titled "Statistics Source Data", and mentioned in all relevant figure legends

Decision Letter, second revision:

Our ref: NCB-A47290B

30th November 2022

Dear Dr. d'Adda di Fagagna,

Thank you for submitting your revised manuscript "SARS-CoV-2 infection causes DNA damage through multiple mechanisms" (NCB-A47290B) and for your patience as we were considering the reviews. It has now been seen by the original referees and their comments are below. The reviewers find that the paper has improved in revision. Yet, Rev#1 had a number of concerns about the revision and how their original points were addressed. We discussed these comments in depth editorially and disagree with Rev#1 regarding the fit for NCB in terms of advance. We also sought input from Rev#3 on the more experimental/technical comments of Rev#1. Please see Rev#3's additional comments at the end of Rev#3's review (below). Based on our discussions of Rev#3's input and our own evaluations of Rev#1's points, we overall feel that the final points by Rev#1 can be clarified through minor changes and edits. No further experimentation should be needed.

Therefore, we'll be happy in principle to publish the study in Nature Cell Biology, pending minor revisions to satisfy the referees' final requests and to comply with our editorial and formatting guidelines. We also have two editorial points that are important and that will need to be addressed as you prepare your final files:

1- Fig 5CD seems to show a colocalization of Chk1 with p62 (in various conditions) and quantification of the percentage of cells with such colocalization (please let us know if we are mis-understanding the labelling). This should be described directly as 'colocalization with p62' -- not inferred to be 'autophagosomes' -- in the figure/figure legend and manuscript text. It will be essential to make the data description more accurate and make all labels clearer to avoid over-interpretations.

2- Consistent with the reviewer feedback and per our standard process, please note that we will require reorganization of the figures to fit our formatting requirements in order to proceed with the manuscript. All figures must fit into a single standard page (i.e., 1 page per figure only, with all panels on the page) and adhere to a maximum page size of roughly 180mm wide x 200mm high. To ensure legibility once figures are re-sized, please use a font size of no smaller than 6pt Arial or Helvetica throughout the figures. In addition, all key data should be presented in the main figures, with Extended Data only presenting supportive information. All Supplementary Figures will have to be converted into Extended Data Figures. There is a limit of 8 main Figures for Articles and 10 Extended data figures (each Main and Extended Data Figure spanning a single page only). We will provide all these details again in our checklist (please see below) but wanted to already stress that these changes

will be required.

****If the current version of your manuscript is in a PDF format, please email us a copy of the file in an editable format (Microsoft Word or LaTeX)-- we can not proceed with PDFs at this stage.****

****We are now performing detailed checks on your paper and will send you a checklist detailing our editorial and formatting requirements in about 1-2 weeks. Please do not upload the final materials and make any revisions until you receive this additional information from us.****

Thank you again for your interest in Nature Cell Biology. Please do not hesitate to contact me if you have any questions.

Sincerely,

Melina

Melina Casadio, PhD
Senior Editor, Nature Cell Biology
ORCID ID: <https://orcid.org/0000-0003-2389-2243>

Reviewer #1 (Remarks to the Author):

In previous version, the authors only presented that SARS-CoV-2 infection can cause DNA damage, with mechanistic aspects remain unaddressed, very confusing or even not supported by their data. In this revised version, point-by-point responses were provided by the authors. However, I still found this revision, in quality, has not reached the level of Nature Cell Biology, and the overall logic and data reliability need to be further strengthened. Particularly in several places that they simply deleted the previous data and replaced it with new images or inconsistent statistical results, and they could not give a reasonable explanation for the concerns raised.

The first and most important concern is, multiple reports have been published with their major conclusions related to DNA damage responses caused by SARS-CoV-2, exemplified in PMID: 35045565 and PMID: 36254583 and also others not listed; Thus current study already lose quite of its novelty. Secondly, although some new data have been added to the returned version, the most important mice infection experiment with/without IR has not been performed. The third, many findings are not clear and the molecular mechanism (as also raised by reviewer #2 and #3) is still confusing.

In conclusion, I still feel regrettably that, there are many problems with the manuscript (logic and data reliability) and many open questions, which should be investigated carefully. Besides, a number of concerns are listed below:

Major points:

Q1: The authors did not explain why chk2 phosphorylation was not detected within 48 hours after viral infection. As described in the literature cited by the author, ATM mediated photosynthesis of T68 and promote CHK2 activation. At the 24 and 48 h time points after virus infection, several DDR markers as well as ATM are activated. (Do the authors believe that there is difference between DNA damage caused by SARS2 virus and IR?)

Q2: RAD51 is an important marker in the process of DNA damage, it is necessary to put this result in the text and explain the possible mechanism. On the other hand, the representative immunofluorescence image given by the author shows that when there is no viral infection, the cells already have about 50% RAD51 foci. This is apparently not so reliable.

Q3: "Whether CHK1 overexpression in Hun7 cells can rescue the SARS-CoV-2-induced the pro-inflammatory cytokines expression, RRM2 reduction and impaired S-phase progression? It would be an added value to the manuscript."

In this article cited by the author, most of the cells transfected with full-length Chk1 or kinase-dead truncated Chk1-(1-299/D130A) displayed no γ -H2AX signals. This result is inconsistent with the result shown by the authors. The cleavage bands shown may be a degradation of the chk1 protein. I am worried about this result.

Q5: Where is the expression of NSP13? IN Fig S3B, The "N" marked on the x-axis should be "M".

Q6: Although the authors have performed some supplementary experiments to explain the possible molecular mechanism, it is not clear and insightful. How ORF6 affect the ubiquitination level of CHK1 (especially the K48 linked polyubiquitin chain)? How does NSP13 degrade CHK1 through autophagy pathway? These important issues are worthy of further study and are very important to explain the core molecular mechanism of their findings. I believe every NCB paper should have profound mechanism which should also be novel, not just presenting some phenotypes.

Q9: Although the author has replaced the figure, it is difficult to believe the conclusion drawn by the author because of the inconsistent localization of N protein in the same assay.

Q10: In experiment using Calu-3 cells and in vivo experiment, SARS-CoV-2 infection cannot cause the reduction of 53BP1 foci (Fig S4A-B and Fig 5). This conclusion is inconsistent with the new results in the revised figures (Fig. S5A, B). In the same experiment of the previous version, the author drew different conclusions in this version, which makes it difficult to believe the authenticity of the data.

Q16: The authors seem not answering this concern. Are there more important upstream proteins in the DNA damage pathway that are affected? Was the type of DNA damage caused by SARS-COV-2 mainly double-strand breaks?

Minor points:

Q5: The representative images of lung and nasal mucosal tissues in figure 6C and 6E are quite different from those in figure S6A-B. Please provide an explanation. Why did the author delete the original high-quality figure S6A 6B in the revision?

Reviewer #2 (Remarks to the Author):

The authors have performed an extensive work to respond to the issues raised and have adequately addressed them. Therefore I have no more comments and recommend publication of this manuscript.

Reviewer #3 (Remarks to the Author):

The authors have addressed my comments. In particular, the use of COVID-19 patients' lungs and nasal mucosa is a significant addition to the analysis performed with cancer cell lines. The use of mice infected with SARS-CoV-2 is also a strength of the manuscript (mouse data were already present in the original version). I consider that the manuscript is now suitable for publication, pending minor points (see below). I believe that this manuscript presents important information to broadly understand host-viral pathogen interactions, as they relate specifically to the induction of a DNA damage response (DDR). Most viruses will need to develop mechanisms to modulate the DDR, as exemplified here.

Minor Point:

The figures seem too busy. The main figures often extend over 2 pages each and the font is already too small. Fig 2 covers 3 pages. How will these figures be visible on the print form of the journal? The authors need to invest some effort in preparing better figures. They can also print them at the size in which they would be at the printed journal to see if they are readable.

Additional comments to the authors in response to the comments of Reviewer #1

According to Reviewer 1: "The first and most important concern is, multiple reports have been published with their major conclusions related to DNA damage responses caused by SARS-CoV-2, exemplified in PMID: 35045565 and PMID: 36254583 and also others not listed; Thus current study already lose quite of its novelty."

Essentially the Reviewer is saying that the authors have been scooped. However, I looked at these publications. The first one mentions a role of cGAS-STING, which the authors of this manuscript also discuss, but otherwise the main focus of the two studies is different. Thus, I don't think that the novelty of the current manuscript is affected in a significant way. The second study examines why SARS-CoV-2 affects aged individuals more severely than young individuals and has nothing to do with the current manuscript. Thus, I don't share this "first and most important" concern with Reviewer 1.

Major points

Q1: The Reviewer refers to ATM-mediated "photosynthesis". This novel function of ATM deserves to be published in Nature, since mammals are generally not known to perform photosynthesis, except, of course, if the Reviewer did not check what he/she was writing, which I am afraid may be the case. The Reviewer concludes this point by the following question: "Do the authors believe that there is difference between DNA damage caused by SARS2 virus and IR?" I don't understand why the Reviewer is asking this question. The main point of the manuscript is that there is a different DDR and DNA damage in cells infected by SARS2 versus those exposed to IR. So the answer to the Reviewer's question is a definite Yes and the answer is supported by the data shown.

Q2: The Reviewer mentions that RAD51 foci is an important marker in the process of DNA damage, but it not happy with the basal levels of RAD51 foci. In fact, scoring RAD51 foci is known to be problematic. One needs to determine for each cell line, a threshold for the number of RAD51 foci, above which a cell would be considered RAD51-positive. Otherwise, every replicating cell will have a

small number (1-10, depending on cell line) of RAD51 foci. The DDR markers that are most relevant to this manuscript are Chk1, Chk2 and 53BP1, not RAD51. The authors examined RAD51 foci, because Reviewer 1 requested them to do so. I personally don't think that the RAD51 data should be included in the final version of the manuscript, which is also the opinion of the authors.

Q3: In this comment Reviewer 1 requested two different experiments. The authors did the first experiment and the results supported very nicely the main premise of the study. They also attempted to do the second experiment, but they found that ectopic expression of Chk1 causes DNA damage on its own, so it was not possible to proceed further. This is not a major point of the study and I don't see why Reviewer 1 is making an issue out of this.

Q5: It seems that the Reviewer has spotted a typo in Fig. S3B. The authors should correct the typo.

Q6: The Reviewer says that the authors have not provided the "profound mechanism which should also be novel" that "every NCB paper should have". The mechanism relates to how ORF6 and NSP13 target Chk1 for degradation. The authors provide a lot of mechanistic information in Figs 4 and 5. I am satisfied by the mechanistic data provided.

Q9: The Reviewer considers that there is inconsistent localization of the N protein in various experiments. This is also true in the literature, as the authors state and could relate to the levels of expression of the protein under various conditions. Since the N protein is present in both the nucleus and the cytoplasm (based on the literature and the staining performed by the authors), I don't see how this concern impacts the conclusions of the authors.

Q10: The Reviewer states that 53BP1 behaves differently in different cell lines, but the authors clearly show that the 53BP1 foci signal decreases in all cell lines and that 53BP1 co-localization with gH2AX foci is reduced in all cell lines. I don't understand what the Reviewer is worried about.

Q16: The Reviewer says that the authors do not seem to answer the questions: "Are there more important upstream proteins in the DNA damage pathway that are affected? Was the type of DNA damage caused by SARS-COV-2 mainly double-strand breaks?" However, the authors answered this question and the answer is also in the abstract of the manuscript.

Minor points:

Q5: It seems that the authors obtained better images in the course of revising the manuscript.

Decision Letter, final checks:

Our ref: NCB-A47290B

9th December 2022

Dear Dr. d'Adda di Fagagna,

Thank you for your patience as we've prepared the guidelines for final submission of your Nature Cell Biology manuscript, "SARS-CoV-2 infection causes DNA damage through multiple mechanisms" (NCB-

A47290B). Please carefully follow the step-by-step instructions provided in the attached file, and add a response in each row of the table to indicate the changes that you have made. Please also check and comment on any additional marked-up edits we have proposed within the text. Ensuring that each point is addressed will help to ensure that your revised manuscript can be swiftly handed over to our production team.

In recognition of the time and expertise our reviewers provide to Nature Cell Biology's editorial process, we would like to formally acknowledge their contribution to the external peer review of your manuscript entitled "SARS-CoV-2 infection causes DNA damage through multiple mechanisms". For those reviewers who give their assent, we will be publishing their names alongside the published article.

Nature Cell Biology offers a Transparent Peer Review option for new original research manuscripts submitted after December 1st, 2019. As part of this initiative, we encourage our authors to support increased transparency into the peer review process by agreeing to have the reviewer comments, author rebuttal letters, and editorial decision letters published as a Supplementary item. When you submit your final files please clearly state in your cover letter whether or not you would like to participate in this initiative. Please note that failure to state your preference will result in delays in accepting your manuscript for publication.

Cover suggestions

As you prepare your final files we encourage you to consider whether you have any images or illustrations that may be appropriate for use on the cover of Nature Cell Biology.

Nature Cell Biology has now transitioned to a unified Rights Collection system which will allow our Author Services team to quickly and easily collect the rights and permissions required to publish your work. Approximately 10 days after your paper is formally accepted, you will receive an email in providing you with a link to complete the grant of rights. If your paper is eligible for Open Access, our Author Services team will also be in touch regarding any additional information that may be required to arrange payment for your article.

Please note that *Nature Cell Biology* is a Transformative Journal (TJ). Authors may publish their research with us through the traditional subscription access route or make their paper immediately open access through payment of an article-processing charge (APC). Authors will not be required to make a final decision about access to their article until it has been accepted. Find out more about Transformative Journals

Please use the following link for uploading these materials:
[REDACTED]

Best regards,

Kendra Donahue
Staff
Nature Cell Biology

On behalf of

Melina Casadio, PhD

Senior Editor, Nature Cell Biology

ORCID ID: <https://orcid.org/0000-0003-2389-2243>

Reviewer #1:

Remarks to the Author:

In previous version, the authors only presented that SARS-CoV-2 infection can cause DNA damage, with mechanistic aspects remain unaddressed, very confusing or even not supported by their data. In this revised version, point-by-point responses were provided by the authors. However, I still found this revision, in quality, has not reached the level of Nature Cell Biology, and the overall logic and data reliability need to be further strengthened. Particularly in several places that they simply deleted the previous data and replaced it with new images or inconsistent statistical results, and they could not give a reasonable explanation for the concerns raised.

The first and most important concern is, multiple reports have been published with their major conclusions related to DNA damage responses caused by SARS-CoV-2, exemplified in PMID: 35045565 and PMID: 36254583 and also others not listed; Thus current study already lose quite of its novelty. Secondly, although some new data have been added to the returned version, the most important mice infection experiment with/without IR has not been performed. The third, many findings are not clear and the molecular mechanism (as also raised by reviewer #2 and #3) is still confusing.

In conclusion, I still feel regrettably that, there are many problems with the manuscript (logic and data reliability) and many open questions, which should be investigated carefully. Besides, a number of concerns are listed below:

Major points:

Q1: The authors did not explain why chk2 phosphorylation was not detected within 48 hours after viral infection. As described in the literature cited by the author, ATM mediated photosynthesis of T68 and promote CHK2 activation. At the 24 and 48 h time points after virus infection, several DDR markers as well as ATM are activated. (Do the authors believe that there is difference between DNA damage caused by SARS2 virus and IR?)

Q2: RAD51 is an important marker in the process of DNA damage, it is necessary to put this result in the text and explain the possible mechanism. On the other hand, the representative immunofluorescence image given by the author shows that when there is no viral infection, the cells already have about 50% RAD51 foci. This is apparently not so reliable.

Q3: "Whether CHK1 overexpression in Hun7 cells can rescue the SARS-CoV-2-induced the pro-inflammatory cytokines expression, RRM2 reduction and impaired S-phase progression? It would be an added value to the manuscript."

In this article cited by the author, most of the cells transfected with full-length Chk1 or kinase-dead truncated Chk1-(1-299/D130A) displayed no γ -H2AX signals. This result is inconsistent with the result shown by the authors. The cleavage bands shown may be a degradation of the chk1 protein. I am worried about this result.

Q5: Where is the expression of NSP13? IN Fig S3B, The "N" marked on the x-axis should be "M".

Q6: Although the authors have performed some supplementary experiments to explain the possible

molecular mechanism, it is not clear and insightful. How ORF6 affect the ubiquitination level of CHK1 (especially the K48 linked polyubiquitin chain)? How does NSP13 degrade CHK1 through autophagy pathway? These important issues are worthy of further study and are very important to explain the core molecular mechanism of their findings. I believe every NCB paper should have profound mechanism which should also be novel, not just presenting some phenotypes.

Q9: Although the author has replaced the figure, it is difficult to believe the conclusion drawn by the author because of the inconsistent localization of N protein in the same assay.

Q10: In experiment using Calu-3 cells and in vivo experiment, SARS-CoV-2 infection cannot cause the reduction of 53BP1 foci (Fig S4A-B and Fig 5). This conclusion is inconsistent with the new results in the revised figures (Fig. S5A, B). In the same experiment of the previous version, the author drew different conclusions in this version, which makes it difficult to believe the authenticity of the data.

Q16: The authors seem not answering this concern. Are there more important upstream proteins in the DNA damage pathway that are affected? Was the type of DNA damage caused by SARS-COV-2 mainly double-strand breaks?

Minor points:

Q5: The representative images of lung and nasal mucosal tissues in figure 6C and 6E are quite different from those in figure S6A-B. Please provide an explanation. Why did the author delete the original high-quality figure S6A 6B in the revision?

Reviewer #2:

Remarks to the Author:

The authors have performed an extensive work to respond to the issues raised and have adequately addressed them. Therefore I have no more comments and recommend publication of this manuscript.

Reviewer #3:

Remarks to the Author:

The authors have addressed my comments. In particular, the use of COVID-19 patients' lungs and nasal mucosa is a significant addition to the analysis performed with cancer cell lines. The use of mice infected with SARS-CoV-2 is also a strength of the manuscript (mouse data were already present in the original version). I consider that the manuscript is now suitable for publication, pending minor points (see below). I believe that this manuscript presents important information to broadly understand host-viral pathogen interactions, as they relate specifically to the induction of a DNA damage response (DDR). Most viruses will need to develop mechanisms to modulate the DDR, as exemplified here.

Minor Point:

The figures seem too busy. The main figures often extend over 2 pages each and the font is already too small. Fig 2 covers 3 pages. How will these figures be visible on the print form of the journal? The authors need to invest some effort in preparing better figures. They can also print them at the size in

which they would be at the printed journal to see if they are readable.

Additional comments to the authors in response to the comments of Reviewer #1

According to Reviewer 1: "The first and most important concern is, multiple reports have been published with their major conclusions related to DNA damage responses caused by SARS-CoV-2, exemplified in PMID: 35045565 and PMID: 36254583 and also others not listed; Thus current study already lose quite of its novelty."

Essentially the Reviewer is saying that the authors have been scooped. However, I looked at these publications. The first one mentions a role of cGAS-STING, which the authors of this manuscript also discuss, but otherwise the main focus of the two studies is different. Thus, I don't think that the novelty of the current manuscript is affected in a significant way. The second study examines why SARS-CoV-2 affects aged individuals more severely than young individuals and has nothing to do with the current manuscript. Thus, I don't share this "first and most important" concern with Reviewer 1.

Major points

Q1: The Reviewer refers to ATM-mediated "photosynthesis". This novel function of ATM deserves to be published in Nature, since mammals are generally not known to perform photosynthesis, except, of course, if the Reviewer did not check what he/she was writing, which I am afraid may be the case. The Reviewer concludes this point by the following question: "Do the authors believe that there is difference between DNA damage caused by SARS2 virus and IR?" I don't understand why the Reviewer is asking this question. The main point of the manuscript is that there is a different DDR and DNA damage in cells infected by SARS2 versus those exposed to IR. So the answer to the Reviewer's question is a definite Yes and the answer is supported by the data shown.

Q2: The Reviewer mentions that RAD51 foci is an important marker in the process of DNA damage, but it not happy with the basal levels of RAD51 foci. In fact, scoring RAD51 foci is known to be problematic. One needs to determine for each cell line, a threshold for the number of RAD51 foci, above which a cell would be considered RAD51-positive. Otherwise, every replicating cell will have a small number (1-10, depending on cell line) of RAD51 foci. The DDR markers that are most relevant to this manuscript are Chk1, Chk2 and 53BP1, not RAD51. The authors examined RAD51 foci, because Reviewer 1 requested them to do so. I personally don't think that the RAD51 data should be included in the final version of the manuscript, which is also the opinion of the authors.

Q3: In this comment Reviewer 1 requested two different experiments. The authors did the first experiment and the results supported very nicely the main premise of the study. They also attempted to do the second experiment, but they found that ectopic expression of Chk1 causes DNA damage on its own, so it was not possible to proceed further. This is not a major point of the study and I don't see why Reviewer 1 is making an issue out of this.

Q5: It seems that the Reviewer has spotted a typo in Fig. S3B. The authors should correct the typo.

Q6: The Reviewer says that the authors have not provided the "profound mechanism which should also be novel" that "every NCB paper should have". The mechanism relates to how ORF6 and NSP13

target Chk1 for degradation. The authors provide a lot of mechanistic information in Figs 4 and 5. I am satisfied by the mechanistic data provided.

Q9: The Reviewer considers that there is inconsistent localization of the N protein in various experiments. This is also true in the literature, as the authors state and could relate to the levels of expression of the protein under various conditions. Since the N protein is present in both the nucleus and the cytoplasm (based on the literature and the staining performed by the authors), I don't see how this concern impacts the conclusions of the authors.

Q10: The Reviewer states that 53BP1 behaves differently in different cell lines, but the authors clearly show that the 53BP1 foci signal decreases in all cell lines and that 53BP1 co-localization with gH2AX foci is reduced in all cell lines. I don't understand what the Reviewer is worried about.

Q16: The Reviewer says that the authors do not seem to answer the questions: "Are there more important upstream proteins in the DNA damage pathway that are affected? Was the type of DNA damage caused by SARS-COV-2 mainly double-strand breaks?" However, the authors answered this question and the answer is also in the abstract of the manuscript.

Minor points:

Q5: It seems that the authors obtained better images in the course of revising the manuscript.

Author Rebuttal, Third Revision:

We would like to thank the Editor and all the reviewers for their suggestions, which, we believe, improved the scientific quality of our manuscript.

Reviewer #1 (Remarks to the Author):

In previous version, the authors only presented that SARS-CoV-2 infection can cause DNA damage, with mechanistic aspects remain unaddressed, very confusing or even not supported by their data. In this revised version, point-by-point responses were provided by the authors. However, I still found this revision, in quality, has not reached the level of Nature Cell Biology, and the overall logic and data reliability need to be further strengthened. Particularly in several places that they simply deleted the previous data and replaced it with new images or inconsistent statistical results, and they could not give a reasonable explanation for the concerns raised.

The first and most important concern is, multiple reports have been published with their major conclusions related to DNA damage responses caused by SARS-CoV-2, exemplified in PMID: 35045565 and PMID: 36254583 and also others not listed; Thus current study already lose quite of its novelty.

We think that the main focus of the two papers here mentioned is different from ours. In particular, the first one describes the role of the cGAS-STING pathway in activating an inflammatory response, while the second study provides a possible explanation why SARS-CoV-2 infection affects individuals in an age-associated manner.

More importantly, we believe that the novelty of our manuscript lies in providing mechanistic insights behind DNA damage accumulation, activation of the inflammatory pathways and cellular senescence, events that are widely observed following SARS-CoV-2 infection. We discovered that SARS-CoV-2 virus hijacks nucleotides metabolism, the most fundamental machinery of the cell, to promote its own replication by causing CHK1 degradation. We identified two distinct viral products (ORF6 and NSP13) that cause CHK1 loss, thus reducing the expression of ribonucleotide reductase enzyme (RRM2) that turns rNTP into dNTP. The consequences are reduced dNTP supply, DNA replication stress, DDR activation, cellular senescence and expression of proinflammatory cytokines, INCLUDING cGAS/STING engagement. CHK1 inactivation is sufficient to recapitulate these events, and dNTP supplementation rescues these phenotypes in infected cells, overall establishing a clear causative link among the events we uncovered.

We also characterized two novel mechanisms by which SARS-CoV-2 infection leads to CHK1 degradation: ORF6 protein-mediated proteasomal degradation and NSP13-mediated autophagic route.

In addition to these mechanisms of DNA damage generation, we also discovered that DNA damage repair is impaired due to the interference of SARS-CoV-2 N protein in binding damage-induced RNAs that control 53BP1 liquid-liquid phase separation events.

We believe all these results are novel and relevant.

Secondly, although some new data have been added to the returned version, the most important mice infection experiment with/without IR has not been performed.

As explained previously, very few mouse facilities have a BSL-3 (biosafety level 3) designation that allow them to handle mice infected with SARS-CoV-2 – we were lucky to find one for our experiments. Among those we identified and we could access, none had an irradiator inside the BSL-3 area, rendering the feasibility of this experiment an insurmountable obstacle.

The third, many findings are not clear and the molecular mechanism (as also raised by reviewer #2 and #3) is still confusing.

We believe we have provided an extensive *in vitro* characterization of three different mechanisms (each one involving a specific SARS-CoV-2 product), underlying the events observed following viral infection, and validated them in two *in vivo* model systems (infected mice and COVID-19 patients).

In conclusion, I still feel regrettably that, there are many problems with the manuscript (logic and data reliability) and many open questions, which should be investigated carefully. Besides, a number of concerns are listed below:

Major points:

Q1: The authors did not explain why chk2 phosphorylation was not detected within 48 hours after viral infection. As described in the literature cited by the author, ATM mediated photosynthesis of T68 and promote CHK2 activation. At the 24 and 48 h time points after virus infection, several DDR markers as well as ATM are activated. (Do the authors believe that there is difference between DNA damage caused by SARS2 virus and IR?)

We take this opportunity to clarify this point: upon IR, threonine 68 of CHK2 is promptly but transiently phosphorylated (compare 15' post-IR and 60' post-IR, in agreement with the published literature:

<https://pubmed.ncbi.nlm.nih.gov/12805407/>;

<https://pubmed.ncbi.nlm.nih.gov/25404613/>). Upon viral infection, at the 1-hour time point studied, no DDR marker is activated, indicating that no DNA damage has yet been generated, thus explaining also the lack of CHK2 phosphorylation. At the 24 and 48 hours time points studied, several DDR markers are activated (such as pKAP1 and gH2AX) but CHK2 is not phosphorylated. Therefore, looks like that SARS-CoV-2 infection impacts on genome integrity differently from transient and acute IR.

Q2: RAD51 is an important marker in the process of DNA damage, it is necessary to put this result in the text and explain the possible mechanism. On the other hand, the representative immunofluorescence image given by the author shows that when there is no viral infection, the cells already have about 50% RAD51 foci. This is apparently not so reliable.

We investigated RAD51 levels as requested by this Reviewer, but the potential mechanisms causing RAD51 reduction and why Huh7 cells have basal levels of RAD51 foci is, we believe, beyond the scope of the present manuscript.

Q3: “Whether CHK1 overexpression in Hun7 cells can rescue the SARS-CoV-2-induced the pro-inflammatory cytokines expression, RRM2 reduction and impaired S-phase progression? It would be an added value to the manuscript.”

In this article cited by the author, most of the cells transfected with full-length Chk1 or kinase-dead truncated Chk1-(1–299/D130A) displayed no γ -H2AX signals. This result is inconsistent with the result shown by the authors. The cleavage bands shown may be a degradation of the chk1 protein. I am worried about this result. **In the paper that we mentioned, the authors showed that the over-expression of the truncated form of CHK1 (1-299, the N-terminal cleavage fragment and not the kinase-dead) is inducing high levels of γ -H2AX signal. The CHK1 fragments observed in our experiments are consistent with the reported apoptosis-induced cleavage of CHK1, which is associated with DNA damage accumulation, making results hard to be interpreted.**

Q5: Where is the expression of NSP13? IN Fig S3B, The “N” marked on the x-axis should be “M”.

NSP13 expression was already shown in Fig. 3 (now called Fig. 4).

The N marked on the x-axis is correct. We have now specified, in the legend to the new Extended Data Figure 4, that Huh7 cells did not express detectable levels of NSP4, NSP11, ORF9c, ORF10, M and therefore CHK1 and RRM2 protein levels were not quantified in these samples.

Q6: Although the authors have performed some supplementary experiments to explain the possible molecular mechanism, it is not clear and insightful. How ORF6 affect the ubiquitination level of CHK1 (especially the K48 linked polyubiquitin chain)? How does NSP13 degrade CHK1 through autophagy pathway? These important issues are worthy of further study and are very important to explain the core molecular mechanism of their findings. I believe every NCB paper should have profound mechanism which should also be novel, not just presenting some phenotypes.

Prompted by this and the other reviewers, we have extensively worked to uncover two novel mechanisms by which SARS-CoV-2 infection causes CHK1 degradation. The first one involves ORF6 that, by interacting with the nuclear pore complex, disrupts protein trafficking (<https://www.pnas.org/doi/10.1073/pnas.2016650117>)

and prevents CHK1 translocation from the cytoplasm to the nucleus, causing its degradation via the proteasome. Indeed, treatment with the proteasome inhibitor MG132 leads to the accumulation of poly-ubiquitinated-CHK1, ultimately rescuing CHK1 levels in ORF6 expressing cells. In addition, a point mutation in ORF6 disrupting its interaction with the nuclear pore complex (<https://www.pnas.org/doi/10.1073/pnas.2016650117>) is sufficient to abolish ORF6 impact on CHK1 levels and on DNA damage generation.

The other mechanism unveiled is NSP13 downregulating CHK1 protein levels through the autophagic route. Indeed, autophagy inhibition, either through a set of specific pharmacological inhibitors or RNAi-mediated knock down of key autophagy components, impedes CHK1 loss caused by NSP13 expression (new Figure 5).

Overall, we did our best to produce sufficient evidence to address referees' requests on the mechanisms of SARS-CoV-2-triggered CHK1 degradation.

Q9: Although the author has replaced the figure, it is difficult to believe the conclusion drawn by the author because of the inconsistent localization of N protein in the same assay.

SARS-CoV-2 N protein is abundantly expressed mainly in the cytoplasm of infected cells, and we demonstrated also its nuclear localization by confocal microscopy, consistent with recent reports cited in the rebuttal (<https://www.nature.com/articles/s43587-022-00170-7> <https://www.embopress.org/doi/full/10.15252/msb.202110396> <https://journals.plos.org/plosbiology/article?id=10.1371/journal.pbio.3001158>).

Q10: In experiment using Calu-3 cells and in vivo experiment, SARS-CoV-2 infection cannot cause the reduction of 53BP1 foci (Fig S4A-B and Fig 5). This conclusion is inconsistent with the new results in the revised figures (Fig. S5A, B). In the same experiment of the previous version, the author drew different conclusions in this version, which makes it difficult to believe the authenticity of the data.

We strengthened this observation in Calu-3 cells by performing an additional experiment of infection that, combined with the previous ones, showed a statistically significant reduction of 53BP1 foci in infected Calu-3. These data are strongly supported by the results obtained in Huh7 cells (both infected and transfected with the SARS-CoV-2 N-protein), HNEpC and in vivo in mice and COVID-19 patients.

Q16: The authors seem not answering this concern. Are there more important upstream proteins in the DNA damage pathway that are affected? Was the type of DNA damage caused by SARS-COV-2 mainly double-strand breaks?

We analyzed what we think are the most upstream DDR proteins: ATM, ATR and DNA-PK. We analysed both SSB and DSB by alkaline comet assays, as specified in the manuscript.

Minor points:

Q5: The representative images of lung and nasal mucosal tissues in figure 6C and 6E are quite different from those in figure S6A-B. Please provide an explanation. Why did the author delete the original high-quality figure S6A 6B in the revision?
We replaced these images with better and more representative ones.

Reviewer #2 (Remarks to the Author):

The authors have performed an extensive work to respond to the issues raised and have adequately addressed them. Therefore I have no more comments and recommend publication of this manuscript.

Reviewer #3 (Remarks to the Author):

The authors have addressed my comments. In particular, the use of COVID-19 patients' lungs and nasal mucosa is a significant addition to the analysis performed with cancer cell lines. The use of mice infected with SARS-CoV-2 is also a strength of the manuscript (mouse data were already present in the original version). I consider that the manuscript is now suitable for publication, pending minor points (see below). I believe that this manuscript presents important information to broadly understand host-viral pathogen interactions, as they relate specifically to the induction of a DNA

damage response (DDR). Most viruses will need to develop mechanisms to modulate the DDR, as exemplified here.

Minor Point:

The figures seem too busy. The main figures often extend over 2 pages each and the font is already too small. Fig 2 covers 3 pages. How will these figures be visible on the print form of the journal? The authors need to invest some effort in preparing better figures. They can also print them at the size in which they would be at the printed journal to see if they are readable.

We extensively worked to improve the quality and the organization of the figures.

Additional comments to the authors in response to the comments of Reviewer #1

According to Reviewer 1: "The first and most important concern is, multiple reports have been published with their major conclusions related to DNA damage responses caused by SARS-CoV-2, exemplified in PMID: 35045565 and PMID: 36254583 and also others not listed; Thus current study already lose quite of its novelty."

Essentially the Reviewer is saying that the authors have been scooped. However, I looked at these publications. The first one mentions a role of cGAS-STING, which the authors of this manuscript also discuss, but otherwise the main focus of the two studies is different. Thus, I don't think that the novelty of the current manuscript is affected in a significant way. The second study examines why SARS-CoV-2 affects aged individuals more severely than young individuals and has nothing to do with the current manuscript. Thus, I don't share this "first and most important" concern with Reviewer 1.

Major points

Q1: The Reviewer refers to ATM-mediated "photosynthesis". This novel function of ATM deserves to be published in Nature, since mammals are generally not known to perform photosynthesis, except, of course, if the Reviewer did not check what he/she was writing, which I am afraid may be the case. The Reviewer concludes this point by the following question: "Do the authors believe that there is difference between DNA damage caused by SARS2 virus and IR?" I don't understand why the Reviewer is asking this question. The main point of the manuscript is that there is a different DDR and DNA damage in cells infected by SARS2 versus those exposed to IR. So the answer to the Reviewer's question is a definite Yes and the answer is supported by the data shown.

Q2: The Reviewer mentions that RAD51 foci is an important marker in the process of DNA damage, but it not happy with the basal levels of RAD51 foci. In fact, scoring RAD51 foci is known to be problematic. One needs to determine for each cell line, a threshold for the number of RAD51 foci, above which a cell would be considered RAD51-positive. Otherwise, every replicating cell will have a small number (1-10, depending on cell line) of RAD51 foci. The DDR markers that are most relevant to this manuscript are Chk1, Chk2 and 53BP1, not RAD51. The authors examined RAD51 foci, because Reviewer 1 requested them to do so. I personally don't think that the

RAD51 data should be included in the final version of the manuscript, which is also the opinion of the authors.

Q3: In this comment Reviewer 1 requested two different experiments. The authors did the first experiment and the results supported very nicely the main premise of the study. They also attempted to do the second experiment, but they found that ectopic expression of Chk1 causes DNA damage on its own, so it was not possible to proceed further. This is not a major point of the study and I don't see why Reviewer 1 is making an issue out of this.

Q5: It seems that the Reviewer has spotted a typo in Fig. S3B. The authors should correct the typo.

In fact, this is not a typo. The label refers to viral N protein. However, we have edited the legend of this figure to improve its clarity.

Q6: The Reviewer says that the authors have not provided the "profound mechanism which should also be novel" that "every NCB paper should have". The mechanism relates to how ORF6 and NSP13 target Chk1 for degradation. The authors provide a lot of mechanistic information in Figs 4 and 5. I am satisfied by the mechanistic data provided.

Q9: The Reviewer considers that there is inconsistent localization of the N protein in various experiments. This is also true in the literature, as the authors state and could relate to the levels of expression of the protein under various conditions. Since the N protein is present in both the nucleus and the cytoplasm (based on the literature and the staining performed by the authors), I don't see how this concern impacts the conclusions of the authors.

Q10: The Reviewer states that 53BP1 behaves differently in different cell lines, but the authors clearly show that the 53BP1 foci signal decreases in all cell lines and that 53BP1 co-localization with gH2AX foci is reduced in all cell lines. I don't understand what the Reviewer is worried about.

Q16: The Reviewer says that the authors do not seem to answer the questions: "Are there more important upstream proteins in the DNA damage pathway that are affected? Was the type of DNA damage caused by SARS-COV-2 mainly double-strand breaks?" However, the authors answered this question and the answer is also in the abstract of the manuscript.

Minor points:

Q5: It seems that the authors obtained better images in the course of revising the manuscript.

Final Decision Letter:

Dear Dr d'Adda di Fagagna,

I am pleased to inform you that your manuscript, "SARS-CoV-2 infection induces DNA damage, through CHK1 degradation and impaired 53BP1 recruitment, and cellular senescence", has now been accepted for publication in Nature Cell Biology. Congratulations on this very nice work!

Please note that *Nature Cell Biology* is a Transformative Journal (TJ). Authors may publish their research with us through the traditional subscription access route or make their paper immediately

open access through payment of an article-processing charge (APC). Authors will not be required to make a final decision about access to their article until it has been accepted. Find out more about Transformative Journals

If you have not already done so, we strongly recommend that you upload the step-by-step protocols used in this manuscript to the Protocol Exchange (www.nature.com/protocolexchange), an open online resource established by Nature Protocols that allows researchers to share their detailed experimental know-how. All uploaded protocols are made freely available, assigned DOIs for ease of citation and are fully searchable through nature.com. Protocols and Nature Portfolio journal papers in which they are used can be linked to one another, and this link is clearly and prominently visible in the online versions of both papers. Authors who performed the specific experiments can act as primary authors for the Protocol as they will be best placed to share the methodology details, but the Corresponding Author of the present research paper should be included as one of the authors. By uploading your Protocols to Protocol Exchange, you are enabling researchers to more readily reproduce or adapt the methodology you use, as well as increasing the visibility of your protocols and papers. You can also establish a dedicated page to collect your lab Protocols. Further information can be found at www.nature.com/protocolexchange/about

With kind regards,

Melina

Melina Casadio, PhD

Senior Editor, Nature Cell Biology

ORCID ID: <https://orcid.org/0000-0003-2389-2243>

Click here if you would like to recommend Nature Cell Biology to your librarian

<http://www.nature.com/subscriptions/recommend.html#forms>

** Visit the Springer Nature Editorial and Publishing website at www.springernature.com/editorial-and-publishing-jobs for more information about our career opportunities. If you have any questions please click here.**